# Individual Regret in Cooperative Stochastic Multi-Armed Bandits

**Idan Barnea**
Blavatnik School of Computer Science and AI
Tel Aviv University, Israel
idanbarnea1@mail.tau.ac.il

**Tal Lancewicki**
Blavatnik School of Computer Science and AI
Tel Aviv University, Israel
lancewicki@mail.tau.ac.il

**Yishay Mansour**
Blavatnik School of Computer Science and AI
Tel Aviv University, Israel
Google Research, Tel Aviv, Israel

## Abstract

We study the regret in stochastic Multi-Armed Bandits (MAB) with multiple agents that communicate over an arbitrary connected communication graph. We analyzed a variant of Cooperative Successive Elimination algorithm, `Coop-SE`, and show an individual regret bound of $O(\mathcal{R}/m + A^2 + A\sqrt{\log T})$ and a nearly matching lower bound. Here $A$ is the number of actions, $T$ the time horizon, $m$ the number of agents, and $\mathcal{R} = \sum_{\Delta_i > 0} \log(T)/\Delta_i$ is the optimal single agent regret, where $\Delta_i$ is the sub-optimality gap of action $i$. Our work is the first to show an individual regret bound in cooperative stochastic MAB that is independent of the graph's diameter.

When considering communication networks there are additional considerations beyond regret, such as message size and number of communication rounds. First, we show that our regret bound holds even if we restrict the messages to be of logarithmic size. Second, for logarithmic number of communication rounds, we obtain a regret bound of $O(\mathcal{R}/m + A\log T)$.

## 1   Introduction

Multi-Armed Bandit (MAB) is a fundamental framework for studying sequential decision making, with an expanding scope of practical applications (see, [22, 33]). Recent research expanded the classic MAB problem into a cooperative setting, sometimes referred to as cooperative multiplayer or multi-agent MAB, where multiple agents share the same goal and can communicate with each other.

A significant focus of recent research has centered on cooperating agents within a communication graph, often referred to as a communication network. This framework, in which all agents address the same problem, dates back to Landgren et al. [18] for stochastic rewards and Cesa-Bianchi et al. [5] for the nonstochastic case. In this setting, agents transmit information to adjacent neighbors, from which it continues to propagate throughout the entire network while encountering a delay at each step. Communication graphs paired with stochastic Multi-Armed Bandits provide a framework for distributed decision-making under uncertainty. As an example of our setting, consider computer networks. In large-scale High Performance Computing (HPC) or AI systems, individual machines often have flexible hardware, e.g., tunable cores, caches, or memory controllers, that adapt based on workload pattern. These computers (agents), connected over a network (graph), must quickly choose a hardware configuration (action) to optimize their performance (reward). Social networks are good examples as well: individuals share experiences directly with friends, forming a natural

communication structure for learning to propagate and improve collective decision-making. In the non-stochastic setting Bar-On and Mansour [3] showed a nearly optimal individual regret bound of $\tilde{O}(\sqrt{(1 + A/N(v))T})$, where $N(v)$ is the number neighbors for the agent $v$. This setting differs from the stochastic case: notably, the optimal bound in the non-stochastic setting includes a $\sqrt{T}$ term, even if $m \approx T$, since the full information lower bound is $\Omega(\sqrt{T})$. On the other hand, in the stochastic setting, we achieve something much stronger. With sufficiently many agents, our worst-case regret can be as small as $O(A^2 + A\sqrt{\log(T)})$. Importantly, this is independent of the sub-optimality gaps.

The literature of cooperative stochastic MAB distinguishes between group regret (a.k.a. average regret) [18–21, 6, 29, 35, 39, 8] and individual regret [11, 36], where the latter is much stronger and more challenging to achieve. Additionally, significant attention is given to minimizing the number of messages each agent sends [32, 7, 25, 26, 28, 1, 30] and reducing message size [1]

## 1.1 Graph diameter and individual regret bounds

Let the single-agent regret bound be denoted with $\mathcal{R} := \sum_i \log(T)/\Delta_i$. For cooperation to be meaningful, one needs sufficiently large number of agents $m$. Our goal is to reduce the individual regret from $\mathcal{R}$ to $\mathcal{R}/m$. A potential problem can be if regret bounds include an additive term of the order of $D$, the graph's diameter, which can be large in practical scenarios. A regret of the form $\mathcal{R}/m + D$ might provide a limited guarantee: for a cycle graph ($D = \Theta(m)$) it will give $\mathcal{R}/m + m$ which is at least $\sqrt{\mathcal{R}} = (\sum_i \frac{\log T}{\Delta_i})^{1/2}$ for any $m$; for a grid graph ($D = \Theta(\sqrt{m})$) the regret will be at least $\mathcal{R}^{1/3} = (\sum_i \frac{\log T}{\Delta_i})^{1/3}$. Note that in these scenarios, the inverse dependency on $\Delta_i$ remains, even if the number of agents $m$ goes to infinity and the term $\mathcal{R}/m$ vanishes. This explains our desire to avoid this additive $D$.

Our diameter-free bound is only $A^2 + A\sqrt{\log(T)}$ for sufficiently large $m$, entirely independent of the gaps. An interesting case is when the gaps are small, leading to high single-agent regret. For example, when $\Delta_i = \sqrt{A/T}$, we get $\mathcal{R} \approx \sqrt{AT}$. In this case, regret bounds with an additive $D$ term may still yield $\mathcal{R}^{1/3} \approx (AT)^{1/6}$ for grid graphs, whereas our diameter-free bound grows is only $A^2 + A\sqrt{\log(T)}$.

To the best of our knowledge, this is the first paper to show a graph-independent individual regret bound. Additionally, we present a similar individual regret bound for scenarios of small message size, as well as for scenarios of limited number of messages.

## 1.2 Key contributions

Our key contributions are as follows:

- We prove that `Coop-SE` (Algorithm 3) achieves a near-optimal individual regret bound of $O(\mathcal{R}/m + A^2 + A\sqrt{\log(T)})$, which is independent of the graph's diameter. The regret bound in minimax form is $O(\sqrt{TA\log(T)/m} + A^2 + A\sqrt{\log(T)})$. When `Coop-SE` is played with random action choices instead of round-robin we get $O(\mathcal{R}/m + A\log(T))$ and accordingly, $O(\sqrt{TA\log(T)/m} + A\log(T))$.

- We show a lower bound for the individual regret of $\Omega(\sqrt{TA/m} + \sqrt{A})$, which almost matches our upper bound in the minimax form.

- For settings with restricted message sizes of $O(\log(mA))$, also known of the CONGEST model (see [31]), we introduce `Coop-SE-CONGEST`, which achieves an individual regret of $O(\mathcal{R}/m + A^2 + A\sqrt{\log(T)})$. In minimax form $O(\sqrt{TA\log(T)/m} + A^2 + A\sqrt{\log(T)})$.

- For scenarios where agents are limited to $O(\log(T))$ communication rounds, we present `Coop-SE-Comm-Cost`, that achieves individual regret of $O(\log(A)\mathcal{R}/m + A\log(A)\log(T))$, and in the minimax form $O(\sqrt{TA\log(A)\log(T)/m} + A\log(A)\log(T))$.

Kolla et al. [16] raised the question of whether it is feasible to surpass the performance of well-established single-agent policies, such as UCB [2] and SE, when these policies are executed independently across the network. While this question has also been explored in several prior works (see, for example, [38–40]), our contribution provides a complementary perspective by analyzing individual

Table 1: Performance Comparison of Multi-Armed Bandit Algorithms in Cooperative Settings. Notation: Horizon $T$; Number of agents $m$; Actions $A$; Graph's diameter $D$; Graph $\mathcal{G}$; $\mathcal{R} = \sum_{\Delta_i > 0} \frac{\log T}{\Delta_i}$ is the optimal single agent instance-dependent regret.

| Algorithm | Regret | Indiv. regret | Message size | Comm. rounds | Requires only local graph info. |
|---|---|---|---|---|---|
| Coop-UCB2 [21] | $\mathcal{R}/m + Af(\mathcal{G})^{\dagger}$ | ✗ | $A\log(mT)$ | $T$ | ✗ |
| DDUCB [29] | $\mathcal{R}/m + Ah(\mathcal{G})^{\dagger}$ | ✗ | $A\log(mT)$ | $T$ | ✗ |
| UCB-TCOM [36] | $\mathcal{R}/m + AD$ | ✓ | $m\log(AT)$ | $D + \log(\frac{\log T}{\Delta_{min}})$ | ✗ |
| Coop-SE | $\mathcal{R}/m$ $+A\min\{A + \sqrt{\log T}, D\}$ | ✓ | $mA\log(T)$ | $T$ | ✓ |
| Coop-SE-CONGEST | $\mathcal{R}/m + A^2 + A\sqrt{\log T}$ | ✓ | $\log(\mathbf{mA})$ | $T$ | ✗ |
| Coop-SE-Comm-Cost | $\mathcal{R}\log(\mathcal{A})/m$ $+A\log T \log A$ | ✓ | $A\log(m)$ | $\log(T)$ | ✗ |

regret that is independent of the graph's diameter. The combination of our lower bound and the analysis of our `Coop-SE` algorithm thus refines the understanding of this question in the cooperative setting.

## 1.3 Relation to Prior Algorithms

We start with a brief explanation of the Successive Elimination (SE) algorithm [13]. SE is a classical multi-armed bandit algorithm that achieves low regret for a single agent. It progressively eliminates suboptimal actions by repeatedly sampling all active actions, estimating their mean rewards, and removing any action whose confidence interval is clearly worse than that of some other action. In cooperative multi-agent settings, variants of SE have been studied in which agents exchange rewards and elimination signals and use them to refine their own action sets Yang et al. [38, 39], Zhang et al. [40]. Our proposed algorithm, `Coop-SE`, builds upon these works: it employs Successive Elimination combined with message passing, and allows each agent to use the elimination signals of others in order to eliminate its own actions. The main contribution of this paper is a new analysis of the `Coop-SE` algorithm.

## 1.4 Related work

**Average regret** was studied by Landgren et al. [18, 19, 20, 21], Martínez-Rubio et al. [29], Chen et al. [8] who achieved an average regret guarantee of $O(\mathcal{R}/m + A\tilde{f}(\mathcal{G}))$, where $\tilde{f}(\mathcal{G})$ is a function of the eigenvalues of the adjacency matrix of the graph, which is related to expansion properties. The consensus-based algorithm `Coop-UCB2` presented in Landgren et al. [21] requires the construction of a matrix based on the graph's structure. This dependency means the algorithm cannot rely solely on local information. The algorithm's regret bound is $\mathcal{R}/m + A \cdot f(\mathcal{G})$, where $f(\mathcal{G})$ represents a graph-dependent function. For certain graph topologies, such as cycles, this function $f(\mathcal{G})$ may be at least $m$. Martínez-Rubio et al. [29] presented `DDUCB`, a consensus-based algorithm that requires knowledge of the graph's topology. Their regret bound is $\mathcal{R}/m + A \cdot h(\mathcal{G})$, where the function $h$ is defined as $h(\mathcal{G}) = \log(m)/\sqrt{\log(1/|\lambda_2|)}$. Here, $\lambda_2$ is the second largest eigenvalue (in absolute value) of the communication matrix, also known as the gossip matrix (see [37, 12, 7]). The eigenvalue $\lambda_2$ is related to how the graph expands and can be very close to one for some graphs. For instance, in a circle graph, $\lambda_2 = \cos(2\pi/m) \approx 1 - 1/m^2$, resulting in $h(\mathcal{G}) \approx m$. Consequently, the average regret bound of `DDUCB` becomes $\mathcal{R}/m + Am$.

Gossip algorithms traditionally operate through networks where nodes communicate along graph edges to achieve consensus, typically by converging to average values across all nodes. However, the `Coop-SE` algorithm takes a different approach. Instead of seeking to synchronize nodes to common average values, we focus on ensuring nodes maintain similar sets of active actions. This key distinction drives the innovation in our approach.

Average regret was also studied by [35, 6] as well. Wang et al. [35] has an additive term in the regret that scales with the diameter of the graph. Chakraborty et al. [6] consider a model with non-fresh randomness, where the reward for each action is generated once per timestep, and agents choosing the same action receive the same feedback. Even with full communication, the best attainable regret is that of full information.

**Small number of messages and small messages.** Wang et al. [36] show an individual regret guarantee of $O(\mathcal{R}/m + AD)$, where each agent sends at most $O(D\log(\log(T)/\Delta))$ messages. Their algorithm, `UCB-TCOM`, needs to know the value of $D$ in advance and uses it to synchronize between agents. Yang et al. [39] present a SE algorithm achieving a similar regret bound, but their analysis is limited to fully-connected graphs, with each agent sending $O(\log(1/\Delta))$ messages. Note that whenever $\Delta = \Theta(T^{-\alpha})$ for $\alpha \in (0,1)$, the number of messages in both of these works is of order of $\log(T)$. Madhushani and Leonard [26] introduces an algorithm with $\log(T)$ communication steps, and their regret scales as $\log(T)\chi(G)/(\Delta m)$, where $\chi(G)$ is the clique cover number of the graph. For instance, for trees, cycles, and grids, the regret bound is on the order of $\log T/\Delta$, similar to non-cooperative scenarios, while for fully connected graphs, it is $\log(T)/(\Delta m)$. Agarwal et al. [1] have $D\log(T)$ communication rounds and only $\log(A)$ bits per message, but their regret scales as $\sqrt{(A/m + deg(\mathcal{G}))D^3 T}$, where $deg(\mathcal{G})$ is the maximum degree of the graph. Note that their regret bound is at least $\sqrt{T}$.

Other related problems, such as directed communication, cooperation in Markov-Decision-Processes (MDPs) and best-arm identification, have also been studied. For a more comprehensive discussion, we refer the reader to Appendix A.

## 2 Model and problem formulation

**Stochastic MAB (SMAB):** A stochastic Multi-armed bandit problem has $A$ actions, denoted by $\mathbb{A} = \{1, \dots, A\}$. Each action $a \in \mathbb{A}$ has a reward distribution $\mathcal{D}_a$, whose support is $[0, 1]$, and its expectation is $\mu_a = \mathbb{E}_{r \sim \mathcal{D}_a}[r]$. An optimal action is denoted with $a^\star$, where $a^\star \in \arg\max_{a \in \mathbb{A}} \mu_a$, and $\mu^\star = \mu_{a^\star}$. The gap of a sub-optimal action $a$ is $\Delta_a = \mu^\star - \mu_a$.

**Multi-agent MAB:** We have an undirected connected graph $\mathcal{G}(V, E)$, where $V$ is the set of vertices and $E$ the set of edges. Every vertex represents an agent. An agent $u$ is a neighbor of agent $v$ iff $(v, u) \in E$. The diameter of the graph is denoted by $D$. Let $N^v_{\leq d}$ be the set of agents at a distance at most $d$ from agent $v$, i.e., $N^v_{\leq d} := \{u \in V | d_{\mathcal{G}}(v, u) \leq d\}$, where $d_{\mathcal{G}}(v, u)$ is the minimal path length (number of edges) from $v$ to $u$ in $\mathcal{G}$. For simplicity, we assume $m, A \leq \text{poly}(T)$.

There are $T$ rounds of play. Each agent $v \in V$, in each round of play $t \in [T]$ does the following: (1) selects an action $a^v_t \in \mathbb{A}$ and observes a reward $r^v_t \sim \mathcal{D}_{a^v_t}$ (note that the rewards of the different agents are different random variables). (2) sends messages to neighboring agents $u \in N^v_{\leq 1}$. (3) receives messages from neighboring agents $u \in N^v_{\leq 1}$. See the protocol in Algorithm 1.

**Regret definition:** The individual (pseudo) regret of an agent $v$ is defined by $\mathfrak{R}^v_T = \mathbb{E}[\sum_{t=1}^{T}(\mu^* - r^v_t(a^v_t))]$.[1] In this paper, we focus on minimizing the individual (pseudo) regret of every agent.

**Events and Messages:** Our algorithms will have the agents broadcasting about the progress they make. There will be two types of progress. The first is a new observation of a reward, which will be a *reward event*. The second is a decision to eliminate a certain action. This will be an *elimination event*. Formally, an event is a tuple describing reward, or a tuple describing an elimination of an action. A reward event is $(\texttt{rwd}, t, v, a, r)$, where $t$ is the timestep, $v$ is the agent's ID, $a = a^v_t$ is the action, and $r = r^v_t(a^v_t)$ is the reward. An elimination event is $(\texttt{elim}, v, a)$, where $v$ is the agent and $a$ is the eliminated action. To denote individual elements within an event tuple, we use subscript notation. For example, if we have an event $event = (\texttt{rwd}, t, v, a, r)$, we denote the action $a$ using $event_a$. We define a message to be a set of events.

---

[†] These functions can be as large as $m$, see discussion in related work. $h(\mathcal{G}) = \log(m)/\sqrt{\log(1/|\lambda_2|)}$. Here $f(\mathcal{G})$ as defined from Corollary 2 and Eq. (19) in [21]; the definition is very complex, but note that it may be as large as $m$.

[1] The expectation of the pseudo regret is also over the randomness of the algorithm. We will refer to the pseudo regret as the regret for the rest of the article.

---

**Algorithm 1** Stochastic MAB on Graph. Protocol for agent $v$

---

1: **for** $t \in [T]$ **do**
2:    Agent $v$ picks an action $a_t^v \in \mathbb{A}$.
3:    Environment samples a reward, $r_t^v(a_t^v) \sim \mathcal{D}_{a_t^v}$.
4:    Agent $v$ observes reward $r_t^v(a_t^v)$.
5:    Agent $v$ sends messages $m_t^{v,u}$ to each neighbor $u$.
6:    Agent $v$ receives messages $m_t^{u,v}$ from each neighbor $u$.
7: **end for**

---

## 3 Warm-up: diameter dependency

In this section, we refine the best-known regret bound that depends on the diameter. Specifically, we reduce the additive term from $DA$ to $D\log(A) + A$. Our analysis builds on recent techniques developed for delayed MAB settings, leading to a relatively simple proof (see Appendix D). This section serves as a warm-up, introducing a straightforward algorithm that still relies on the graph's diameter, and provides a useful reference point for the subsequent sections, where this dependency is removed.

Our algorithm in this section, `Sus-Act` (Algorithm 2), builds on SE, but utilizes only samples that are already observed by all agents. Specifically, at any timestep $t$, each agent has already observed the samples collected by every other agent up to time $t - D$, where $D$ is the diameter of the graph. Thus, at time $t$, `Sus-Act` computes its LCB and UCB (Lower/Upper Confidence Bounds) based on all samples from actions taken up to time $t - D$. Consequently, all agents have exactly the same LCB and UCB, leading them to select identical actions and experience the same individual regret. Notice that the actions' tie-breaking is the same across all agents.

Conceptually, the shared information between agents increases the number of observed samples per played action by a factor of $m$, the number of agents. On the other hand, samples from the last $D$ steps are not processed (and will be processed when their delay would be exactly $D$). This is equivalent to an environment with $D$-steps delayed feedback, typically introducing an additive $D$ term to the regret.

**Theorem 1.** *When each agent plays algorithm 2, the individual regret of each agent is,*

$$\mathfrak{R}_T^v = O\left( \sum_{\Delta_i > 0} \frac{\log(T)}{m\Delta_i} + D\log A + A \right).$$

The proof of Theorem 1 is deferred to the supplementary material. In summary, when the regret bound depends on the diameter, the analysis remains relatively straightforward. In the following sections, we move beyond this setting and show how to remove the diameter dependence altogether, while retaining comparable guarantees.

---

**Algorithm 2** `Sus-Act`: Successive Elimination with Suspended Act (see Algorithm 7 for detailed version)

---

1: **Input:** Diameter $D$
2: **for** $t = 1, 2, \ldots, T$ **do**
3:    The agent plays Successive Elimination using all information available up to time $t - D$.
4:    Send and receive rewards: The agent sends her reward from the current round and forwards to all neighbors any previously unsent rewards (message passing).
5: **end for**

---

## 4 The `Coop-SE` algorithm and individual regret guarantees

We study `Coop-SE`, Algorithm 3, which is a particular variant of Cooperative Successive Elimination rather than a new algorithmic concept. `Coop-SE` is fully decentralized, and each agent plays it independently. In `Coop-SE`, each agent runs SE with all the information available to it, while exchanging messages with neighbors that contain both locally generated and relayed information, including observed rewards and elimination signals (i.e., message passing).

---

**Algorithm 3** Cooperative Successive Elimination (`Coop-SE`) - simplified version (see Algorithm 6 for full pseudo-code)

---

1: **Init:** Set the active actions set to be all actions $\mathcal{A} = \mathbb{A}$.
2: **for** $t = 1, ..., T$ **do**
3:     Eliminate actions from incoming elimination-messages
4:     Calculate counts and empirical mean for each active action based on all seen messages
5:     Calculate $UCB_t$ and $LCB_t$ based on the above counts and means (see Definition 12)
6:     $E = \{a \in \mathcal{A} \mid \exists a' \in \mathcal{A} \text{ such that } UCB_t(a) < LCB_t(a')\}; \mathcal{A} = \mathcal{A} \setminus E$
7:     Choose action in round-robin from the active action, $a_t \in \mathcal{A}$. Play it and get a reward $r_t(a_t)$
8:     Send eliminations $E$, reward $(a_t, r_t)$, and all messages received at $t-1$ (message passing)
9:     Receive messages from the neighboring agents
10: **end for**

---

This increased information significantly reduces the regret compared to the non-cooperative setting. The formal description of the algorithm is provided in Algorithm 6. Our main result is the following theorem.

**Theorem 2.** *When each agent plays* `Coop-SE` *(Algorithm 3) the regret of each agent $v \in V$ is,*

$$\mathfrak{R}_T^v = O\bigg( \sum_{\Delta_i > 0} \frac{\log(T)}{m\Delta_i} + A^2 + A\sqrt{\log(T)} \bigg).$$

To the best of our knowledge, this is the only individual regret bound that is independent of the graph diameter. For comparison with prior work, we also derive an improved variant of the bound that includes the diameter: $\mathfrak{R}_T^v = O\bigg( \bigg( \sum_{\Delta_i > 0} \frac{\log(T)}{m\Delta_i} \bigg) + A \cdot \min\{A + \sqrt{\log(T)}, D\} \bigg)$. The diameter term arises from a refined analysis but is not required by the algorithm; agents do not need to know $D$. The full proof is provided in the appendix.

In Section 6, we present a lower bound of $\Omega(\sqrt{TA/m} + \sqrt{A})$, which almost matches the upper bound of the individual minimax regret of `Coop-SE`. We note that a slight variant of the algorithm, which samples actions uniformly from the set of active arms rather than using round-robin selection, incurs an additive term of $A \log T$ instead of $A^2 + A\sqrt{\log T}$ (see $Theorem$ 8). This is tighter whenever $A \gg \log T$—see Appendix F for more details. However, the precise dependence on $A$ in this additive term remains an open question.

An important insight that follows from these theorems is that for a sufficiently large number of agents, e.g., when $m = \mathcal{R}$, we achieve an individual regret bound of $O(A^2 + A\sqrt{\log T})$ that does not depends on the sub-optimality gaps.

In the following section, we present the key ideas employed in the analysis of the individual regret.

## 5 Individual regret analysis

In this section, we provide a proof sketch that outlines the key steps in our analysis. We analyze the regret of an arbitrary agent $v$, and all the definitions are referenced to this agent unless explicitly stated otherwise.

Our proof heavily relies on a notion we call *stages*. These are the time intervals between the eliminations of agent $v$. Formally, a stage $j \in [A]$ is the interval $[t_j, t_{j+1})$ where $t_1 = 1$, and $t_{j+1}$ is the timestep of the $j$'th elimination. We'll also denote by $\tau_j$ the length of the $j$'th stage and the number of active actions in that stage by $A_j := A - j + 1$.

The stages is one of our core ideas, and they allow us to do the following. We bound the agent's regret in terms of stage length, and we bound the stage length as a function of the number of samples. Finally, we bound the number of samples in the standard approach. By combining these results, we obtain our main theorem.

**Bounding the regret in term of stage length**     We start by bounding agent $v$'s regret in terms of stage lengths. Fix a sub-optimal action $a$ and assume that $i$ is the last stage in which $a$ was active.

Since $v$ chooses active actions in round-robin, in each stage $j \leq i$, she samples $a$ approximately $\tau_j / A_j$ times. Thus, the total number of times $v$ plays $a$ is approximately $\sum_{j=1}^{i} \frac{\tau_j}{A_j}$ and we can roughly bound the regret with,

$$\mathfrak{R}_T^v \lesssim \sum_{i=1}^{A} \sum_{j=1}^{i} \frac{\tau_j}{A_j} \Delta_i, \tag{1}$$

where we slightly abuse notation and let $\Delta_i$ be the sub-optimality gap of the action that was eliminated at the end of stage $i$.

**Number of samples in terms of stage length**  Consider the $j$'th stage and an action $a$ which is still active in that stage. For the sake of intuition, assume that the agents are completely synchronized, i.e., have the same set of active actions. In the first quarter of the stage, each agent who is close to $v$ contributes to $v$'s information approximately $\tau_j / (4A_j)$ samples of action $a$. Since there were $\tau_j / 4$ timesteps and each agent chooses action out of $A_j$. Moreover, these samples are observed by $v$ with a delay of at most $\tau_j / 4$ and thus will reach $v$ before the end of the stage.

If action $a$ is active in the first $i$ stages, we would expect that the number of samples that reaches $v$ for that action $a$ from the first $i$ stages is at least of order of $\sum_{j=1}^{i} \frac{\tau_j}{A_j} \cdot |N_{\leq \tau_j / 4}|$, where $N_{\leq \tau_j / 4}$ is $v$'s neighborhood of radius $\tau_j / 4$.

The above result implies that the amount of observed feedback from each stage is boosted by a factor $|N_{\leq \tau_j / 4}|$ compared to the number of times that $v$ itself chooses the action.

However, in general, the agent's policies are *not* completely synchronized, and thus, we need a stronger argument to rigorously establish the above claim. In the next subsection, we show that under `Coop-SE`, the agents implicitly synchronize with each other.

**Implicit synchronization of neighborhoods over intervals**  We now outline another core idea of our work: how agents implicitly synchronize under our algorithm, a key component for proving individual regret.

**Lemma 1.** *Consider an agent $v$. Let $j$ be a stage index such that $\tau_j^v > 16$. Then every agent $u \in N_{\leq \tau_j^v / 4}^v$ plays the same policy (i.e., has the same set of active actions) at time interval $[t_j^v + \lceil \tau_j^v / 4 \rceil, t_j^v + \lfloor \tau_j^v / 2 \rfloor]$.*

*Proof sketch.* Let us denote the active set of actions of $v$ at stage $j$ with $\mathcal{A}_j^v$. Since $d_{\mathcal{G}}(u, v) \leq \tau_j^v / 4$, agent $u$ receives all eliminations of $\mathbb{A} \setminus \mathcal{A}_j^v$ from $v$ no later than $t_j^v + \tau_j^v / 4$. Hence, after this timestep $u$'s active actions in $[t_j^v + \tau_j^v / 4, t_j^v + \tau_j^v / 2]$ must be contained in $\mathcal{A}_j^v$. For the reverse direction, let $a \in \mathcal{A}_j^v$. Assume by contradiction that $u$ encounters an elimination of $a$ before $t_j^v + \tau_j^v / 2$. Since $d_{\mathcal{G}}(u, v) \leq \tau_j^v / 4$, this elimination reaches $v$ within no more than $\tau_j^v / 4$ additional steps. Therefore, $v$ gets the elimination at $t_j^v + 3/4 \tau_j^v < t_{j+1}^v$, contradicting the stage definition which requires the stage to end precisely when an active action is eliminated. Thus, all agents in $N_{\leq \tau_j^v / 4}^v$ maintain exactly $\mathcal{A}_j^v$ as active actions throughout $[t_j^v + \tau_j^v / 4, t_j^v + \tau_j^v / 2]$. See Appendix E.3 for the detailed proof.  □

**Combining the results**  With the above result we get that in each round $j$ that the action $a$ was active, the agent $v$ gets at least $\tau_j / A_j |N_{\leq \tau_j / 4}|$ samples. Hence, the number of samples of an action $a$ that was active in the end of stage $i$ can be bounded from below. Let us denote the last round in stage $i$ with $t_i' := t_{i+1} - 1$. We get $n_{t_i'}(a) \gtrsim \sum_{j=1}^{i} \tau_j / A_j |N_{\leq \tau_j / 4}|$. Using standard concentration bounds, we show that the number of samples $v$ can see from a sub-optimal action $a$, without eliminating it, is approximately $1/\Delta_a^2$. Hence, $n_{t_i'}(a) \lesssim 1/\Delta_i^2$

On the other hand, we can bound $N_{\leq \tau_j / 4}$ with the stages and the number of agents $m$: $N_{\leq \tau_j / 4} \geq \min\{m, \frac{\tau_j}{4}\}$.

Using the lower bound on the number of samples we get,

$$\frac{1}{\Delta_i^2} \geq n_{t_i'}(a) \geq \sum_{j=1}^{i} |N_{\leq \tau_j / 4}| \frac{\tau_j}{16 A_j} \geq \sum_{j=1}^{i} \min\{m, \frac{\tau_j}{4}\} \frac{\tau_j}{16 A_j}.$$

We split the analysis into stages where $m < \tau_j/4$ and stages where $m \geq \tau_j/4$. The first case is simpler, while the second is bounded using Cauchy–Schwarz. These stages are used only for analysis, we ultimately bound $\sum_{j=1}^{i} \frac{\tau_j}{A_j} \Delta_i$ using only $m$, $\Delta_i$, $A$, and $T$. Full details are deferred to the appendix (see Appendix E.3 for this part).

## 6 Lower bound

In this section, we present a lower bound, demonstrating that our algorithm achieves near-optimal individual regret.

The problem we study in this paper is obtaining an upper bound on individual regret which is independent of the graph's diameter or other graph properties. This means the bound should hold for *any* communication graph, and since we focus on individual regret, it must hold for *every* agent. Consequently, when proving a lower bound, we can consider any graph, including the worst-case one, and we only need to identify at least one agent that incurs this level of regret.

Note that the lower bound is stated in the minimax form, where the sub-optimality gaps are on the order of $\sqrt{1/T}$. We believe this formulation captures the essence of the problem more clearly, though it can be equivalently expressed in a problem-specific form. For consistency, we also provide the minimax forms of the regret in Section 1.2 and in the appendix.

**Theorem 3.** *For every algorithm, and for every $T, A, m$, there exists a problem instance of the cooperative stochastic MAB over a communication graph such that there exists an agent for which the individual minimax regret is at least, $\Omega(\sqrt{AT/m} + \sqrt{A})$.*

Note that the statement specifies *"there exists an agent"*, and cannot be improved to *"for every agent"*. This is because, with at least $A$ agents, it is always possible for one agent to have zero regret by assigning each agent to select a distinct action for the entire horizon.

The primary implication of the lower bound is that even if $m \to \infty$, the individual regret still scales with the number of actions. The main gap from our upper bound is the exact dependency in $A$ in the additive term as well as the logarithmic dependency; these gaps still remain open questions.

The lower bound combines two separate lower bounds, $\Omega(\sqrt{AT/m})$ and $\Omega(\sqrt{A})$. The $\Omega(\sqrt{AT/m})$ bound holds even in a fully connected network and follows directly from the lower bound established by Ito et al. [15]. We also remark that an instance-dependent variation of this lower bound, specifically $\Omega((1 - \mu^\star)\mu^\star \sum_{\Delta_i > 0} \log(T)/(m\Delta_i))$, where $\mu^\star$ is the expectation of the optimal action, can be obtained using the same technique.

Recall that our upper bound holds for any graph and does not depend on the diameter. Thus, in order to show that it cannot be improved in general, it is sufficient to show a lower bound for a specific graph. Hence, to obtain the $\Omega(\sqrt{A})$ we focus on a line graph. The intuition of the proof is the following. We consider a deterministic MAB where one action has reward 1 and all other actions have reward 0. An agent, during the first $\tau$ timesteps receives $\Theta(\tau^2)$ observations. Therefore, if $\tau \lesssim \sqrt{A}/10$, then an agent receives information about at most $A/100$ of the actions. We construct a probability function where each optimal action has equal probability of being selected. Under this distribution, with probability 0.99, the agent fails to observe the optimal action, resulting in an individual regret of at least $0.99 \cdot \sqrt{A}/10$. Therefore, there must exist at least one specific problem instance that induces this regret. For the formal proof see Theorem 10 in the appendix. Note that the technique employed for this lower bound can extend to other graph structures; for instance, a grid graph can yield a lower bound of $A^{1/4}$.

## 7 Communication results

In practical distributed systems, communication constraints can significantly impact the performance of cooperative learning algorithms. We examined two restricted communication settings: The first limits messages to $O(\log(Am))$ bits, which corresponds to the well-known CONGEST model in distributed systems (see Peleg [31]). The second allows agents to send messages in only $O(\log(T))$ timesteps throughout the entire horizon, a constraint sometimes referred to as communication cost in the literature. Our results show that effective cooperative learning remains possible even under these

constraints, with agents maintaining strong individual regret guarantees that do not depend on the diameter.

## 7.1 The CONGEST model

To establish our communication-efficient results, we begin by showing that our base algorithm `Coop-SE`, when operating on a spanning tree, can function effectively with reduced message size of $O(A \log(Am))$. This initial compression serves as a stepping stone toward our full CONGEST model analysis, where we further reduce communication by having agents share information about only single actions at a time.

The key insight is that we can aggregate information about each action without losing accuracy. Instead of transmitting individual reward observations and elimination events, we can simply maintain running sums of rewards and a single elimination flag per action. This compression requires only $O(A \log(Am))$ bits per message, representing each action's new information: observation count, cumulative reward, and elimination status that reached to the agent in the previous round or produced by the agent in the current round. However, this aggregation is only valid when agents don't receive duplicate information. We achieve this by restricting communication to a spanning tree of the network, where messages are forwarded along the tree nodes. This simple modification eliminates redundant transmissions while preserving `Coop-SE`'s regret guarantees, as our bounds are independent of the graph structure. Algorithm `Coop-SE-Restricted` (Algorithm 9 in the appendix) implements these ideas.

For completeness, we note that our analysis assumes all agents share the same spanning tree, which can be computed in a preprocessing phase prior to the execution of the algorithm. Otherwise, we can use one of the many distributed algorithms for this purpose (see [31]).

Moving to the full CONGEST, let us now consider the case that the size of the messages is limited to $O(\log(Am))$ bits — this is done using our algorithm `Coop-SE-CONGEST` (Algorithm 11 in the appendix). Here is the high-level idea of the algorithm. As before, we construct a tree from the original graph, and aggregate messages and avoid duplicates. But rather than sending $A$ messages each of size $O(\log(mA))$, the agent sends in each round only one message regarding only one action. The action for which the agent sends the information is chosen in a round robin, without considering if the action is active or not. The key idea is that the round robin scheduling starts at a different action for each agent. This mechanism ensures that when a message travels from any node $v$ to the root node $w$, apart from the distance between nodes it will not encounter any delay after it has been sent from its originating node. Similarly, messages from $v$ outward from the root will not encounter delay.

Let us denote the spanning tree with $\mathcal{T}$, and the distance on the tree from $v$ to $w$ with $d_{\mathcal{T}}(v, w)$. The mechanism works as follows: agent $v$, located at distance $d := d_{\mathcal{T}}(v, w)$ from the root $w$, sends messages for action $a$ to its parent, i.e., toward the root, at timesteps $t$ such that $a \equiv t + d \pmod{A}$, and to its children whenever $a \equiv t - d \pmod{A}$.

Here is an example of how the idea works. For the up-stream, assume that at timestep $t$ message about action $a$ was sent from $v$ to its parent $\hat{v}$. The message reaches $\hat{v}$, which is at a distance $d - 1$ from the root, at timestep $t + 1$. At timestep $t + 1$ the agent $\hat{v}$ sends this message to $\hat{v}$'s parent since $t + 1 + (d - 1) = t + d \equiv a \pmod{A}$, and the message continues up-stream with no delay. Similarly for the down-stream. Additionally, a message that travels from $v$ to $u$ might wait at most $A$ timesteps at their common ancestor until the round-robin reaches this action. We get the following theorem.

**Theorem 4.** *When all the agents play `Coop-SE-CONGEST`(Algorithm 11 in the appendix) the individual regret of each agent $v \in V$ is bounded by,*

$$\mathfrak{R}_T^v = O\left(\sum_{i \in [A], \Delta_i > 0} \frac{\log(mTA)}{m\Delta_i} + A^2 + A\sqrt{\log(mTA)}\right).$$

## 7.2 Small number of messages

In this section, we present `Coop-SE-Comm-Cost`, a variant of Successive Elimination that requires only $O(\log(T))$ communication rounds per agent. The algorithm operates in phases and communicates along a spanning tree, clustering agents into groups of size at least $\min\{2^i, m\}$ in phase $i$. Within each cluster, the maximum distance between the cluster root and any descendant is at most

$2^{i+1}$. The existence and a computation of such clustering is shown in Lemma 31 and Algorithm 4 in the appendix.

Each phase consists of three $2^{i+1}$-length steps: first, agents send information upward to their cluster root; next, the root determines the active set of actions and broadcasts it downward; finally, agents synchronously sample active actions. During the sampling step, no communication occurs. This structure allows each cluster to collect $\Omega(2^i \cdot \min\{2^i, m\})$ samples while sending messages in only $O(1)$ timesteps per phase.

The analysis's complexity arises because agents determine their current phase's active action set using information from the previous phase rather than the current one. Let $A_i$ be the number of active actions in phase $i$ (for agent $v$). Unlike single-agent phased algorithms, we do not require agents to sample each costly action for $2^i$ steps, since this would underuse cooperation. In our phasing algorithm, the phase lasts $2^i$ steps, and each action is sampled $2^i/A_i$ times and the regret scales as $(2^i/A_i)\Delta_i$. To illustrate the problem, notice that the number of samples used for the elimination at the start of phase $i$ is inversely related to $A_{i-1}$, not to $A_i$, in contrast to the regret. If $A_{i-1} \gg A_i$, it might be that easy-to-eliminate actions were eliminated at the beginning of phase $i$, leaving only the costly ones in phase $i$.

We addressed this challenge through amortized analysis: phases with a low ratio of previous-to-current active actions ($A_{i-1}/A_i \leq 2$) effectively subsidize phases with a high ratio. The complete analysis can be found in the appendix, specifically in Lemma 41. Formally, we get,

**Theorem 5.** *When all agents play* `Coop-SE-Comm-Cost` *(Algorithm 13 in the appendix) the individual regret of each agent is,*

$$\mathfrak{R}_T^v = O\left( \sum_{a \in \mathcal{A}} \frac{\log(mTA)\log(A)}{\Delta_a \cdot m} + A\log(mTA)\log(A) \right).$$

## 8 Future Work

Our work leaves several interesting directions for future works. First, our algorithms can either handle logarithmic message size or logarithmic number of communication rounds. An interesting future work would be to achieve our near-optimal regret bounds while simultaneously maintaining both logarithmic message size and logarithmic number of communication rounds. Second, extending our results to other MAB algorithms, specifically, Upper Confidence Bound (UCB) and Thompson Sampling, would be very interesting. The technical challenge is that in UCB (or Thompson sampling) there are actions which are selected very rarely. Such actions can cause different agents to behave differently. Our methodology builds on having the different agents behave similarly (as is shown through the implicit or explicit synchronization). One can implement explicit synchronization at the cost of the diameter, which will result in a significantly inferior regret.

Another interesting direction for future work is to more explicitly leverage the graph structure to improve cooperation efficiency. In particular, characterizing how the regret depends on the topology of the communication graph may enable reducing the additive term in our bound.

Finally, a natural open question is closing the gap in the additive term between our upper and lower regret bounds.

## Acknowledgments

This project is supported by the European Research Council (ERC) under the European Union's Horizon 2020 research and innovation program (grant agreement No. 882396), by the Israel Science Foundation and the Yandex Initiative for Machine Learning at Tel Aviv University and by a grant from the Tel Aviv University Center for AI and Data Science (TAD).

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

# A  Other Related Work

**Cooperative MAB with heavy tail distributions** was in [11]. They show individual regret bound that scales inversely with the number of neighbors, as opposed to the total number of agents, as in our regret bounds. **Asynchronous model** was considered by [32, 7]. In this model, agents do not have a shared global system clock, and thus, it is a harder setting than that considered in this work. Consequently, the regret bounds they achieve are significantly weaker, scaling as $\sum_{i=1}^{\lceil A/m \rceil + 2} \log(T)/\Delta_i$ where $\Delta_1 \leq \Delta_2 \leq \ldots \leq \Delta_{A-1}$. **Directed communication** graphs and random graphs was studied as well. In [43, 41, 42] they considered directed communication graphs. Their instance-dependent regret has an additive term that is linear in the number of agents. In [28] they considered a setting where the communication graph is stochastic, such that messages have random delays and adversarial corruptions. Their regret has a multiplicative factor that can be as large as the clique cover number. **Best action identification** using cooperation was studied in [14, 34] where the network is fully connected and they also minimize the number of messages. **Heterogeneous agents** which observe their neighbors with some probability and minimize the group regret were also addressed by [24, 27]. In [24] they derived a group regret based on various properties of the graph and in [27] they studied group regret in multi-star networks. The case of each agent having a subset of actions that are relevant to them was studied in [38], and the group regret bound was derived. **Linear contextual MAB** with a network of users of similar linear utility was analyzed in [4]. **Cooperation in Markov-Decision-Processes (MDPs)** has been studied in [23], who have shown group regret guarantees in cooperative stochastic MDPs over a general network. In [17] they considered both the stochastic and non-stochastic cases in cooperative MDPs but only over a fully connected graph.

# B  Summery of Notations

For convenience, the table below summarizes most of the notation used throughout the paper.

| | |
|---|---|
| $\mathcal{D}_a$ | The reward distribution of action $a$ |
| $\mu_a$ | The expected reward of action $a$ |
| $\mu^\star$ | The maximal expected reward |
| $a^\star$ | An optimal action |
| $\Delta_a$ | The sub-optimality gap of action $a$ |
| $N^u_{\leq d}$ | The set of agents at distance at most $d$ from agent $u$ |
| $N_{\leq d}$ | For ease of notation $N_{\leq d} := N^v_{\leq d}$; see Remark 2 |
| $d_\mathcal{G}(v, u)$ | The minimal path length (number of edges) from $v$ to $u$ |
| $t_j$ | The beginning of stage $j$ of agent $v$; see Remark 2 |
| $\tau_j$ | The length of stage $j$ of agent $v$; see Remark 2 |
| $A_j$ | The number of active actions in the $j$'th step of agent $v$; see Remark 2 |
| $\iota$ | $\log(3mTA)$ |
| $n^u_t(a)$ | The number of samples that $u$ observed by the beginning of time $t$ |
| $n_t(a)$ | For ease of notation $n_t(a) := n^v_t(a)$; see Remark 2 |
| $b^u_t(a)$ | The number of times agent $u$ played action $a$ until the beginning of round $t$. |
| $b_t(a)$ | For ease of notation $b_t(a) := b^v_t(a)$; see Remark 2 |
| $p^u_k$ | The policy of agent $u$ at time $k$ |
| $A_\Delta$ | The set of elimination indices (with respect to agent $v$) with gaps larger than $\sqrt{\frac{A\iota}{Tm}}$ |
| $a_i$ | The $i$'th action being eliminaed by agent $v$ |
| $\Delta_i$ | The sub-optimality gap of $a_i$ |
| $G_\tau$ | The set of "Good Intervals": $\{j | \tau_j > 16\}$. |
| $S_\tau$ | The set of "Short Intervals": $\{j | j \in G_\tau \ \& \ \tau_j/4 < m\}$ |
| $a \vee b$ | The maximum between the elements. $a \vee b := \max\{a, b\}$ |

## C Instance independent Bounds

The instance independent bounds follow immediately from the instance dependent bounds. It works generally as follows. The analysis divides the gaps into two. The first group of gaps are the small gaps, where $\Delta_i \leq \sqrt{\frac{\log(T)A}{Tm}}$. This group contributes no more than $T \cdot \sqrt{\frac{\log(T)A}{Tm}} = \sqrt{\frac{\log(T)AT}{m}}$ to the regret. The gaps from second group appear as inverse in the bounds, and we get $\log(T)/(m\Delta_i) \leq \sqrt{\frac{\log(T)T}{Am}}$. Summing over all the actions we get $\sqrt{\frac{\log(T)AT}{m}}$.

## D Omitted Proof from Section 3

**Remark 1.** *The problem independent bound for* Sus-Act *is*

$$\mathfrak{R}_T^v = O\left( \log(mTA)\sqrt{\frac{AT}{m}} + D\log A + A \right).$$

*Proof of Theorem 1.* Without loss of generality, we assume that $D \geq 1$. Let us define the good event as the event in which in every timestep the mean of the action is in the confidence interval. It is described in detailed in the appendix (see Definition 14). From Lemma 2, the complementary event occurs only with probability of at most $1/T^2$, and thus, adds no more than 1 to the regret. For the rest of the proof will assume that the good event holds. Therefore, for all $a$,

$$UCB_t(a^\star) \geq \mu^\star \geq \mu_a \geq LCB_t(a).$$

Where $UCB_t(a)$ is the upper confidence bound that was calculated at the timestep $t$, and similarly $LCB_t(a)$. Therefore $a^\star$ is never eliminated.

Let $n_t(a)$ be the number of suspended counts until the beginning of timestep $t$, i.e., $n_t(a) = \sum_{\tau=1}^{t-D} \sum_v \mathbb{I}\{a_\tau^v = a\}$. Denote by $B_t(a)$ the total number of times that $a$ was played by all agents until the beginning of round $t$. I.e., $B_t(a) = n_{t+D-1}(a) = \sum_{\tau=1}^{t-1} \sum_{v \in V} \mathbb{I}\{a_\tau^v = a\}$. Let $t_a$ be the last elimination step which $a$ was not yet eliminated. By definition, since $a$ was not eliminated,

$$LCB_{t_a}(a^\star) \leq UCB_{t_a}(a).$$

Under the good event,

$$LCB_{t_a}(a^\star) = \hat{\mu}_{t_a}(a^\star) - \sqrt{\frac{2\log(3mTA)}{n_{t_a}(a^\star) \vee 1}} \geq \mu(a^\star) - 2\sqrt{\frac{2\log(3mTA)}{n_{t_a}(a^\star) \vee 1}}$$

$$UCB_{t_a}(a) = \hat{\mu}_{t_a}(a) + \sqrt{\frac{2\log(3mTA)}{n_{t_a}(a) \vee 1}} \leq \mu(a) + 2\sqrt{\frac{2\log(3mTA)}{n_{t_a}(a) \vee 1}},$$

where $x \vee y := \max\{x, y\}$. Combining with the last display we get,

$$\Delta_a \leq 2\sqrt{\frac{\log(3mTA)}{n_{t_a}(a) \vee 1}} + 2\sqrt{\frac{\log(3mTA)}{n_{t_a}(a^\star) \vee 1}} \leq \sqrt{\frac{16(\log(3mTA))}{(n_{t_a}(a) - m) \vee 1}}$$

$$\implies n_{t_a}(a) \leq \frac{16\log(3mTA)}{\Delta_a^2} + m$$

where we've used the fact that active actions are played at the same rate, and thus the number of suspended counts of two active actions differs by at most $m$. Sicne $t_a$ is the last elimination step in which $a$ was not eliminated, $a$ was played is no more than $B_{t_a}(a) + m$ times (in $t_a$ the agents still didn't eliminate $a$). Thus, the total sum of regret form action $a$ is bounded by,

$$(m + B_{t_a}(a))\Delta_a = m \cdot \Delta_a + n_{t_a}(a)\Delta_a + (B_{t_a}(a) - n_{t_a}(a))\Delta_a$$

$$\leq 2m \cdot \Delta_a + \frac{16\log(3mTA)}{\Delta_a} + (B_{t_a}(a) - n_{t_a}(a))\Delta_a.$$

Denote by $\sigma(a)$ the number of active actions at time $t_a$. Notice that every agents waits at least $D$ timetsteps before the first elimination, hence $t_a \geq D$. We can bound the last term above by,

$$B_{t_a}(a) - n_{t_a}(a) = \sum_{\tau=t_a-D+1}^{t_a-1} \sum_v \mathbb{I}\{a_t^v = a\} \tag{2}$$

$$= \sum_v \sum_{\tau=t_a-D+1}^{t_a-1} \mathbb{I}\{a_t^v = a\}$$

$$\leq \sum_v \left(\frac{D}{\sigma(a)} + 1\right) = \frac{mD}{\sigma(a)} + m. \tag{3}$$

where the inequality holds since there are at most $D-1$ timesteps, and the agnet chooses the actions in round robin. By combining (2) and (3), summing over the actions, and noting that $\sum_{a\neq a^\star} \frac{1}{\sigma(a)} \leq \log A + 1$ (regardless of the elimination order, see Lemma 7), we get that the total regret is bounded by,

$$\sum_{a\neq a^\star} \left(2m \cdot \Delta_a + \frac{16\log(3mTA)}{\Delta_a}\right) + mD(\log A + 1) + mA.$$

Finally, since all agents play the exact same actions we get the the individual regret of each agent is bounded by,

$$\mathfrak{R}_T \leq \sum_{a\neq a^\star} \left(2\Delta_a + \frac{16\log(3mTA)}{m\Delta_a}\right) + D(\log A + 1) + A$$

$$\leq \sum_{a\neq a^\star} \frac{16\log(3mTA)}{m\Delta_a} + D(\log A + 1) + 3A,$$

where the last inequality is since $\sum_a \Delta_a \leq A$. This finishes the proof for the gaps dependent bound. Now we will reach the problem-independent bound. All actions with small gaps, $\{a \in A | \Delta_a \leq \sqrt{\frac{A}{Tm}}\}$, contribute no more than $\sqrt{\frac{AT}{m}}$ to the regret. There are at most $T$ round in which the agent chooses actions with small gaps, so their contribution is bounded by $T\sqrt{\frac{A}{Tm}} = \sqrt{\frac{AT}{m}}$. For large gaps, i.e., $\Delta_a > \sqrt{\frac{A}{Tm}}$ we get

$$\sum_{a\neq a^\star, \Delta_a > \sqrt{\frac{A}{Tm}}} \frac{16\log(3mTA)}{m}\sqrt{\frac{Tm}{A}} \leq 16\log(3mTA)\sqrt{\frac{AT}{m}}$$

Putting it all together with the small gaps and with the good event, we get

$$\mathfrak{R}_T \leq 17(\log(3mTA))\sqrt{\frac{AT}{m}} + D(\log A + 1) + 3A + 1$$

$$\square$$

# E    Proof of the Main Theorem

**Remark 2.** *For the ease of notation, the following proof and definitions focus on a specific agent, named $v$.*

## E.1    Definitions

**Definition 1.** *A stage is a timestep-interval when its boundaries are the eliminations. The stage's index is usually denoted by $j$. The time interval is split into $A$ different stages. Assume that the elimination timesteps are $s_1, s_2, \ldots$. The first stage starts at $t = 1$ and ends with the first elimination. I.e., it is the timesteps that are in time interval $[1, s_1)$. The second stage is $[s_1, s_2)$, etc. Denote $t_j$ to be the timestep in which the agent started the $j$'th stage, where $t_1 = 1$ and $t_{A+1} = T + 1$.*

**Definition 2.** *Denote $\tau_j$ to be the length of the $j$'th stage (for agent $v$).*

**Definition 3.** *Denote $A_j := A - j + 1$ to be the number of remained actions in the $j$'th stage.*

**Definition 4.** *Elimination index $i$ of the action $a$ is the stage index in which in its end the action is eliminated. Every action has a unique elimination index.*

*If some actions are eliminated in the same timestep, then the stage is of zero length and the elimination index are chosen arbitrary. The elimination index of $a$ is denoted by $i_a$, and the appropriate action for elimination index $i$ is denoted by $a_i$.*

**Definition 5.** *Denote with $A_\Delta$ the set of elimination indices of large gaps. $A_\Delta = \{i | \Delta_{a_i} \geq \sqrt{\frac{A\iota}{Tm}}\}$.*

**Definition 6.** *For the ease of notation, denote $\Delta_i := \Delta_{a_i}$.*

**Definition 7.** *Define the set of "Good Intervals" to be the set of long enough intervals: $G_\tau = \{j | \tau_j > 16A_j\}$. These are the intervals we will focus in the proofs.*

**Definition 8.** *Denote the group of indices of short stages with $S_\tau$. Specifically,*

$$S_\tau := \{j | j \in G_\tau \ \& \ \tau_j / 4 < m\}$$

**Definition 9.** *Denote the number of samples an agent $u$ sees for action $a$ until the beginning of timestep $t$ with $n_t^u(a)$. For the ease of notation, denote $n_t(a) := n_t^v(a)$.*

**Definition 10.** *Denote by $b_t^u(a)$ the number of times agent $u$ played action $a$ until the beginning of round $t$.*

**Definition 11.** *Denote the maximum of elements with $\vee$, i.e., $a \vee b := \max\{a, b\}$*

**Definition 12.** *Denote the upper confidence bound for agent $u$ for action $a$ with $UCB_{n(a)}^u(a) = \hat\mu(a) + \sqrt{\frac{2\log(3mTA)}{n(a) \vee 1}}$, where $n(a)$ is the number of times agent $u$ observed action $a$, and $\hat\mu(a)$ is the empirical mean calculated by $u$ for action $a$. Similarly, let $LCB_{n(a)}^u(a) = \hat\mu(a) - \sqrt{\frac{2\log(3mTA)}{n(a) \vee 1}}$ denote the corresponding lower confidence bound. In other words, $UCB_{n(a)}^u(a)$ and $LCB_{n(a)}^u(a)$ are the confidence bounds calculated in Algorithm 5 Equation* (20) *when agent $u$ calls this algorithm with parameters $n$, a vector containing the number of observations for each action, and $\hat\mu$, the vector of empirical means.*

**Definition 13.** *Denote the logarithmic term used in Algorithm 5 with $\iota$, i.e., $\iota = \log(3mTA)$*

### E.2 The Good Event

The good event $G^1$ captures the intuition that the true expectation of each action is between the UCB and the LCB.

**Definition 14.** *Define the good event, $G^1$, to be the event in which for every agent $u$, for every action $a$ and for every rwd-event that was received, the empirical mean is in the confidence interval, i.e.,*

$$\mu_a \in [LCB_{n(a)}^u(a), UCB_{n(a)}^u(a),]$$

*where $n(a)$ is the number of rwd-events that were received for this action by the agent $u$.*

**Lemma 2.** *The good event $G^1$ happens with high probability. Specifically,*

$$\mathbb{P}(\neg G^1) \leq \frac{1}{3mT^2A^2} \leq \frac{1}{3T^2}$$

*Proof.* Event $\neg \mathbf{G^1}$: Denote $M_a^u(k)$ to be the $k$'th rwd event agent $u$ received for action $a$. Define $X_n^u(a) := \sum_{k=1}^n (M_a^u(k) - \mu_a)$ and $\lambda_n := \sqrt{\frac{2\iota}{n}}$. Note that $X_n^u(a)$ is a martingale. From Azuma's inequality we get

$$Pr\left(\left|\frac{X_n^u(a)}{n} - \mu_a\right| \geq \lambda_n\right) \leq \frac{1}{3m^3T^3A^3}$$

There are at most $m \cdot T$ rwd events the agent can get. The same holds for every action and for every message. The upper confidence bound ($UCB_n^u(a)$) is defined as $X_n^u(a) + \lambda_n$ and the lower confidence bound ($LCB_n^u(a)$) is defined as $X_n^u(a) - \lambda_n$. From the union bound we get that with

high probability for every agent, for every timestep, for every action and for every `rwd` event message the agent get, the actual mean of the action would be inside the confidence bound. Specifically

$$G^1 := \forall u \in V, \forall a \in A, \forall n \in [m \cdot T](\mu_a \in [LCB_n^u(a), UCB_n^u(a)])$$

$$\mathbb{P}(\neg G^1) \leq \frac{1}{3mT^2A^2} \leq \frac{1}{3T^2}$$

$\square$

### E.3   Proof of Theorem 2

**Lemma 3.** *The complementary event of the good event adds no more than 1 to the regret of each agent.*

*Proof.* From Lemma 2, the complementary event of the good event happens in probability lower than $\frac{1}{T^2}$. Every agent plays $T$ timesteps, and the gaps are bounded by 1, i.e., for every action $a$ we have $\Delta_a \leq 1$. Hence, in expectation, this adds at most $\frac{1}{T} \leq 1$ to the regret. $\square$

In the proof from now on, we assume the good event holds.

**Proof of Lemma 1.** Let $t \in [t_j^v + \lceil \tau_j^v/4 \rceil, t_j^v + \lfloor \tau_j^v/2 \rfloor]$ and let $u \in N_{\leq \tau_j^v/4}^v$. Denote the set active actions of $v$ in the $j$'th stage as $\mathcal{A}_j^v$. We will show that an action $a$ is active for $u$ at $t$ iff $a \in \mathcal{A}_j^v$.

Let $a$ be an active action of $u$ at time $t$, where $t \in [t_j + \lceil \tau_j/4 \rceil, t_j + \lfloor \tau_j/2 \rfloor]^v$. Since $u \in N_{\leq \tau_j^v/4}^v$, we have $d_{\mathcal{G}}(u, v) \leq \tau_j^v/4$. The distance $d_{\mathcal{G}}(u, v)$ is a natural number, so it is at most $\lfloor \tau_j^v/4 \rfloor$. Therefore $u$ gets all $v$'s eliminations (the first $j - 1$ eliminated actions) until the beginning of round $t_j^v + \lfloor \tau_j^v/4 \rfloor$. By the stage's definition, the agent $v$ doesn't encounter any new elimination. Therefore, along the stage, she doesn't send any new elimination event regarding her active actions. Hence, for any $t' \geq t_j^v + \lceil \tau_j^v/4 \rceil$, $u$ does not have any active action which is not in $\mathcal{A}_j^v$. Hence, $a \in \mathcal{A}_j^v$.

Let $a$ be an action in $\mathcal{A}_j^v$. We will show that $a$ is an active action of $u$ at time $t$, where $t \in [t_j^v + \lceil \tau_j^v/4 \rceil, t_j^v + \lfloor \tau_j^v/2 \rfloor]$. Assume for contradiction that $u$, at timestep $t_j^v + \lfloor \tau_j^v/2 \rfloor$ or before, encounters an elimination of $a$. The elimination event should arrive to $v$ in no more than $\lfloor \tau_j^v/4 \rfloor$ timesteps, so $v$ should get the elimination event at most at timestep $t_j^v + \lfloor \tau_j^v/2 \rfloor + \lfloor \tau_j^v/4 \rfloor \leq t_j^v + \frac{3\tau_j^v}{4}$. But $\tau_j^v > 16A_j > 16$, then $\tau_j^v/4 > 4$, so $t_j^v + \frac{3\tau_j^v}{4} < t_{j+1}^v - 4$. Therefore, the elimination event about an action in $\mathcal{A}_j^v$ should arrive to $v$ at least 5 timesteps before stage $j + 1$ begins. This is a contradiction to the definition of stage: a stage ends when an active action in this stage is eliminated, and not before. Therefore, $a$ is an active action of $u$ at $t$.

We get that for every $t \in [t_j^v + \lceil \tau_j^v/4 \rceil, t_j^v + \lfloor \tau_j^v/2 \rfloor]$ and for every $u \in N_{\leq \tau_j^v/4}^v$, the active actions of $u$ at $t$ are exactly $\mathcal{A}_j^v$. In other words, we get that in time interval $[t_j^v + \lceil \tau_j^v/4 \rceil, t_j^v + \lfloor \tau_j^v/2 \rfloor]$ all agents in $N_{\leq \tau_j^v/4}^v$ play the same policy, i.e., choosing randomly from $\mathcal{A}_j^v$. $\square$

**Lemma 4.** *For every action $a$ that was not eliminated before the end of stage $i$, we have*

$$n_{t_{i+1}-1}(a) \geq \sum_{j=1, j \in G_\tau}^i \frac{\tau_j}{8A_j} |N_{\leq \tau_j/4}|.$$

*Proof.* In each stage $j \in G_\tau$, all the samples that each agent in $N_{\leq \tau_j/4}$ produces reach agent $v$ before the end of the stage. Therefore, each agent contibutes at least $\lfloor \tau_j/(4A_j) \rfloor$ samples. Since $j \in G_\tau$, $\lfloor \tau_j/(4A_j) \rfloor \geq \tau_j/(8A_j)$. $\square$

**Lemma 5.** *For every action $a$ that was not eliminated before the end of stage $i$,*

$$\sum_{j=1, j \in G_\tau}^i \frac{\tau_j}{A_j} |N_{\leq \tau_j/4}| \leq \frac{256\iota}{\Delta_a^2}.$$

*Proof.* Fix an action $a$ that was not eliminated before the end of stage $i$. Denote $t' = t_{i+1} - 1$. The action $a$ is still active by agent $v$ at time $t'$, and thus, $UCB_{t'}^v(a) \geq LCB_{t'}^v(a^\star)$. Note the slightly abuse of notation, when $UCB_{t'}^v(a)$ is actually $UCB_{n_{t'}(a)}^v(a)$, and the same for $LCB$. Under the good event $G^1$,

$$\mu_a + 2\lambda_{t'}^v(a) \geq UCB_{t'}^v(a) \geq LCB_{t'}^v(a^\star) \geq \mu_{a^\star} - 2\lambda_{t'}^v(a^\star).$$

Rearranging it we get,

$$\Delta_a \leq 2\sqrt{\frac{2\iota}{n_{t'}(a)}} + 2\sqrt{\frac{2\iota}{n_{t'}(a^\star)}}.$$

Recall that under the good event, $a^\star$ is never eliminated. Thus, we can apply Lemma 4 on both $a$ and $a^\star$ and further bound $\Delta_a$ by,

$$\Delta_a \leq 4\sqrt{\frac{2\iota}{\sum_{j=1, j\in G_\tau}^i \frac{\tau_j}{8A_j}|N_{\leq \tau_j/4}|}},$$

then

$$\Delta_a^2 \leq 16 \frac{2\iota}{\sum_{j=1, j\in G_\tau}^i \frac{\tau_j}{8A_j}|N_{\leq \tau_j/4}|},$$

we get

$$\sum_{j=1, j\in G_\tau}^i \frac{\tau_j}{8A_j}|N_{\leq \tau_j/4}| \leq \frac{32\iota}{\Delta_a^2},$$

and,

$$\sum_{j=1, j\in G_\tau}^i \frac{\tau_j}{8A_j}|N_{\leq \tau_j/4}| \leq \frac{32\iota}{\Delta_a^2}$$

By rearranging terms we get the Lemma's statement. $\qquad\square$

**Lemma 6.** *For any $\tau \geq 0$*

$$\min\{\tau, m\} \leq |N_{\leq \tau}|$$

*Proof.* The graph is connected, so either there exists an agent $u$ at distance $\lfloor \tau \rfloor$ from $v$, in which case $N_{\leq \tau} \geq \lceil \tau \rceil \geq \tau$, or all the agents are at distance at most $\tau$ from $v$, in which case $N_{\leq \tau} = m$. $\qquad\square$

**Lemma 7.** $\sum_{j=1}^A \frac{1}{A_j} \leq \log A + 1$

*Proof.*

$$\sum_{j=1}^A \frac{1}{A_j} = \sum_{j=1}^A \frac{1}{A - j + 1}$$

$$= \sum_{i=1}^A \frac{1}{i}$$

$$= 1 + \sum_{i=2}^A \frac{1}{i}$$

$$\leq 1 + \int_1^A \frac{1}{x} dx$$

$$= 1 + \log A$$

$\qquad\square$

**Lemma 8.** *The regret of agent $v$ (under the good event) of action $a_i$ from stages $j \in G_\tau$ is bounded by*

$$\Delta_i \sum_{j=1}^{i} \frac{2\tau_j}{A_j} \tag{4}$$

*Proof.* In each round that action $a_i$ is active the agent plays it at most $\lceil \tau_j/A_j \rceil$ times. Since we count here only the regret from the "good stages", i.e., $j \in G_\tau$, $\lceil \tau_j/A_j \rceil \leq 2\tau_j/A_j$. $\qquad \square$

**Lemma 9.** *The regret for all actions from the stages $j \notin G_\tau$ is at most $16A^2$.*

*Proof.* There are at most $A$ such stages. Each stage is at most of $16A$ length. The gaps are bound by 1, and the result follows. $\qquad \square$

**Lemma 10.** *For every action elimination index $i$, it holds that*

$$\sum_{j=1, j \in S_\tau}^{i} \frac{\tau_j^2}{4A_j} + \sum_{j=1, j \in G_\tau \setminus S_\tau}^{i} \frac{\tau_j}{A_j} m \leq \frac{256\iota}{\Delta_i^2}$$

*where $S_\tau := \{j | j \in G_\tau \ \& \ \tau_j/4 < m\}$, and $\{\tau_j | j \in [A]\}$ are the stage lengths.*

*Proof.* From Lemma 5,

$$\sum_{j=1, j \in G_\tau}^{i} \frac{\tau_j}{A_j} |N_{\leq \tau_j/4}| \leq \frac{256\iota}{\Delta_{a_i}^2}.$$

On the other hand, using Lemma 6

$$\sum_{j=1, j \in G_\tau}^{i} \frac{\tau_j}{A_j} |N_{\leq \tau_j/4}| \geq \sum_{j=1, j \in G_\tau}^{i} \frac{\tau_j}{A_j} \min\{m, \tau_j/4\}$$

$$= \sum_{j=1, j \in S_\tau}^{i} \frac{\tau_j^2}{4A_j} + \sum_{j=1, j \in G_\tau \setminus S_\tau}^{i} \frac{\tau_j}{A_j} m$$

$\qquad \square$

**Lemma 11.** *The regret for action $a_i$ from stages $j \in G_\tau$ is at most*

$$\frac{512\iota}{m\Delta_i} + 64\sqrt{\iota} \sqrt{\sum_{j=1}^{i} \frac{1}{A_j}}. \tag{5}$$

*Proof.* We'll break the the regret per action into "short stages" and "long stages", where both are "good stages". Specifically, we define $S_\tau = \{j : j \in G_\tau \ \& \ \tau_j/4 < m\}$ and break the regret per action into two:

$$2\left( \sum_{j=1, j \in S_\tau}^{i} \frac{\tau_j}{A_j} \Delta_i + \sum_{j=1, j \in G_\tau \setminus S_\tau}^{i} \frac{\tau_j}{A_j} \Delta_i \right). \tag{6}$$

For the first term above, using Lemma 10

$$\sum_{j=1, j \in G_\tau \setminus S_\tau}^{i} \frac{\tau_j}{A_j} \Delta_i = \frac{\Delta_i}{m} \sum_{j=1, \tau_j \in G_\tau \setminus S_\tau}^{i} \frac{\tau_j}{A_j} m$$

$$\leq \frac{256\iota}{m\Delta_i}. \tag{7}$$

For the second term, using Cauchy–Schwarz inequality

$$\sum_{j=1, j \in S_\tau}^{i} \frac{\tau_j}{A_j} \Delta_i \le \Delta_i \sqrt{\sum_{j=1, j \in S_\tau}^{i} \frac{\tau_j^2}{A_j}} \sqrt{\sum_{j=1, j \in S_\tau}^{i} \frac{1}{A_j}}$$

$$\le \Delta_i \sqrt{\frac{4 \cdot 256\iota}{\Delta_i^2}} \sqrt{\sum_{j=1}^{i} \frac{1}{A_j}}$$

$$\le \sqrt{1024\iota} \sqrt{\sum_{j=1}^{i} \frac{1}{A_j}}$$

$$= 32\sqrt{\iota} \sqrt{\sum_{j=1}^{i} \frac{1}{A_j}}.$$

where the second inequality is from Lemma 10. □

**Lemma 12.** $\sum_{i=1}^{A} \sqrt{\sum_{j=1}^{i} \frac{1}{A_j}} \le A$

*Proof.* Using Cauchy–Schwarz inequality

$$\sum_{i=1}^{A} \sqrt{\sum_{j=1}^{i} \frac{1}{A_j}} \le \sqrt{A} \sqrt{\sum_{i=1}^{A} \sum_{j=1}^{i} \frac{1}{A_j}}$$

$$= \sqrt{A} \sqrt{\sum_{j=1}^{A} \sum_{i=j}^{A} \frac{1}{A_j}}$$

$$= \sqrt{A} \sqrt{\sum_{j=1}^{A} \frac{A - j + 1}{A_j}}$$

$$= \sqrt{A} \sqrt{\sum_{j=1}^{A} 1}$$

$$= A.$$

□

**Theorem 6.** *When all the agents play* `Coop-SE`*, i.e., Algorithm 6, the individual regret of each agent* $v \in V$ *is*

$$\Re_T \le (\sum_{i \in A} \frac{512 \log(3mTA)}{m\Delta_i}) + 64 \cdot A\sqrt{\log(3mTA)} + 16A^2 + 1.$$

*Proof.* From Lemma 3, the complementary event of the good event adds no more than 1 to the regret. Let's assume that the good event hold.

From Lemma 9, the regret of all stages $j \notin G_\tau$ is at most $16A^2$. Using Lemma 12, summing over all actions we get the second term in Equation (5) is bounded by $64A\sqrt{\iota}$. Combining this with the other terms in Equation (5) yields the part of the bound corresponding to the good event. We get,

$$\Re_T \le (\sum_{i \in A} \frac{512 \log(3mTA)}{m\Delta_i}) + 64 \cdot A\sqrt{\log(3mTA)} + 16A^2 + 1.$$

□

**Theorem 7.** *When each agent plays* `Coop-SE` *the regret of each agent is also bounded by*

$$\mathfrak{R}_T \leq (\sum_{i \in A} \frac{512 \log(3mTA)}{m\Delta_i}) + 16AD + 1.$$

*Proof.* When we take into account only stages that are longer than $16D$ the other stages adds no more than $16AD$. The analysis of the regret that stems from the stages $\{j \mid \tau_j \geq 16D\}$ can be simplified, compared to when the stages are $\{j \mid \tau_j \geq 16A_j\}$. In each such stage $j$ in which $\tau_j \geq 16D$, $N_{\leq \tau_j/4}$ is the entire graph. So Lemma 5 becomes

$$\sum_{j=1, \tau_j \geq 16D}^{i} \frac{\tau_j}{A_j} m = \sum_{j=1, \tau_j \geq 16D}^{i} \frac{\tau_j}{A_j} |N_{\leq \tau_j/4}| \leq \frac{256\iota}{\Delta_a^2}.$$

And the regret in Equation (6) becomes only the first part. I.e., Equation (7) is the term that left. The term that was solved with Cauchy-Schwartz does not appear when the neighborhood is the entire graph. $\square$

**Proof of Theorem 2.** The proof follows immediately from the two regret bounds, Theorem 6 and Theorem 7. $\square$

# F    Proofs for Random Choices in `Coop-SE`

**Theorem 8.** *When all the agents play coop-SE-rand, i.e., Algorithm 6 with random choices, the individual regret of each agent $v \in V$ is,*

$$\mathfrak{R}_T^v = O\left(\sqrt{\frac{TA\log(mTA)}{m}} + A\log(mTA)\right).$$

A problem-specific flavor of an individual regret bound can also be established:

**Theorem 9.** *When all the agents play coop-SE-rand, i.e., Algorithm 6 with random choices, the individual regret of each agent $v \in V$ is*

$$\mathfrak{R}_T^v = O\left(\sum_{\Delta_a > 0} \frac{\log(mTA)}{m\Delta_a} + A\log(mTA)\right).$$

## F.1    Definitions

**Definition 15.** *Denote with $A_\Delta$ the set of elimination indices of large gaps. $A_\Delta = \{i | \Delta_{a_i} \geq \sqrt{\frac{A\iota}{Tm}}\}$.*

**Definition 16.** *Define the set of "Good Intervals" to be the set of long enough intervals: $G_\tau = \{j | \tau_j > 16\}$. These are the intervals we will focus in the proofs.*

**Definition 17.** *Denote the group of indices of short stages with $S_\tau$. Specifically,*

$$S_\tau := \{j | j \in G_\tau \ \& \ \tau_j/4 < m\}$$

**Definition 18.** *Denote the number of samples an agent $u$ sees for action $a$ until the beginning of timestep $t$ with $n_t^u(a)$. For the ease of notation, denote $n_t(a) := n_t^v(a)$.*

**Definition 19.** *Denote by $b_t^u(a)$ the number of times agent $u$ played action $a$ until the beginning of round $t$.*

## F.2    The Good Event

The first good event $G^1$ captures is the same as the one that was defined earlier.

**Lemma 13.** *Let $w$ be an agent and let $X_t^w(a) := \mathbb{I}(a_t^w = a)$ be the indicator that $w$ plays action $a$ at timestep $t$. Then for any agent $u$, timestep $t$, and action $a$,*

$$n_t^u(a) = \sum_{k=1}^{t-1} \sum_{w \in N_{\leq t-k}^u} X_k^w(a)$$

*Proof.* Let $w$ be an agent such that $w \neq u$ and $d_{\mathcal{G}}(w, u) = d$. Every $X_k^w(a)$ reaches $u$ at the end of round $k + d - 1$. Therefore, it contributes to $n_{t'}^u(a)$ at timestep $t' = k + d$. We get that for $w \neq u$, $w \in N_{\leq t-k}^u$, $X_k^w(a)$ reaches $u$ until the beginning of timestep $t$.

Now, let $w = u$ and $k < t$. An agent $u$ uses the information she creates only at the next timestep. Since we do not sum the information for the current timestep $t$, i.e., $t - k \geq 1$, the information $u$ creates is summed only for timesteps that passed. In other words, for $w = u$, $X_k^w(a)$ is summed only at timesteps $t' < t$, for them the information reaches $u$ until the beginning of $t$. Therefore, we get that for all $k < t$, $w \in N_{\leq t-k}^u$, $X_k^w(a)$ reaches $u$ until the beginning of timestep $t$. Summing over all the timesteps at which information on action $a$ can be produced and we obtain the result. $\square$

The second good event $G^2$ requires that the number of observations of an action is not much less than the expectation of the number of observations.

**Definition 20.** *Define the good event $G^2$ to be the event in which for all $u \in V$, action $a$ and timestep $t \in T$ simultaneously,*

$$n_t^u(a) \geq \frac{1}{2} \sum_{k=1}^{t-1} \sum_{w \in N_{\leq t-k}^u} p_k^w(a) - 2\iota.$$

The third good event $G^3$ requires that the number of plays of an action is not much more than the expectation of the number of plays.

**Definition 21.** *Define the good event $G^3$ to be the event in which for all $u \in V$, action $a$ and timestep $t \in T$ simultaneously,*

$$b_t^u(a) \leq 2 \sum_{k=1}^{t-1} p_k^u(a) + 12\iota.$$

**Definition 22.** *The good event is the event in which all the previous sub-good-events happen. I.e.,*
$$G := G^1 \cup G^2 \cup G^3$$

The following lemma show that with high probability all the good events hold.

**Lemma 14.** *When all agents play Algorithm 6 with random choices the good event, $G := G^1 \cup G^2 \cup G^3$, happens with probability of at least $1 - \frac{1}{T^2}$.*

*Proof.* We will show that each of the events $\neg G^1$, $\neg G^2$ and $\neg G^3$ happens with probability of at most $\frac{1}{3T^2}$. Thus, by the union bound, $G$ occur with probability of at least $1 - \frac{1}{T^2}$.

**Event $\neg G^2$:** Fix an action $a$ and agent $u$. Let $X_{k,w} = \mathbb{I}\{a_k^w = a\}$ and $\mathcal{F}_{t,w}$ be the sigma algebra induced by the first $t - 1$ rounds; and the actions chosen by the first $w - 1$ agents in round $t$ (where we assume a linear order on the agents - for example the alphabetic order induced by their IDs). Notice that $\mathcal{F}_{t,1}$ is induced simply by the first $t - 1$ rounds. Note that $p_k^w(a)$ is $\mathcal{F}_{k,w}$-measurable, $\mathbb{E}[X_{k,w} \mid \mathcal{F}_{k,w}] = p_k^w(a)$ and that $X_{k,w}$ is $\mathcal{F}_{k,w+1}$-measurable (or if $w$ is the last agent, $X_{k,w}$ is $\mathcal{F}_{k+1,1}$-measurable). By applying Lemma 46, with probability $1 - \frac{1}{9T^2 A^2 m^2}$ for all $t \in [T]$ simultaneously we have,

$$n_t^u(a) = \sum_{k=1}^{t-1} \sum_{w \in N_{\leq t-k}^u} X_k^w(a) \geq \frac{1}{2} \sum_{k=1}^{t-1} \sum_{w \in N_{\leq t-k}^u} p_k^w(a) - 2\iota,$$

where the equality is from Lemma 13. By taking the union bound over all actions, $a$, and agents $u$ we get that $\mathbb{P}(\neg G^2) \leq 1/(9mAT^2) \leq 1/(3T^2)$.

**Event $\neg G^3$:** Fix an action $a$, agent $u$ and timestep $t$. Let $X_k = \mathbb{I}\{a_k^u = a\}$ and $\mathcal{F}_t$ be the sigma algebra induced by the first $t - 1$ rounds. Note that $p_k^w(a)$ is $\mathcal{F}_k$-measurable, $\mathbb{E}[X_k \mid \mathcal{F}_k] = p_k^u(a)$ and that $X_k$ is $\mathcal{F}_{k+1}$-measurable. By applying Lemma 47, with probability $1 - \frac{1}{27T^3 A^3 m^3}$,

$$b_t^u(a) = \sum_{k=1}^{t-1} X_k(a) \leq 2 \sum_{k=1}^{t-1} p_k^u(a) + 12\iota.$$

By taking the union bound over all time steps $t$, actions $a$, and agents $u$ we have $\mathbb{P}(\neg G^3) \leq \frac{1}{27T^2A^2m^2} \leq \frac{1}{3T^2}$.

Taking the union bound over $\neg G^1 \cup \neg G^2 \cup \neg G^3$, and from Lemma 2, we complete the proof. $\qquad\square$

**Lemma 15.** *The complementary event of the good event adds no more than* 1 *to the regret of each agent.*

*Proof.* From Lemma 2, the complementary event of the good event happens in probability lower than $\frac{1}{T^2}$. Every agent plays $T$ timesteps, and the gaps are bounded by 1, i.e., for every action $a$ we have $\Delta_a \leq 1$. Hence, in expectation, this adds at most $\frac{1}{T} \leq 1$ to the regret. $\qquad\square$

### F.3 Proof of Theorem 8

In the proof from now on, we assume the good event $G := G^1 \cup G^2 \cup G^3$ holds.

**Remark 3.** *Note that in the proof of Lemma 1 we used $\tau_j^v > 16$ and not $\tau_j^v > 16A_j$. Hence, the results follows immediately here as well. We will use the same lemma, Lemma 1, here as well.*

**Lemma 16.** *For every action $a$ that was not eliminated before the end of stage $i$, we have*

$$n_{t_{i+1}-1}(a) \geq \sum_{j=1,j\in G_\tau}^{i} \frac{\tau_j}{16A_j}|N_{\leq \tau_j/4}| - 2\iota.$$

*Proof.* Under the good event $G^2$,

$$n_{t_{i+1}-1}(a) \geq \frac{1}{2} \sum_{t=1}^{t_{i+1}-2} \sum_{u \in N_{\leq t_{i+1}-t-2}} p_t^u(a) - 2\iota$$

$$\geq \frac{1}{2} \sum_{j=1}^{i} \sum_{t=t_j}^{t_j+\tau_j-2} \sum_{u \in N_{\leq t_{i+1}-t-1}} p_t^u(a) - 2\iota$$

$$\geq \frac{1}{2} \sum_{j=1,j\in G_\tau}^{i} \sum_{t=t_j+\lceil\tau_j/4\rceil}^{t_j+\lfloor\tau_j/2\rfloor} \sum_{u \in N_{\leq t_{i+1}-t-2}} p_t^u(a) - 2\iota$$

$$\geq \frac{1}{2} \sum_{j=1,j\in G_\tau}^{i} \sum_{t=t_j+\lceil\tau_j/4\rceil}^{t_j+\lfloor\tau_j/2\rfloor} \sum_{u \in N_{\leq \tau_j/4}} p_t^u(a) - 2\iota.$$

The second inequality is by splitting the rounds to stages and summing partially. The third inequality is by summing partially over $j \in G_\tau$ ($\lfloor\tau_j/2\rfloor \leq \tau_j/2 - 1 \leq \tau_j - 2$). The last inequality is since $N_{\leq \tau_j/4} \subseteq N_{\leq t_{i+1}-t-2}$ as for all $j \in [i] \cap G_\tau$ and $t \leq t_j + \lfloor\tau_j/2\rfloor$,

$$t_{i+1} - t - 2 \geq t_{j+1} - t_j - \lfloor\tau_j/2\rfloor - 2 \geq \tau_j - \tau_j/2 - 3 = \tau_j/2 - 3 \geq \tau_j/4.$$

Finally, by Lemma 1, all agents $u \in N_{\leq \tau_j/4}$ play the same policy at time steps $t \in [t_j + \lceil\tau_j/4\rceil, t_j + \lfloor\tau_j/2\rfloor]$ which is uniform over the active actions. I.e., $p_t^u(a) = \frac{1}{A_j}$ for active actions in $[t_j + \lceil\tau_j/4\rceil, t_j + \lfloor\tau_j/2\rfloor]$. The interval $[t_j + \lceil\tau_j/4\rceil, t_j + \lfloor\tau_j/2\rfloor]$ is of size at least $\tau_j/8$, since $t_j + \lceil\tau/4\rceil - t_j + \lfloor\tau/2\rfloor \geq \frac{\tau_j}{2} - \frac{\tau_j}{4} - 2 = \frac{\tau_j}{4} - 2 \geq \frac{\tau_j}{8}$, when the last inequality follows from the that for every $j \in G_\tau, \tau_j > 16$. Thus,

$$\frac{1}{2} \sum_{j=1,j\in G_\tau}^{i} \sum_{t=t_j+\lceil\tau_j/4\rceil}^{t_j+\lfloor\tau_j/2\rfloor} \sum_{u \in N_{\leq \tau_j/4}} p_t^u(a) \geq \sum_{j=1,j\in G_\tau}^{i} \frac{\tau_j}{16A_j}|N_{\leq \tau_j/4}|,$$

as desired. $\qquad\square$

**Lemma 17.** *For every action $a$ that was not eliminated before the end of stage $i$,*

$$\sum_{j=1,j\in G_\tau}^{i} \frac{\tau_j}{A_j}|N_{\leq \tau_j/4}| \leq \frac{544\iota}{\Delta_a^2}.$$

*Proof.* Fix an action $a$ that was not eliminated before the end of stage $i$. Denote $t' = t_{i+1} - 1$. The action $a$ is still active by agent $v$ at time $t'$, and thus, $UCB_{t'}^v(a) \geq LCB_{t'}^v(a^\star)$. Note the slightly abuse of notation, when $UCB_{t'}^v(a)$ is actually $UCB_{n_{t'}(a)}^v(a)$, and the same for $LCB$. Under the good event $G^1$,

$$\mu_a + 2\lambda_{t'}^v(a) \geq UCB_{t'}^v(a) \geq LCB_{t'}^v(a^\star) \geq \mu_{a^\star} - 2\lambda_{t'}^v(a^\star).$$

Rearranging it we get,

$$\Delta_a \leq 2\sqrt{\frac{2\iota}{n_{t'}(a)}} + 2\sqrt{\frac{2\iota}{n_{t'}(a^\star)}}.$$

Recall that under the good event, $a^\star$ is never eliminated. Thus, we can apply Lemma 16 on both $a$ and $a^\star$ and further bound $\Delta_a$ by,

$$\Delta_a \leq 4\sqrt{\frac{2\iota}{\sum_{j=1, j \in G_\tau}^{i} \frac{\tau_j}{16A_j}|N_{\leq \tau_j/4}| - 2\iota}},$$

then

$$\Delta_a^2 \leq 16\frac{2\iota}{\sum_{j=1, j \in G_\tau}^{i} \frac{\tau_j}{16A_j}|N_{\leq \tau_j/4}| - 2\iota},$$

we get

$$\sum_{j=1, j \in G_\tau}^{i} \frac{\tau_j}{16A_j}|N_{\leq \tau_j/4}| - 2\iota \leq \frac{32\iota}{\Delta_a^2},$$

and,

$$\sum_{j=1, j \in G_\tau}^{i} \frac{\tau_j}{16A_j}|N_{\leq \tau_j/4}| \leq \frac{32\iota}{\Delta_a^2} + 2\iota \leq \frac{34\iota}{\Delta_a^2}.$$

By rearranging terms we get the Lemma's statement. $\qquad\square$

**Lemma 18.** *When all agents plays* `Coop-SE` *with random choices, the regret of agent $v$ (under the good event) is bounded by*

$$\mathfrak{R}_T \leq 2 \sum_{i \in A_\Delta} \sum_{j=1, j \in G_\tau}^{i} \frac{\tau_j}{A_j}\Delta_i + \sqrt{\frac{TA\iota}{m}} + 44A\iota \tag{8}$$

*Proof.* Under the good event,

$$b_{t_{i+1}}(a_i) \leq 2 \sum_{t=1}^{t_{i+1}-1} p_k^v(a_i) + 12\iota$$

$$= 2 \sum_{j=1}^{i} \sum_{t=t_j}^{t_j+\tau_j-1} p_k^v(a_i) + 12\iota$$

$$= 2 \sum_{j=1}^{i} \frac{\tau_j}{A_j} + 12\iota$$

Now the regret can be bounded by,

$$\mathfrak{R}_T = \sum_{i \in [A]} b_{t_{i+1}}(a_i) \Delta_i$$

$$\leq 2 \sum_{i \in [A]} \sum_{j=1}^{i} \frac{\tau_j}{A_j} \Delta_i + 12 A \iota$$

$$\leq 2 \sum_{i \in A_\Delta} \sum_{j=1}^{i} \frac{\tau_j}{A_j} \Delta_i + \sum_{i \notin A_\Delta} b_{t_{i+1}}(a_i) \sqrt{\frac{A \iota}{Tm}} + 12 A \iota \qquad (9)$$

$$\leq 2 \sum_{i \in A_\Delta} \sum_{j=1}^{i} \frac{\tau_j}{A_j} \Delta_i + T \sqrt{\frac{A \iota}{Tm}} + 12 A \iota$$

$$\leq 2 \sum_{i \in A_\Delta} \sum_{j=1, j \in G_\tau}^{i} \frac{\tau_j}{A_j} \Delta_i + \sum_{i \in A_\Delta} \sum_{j=1, j \notin G_\tau}^{i} \frac{\tau_j}{A_j} \Delta_i + \sqrt{\frac{TA \iota}{m}} + 12 A \iota$$

$$\leq 2 \sum_{i \in A_\Delta} \sum_{j=1, j \in G_\tau}^{i} \frac{\tau_j}{A_j} \Delta_i + \sqrt{\frac{TA \iota}{m}} + 44 A \iota,$$

where the last is since,

$$\sum_{i \in A_\Delta} \sum_{j=1, j \notin G_\tau}^{i} \frac{\tau_j}{A_j} \Delta_i \leq \sum_{i \in A_\Delta} \sum_{j=1}^{i} \frac{16}{A_j} \leq A \sum_{j=1}^{A} \frac{16}{A_j} \leq 32 A \log A.$$

as $\sum_{j=1}^{A} \frac{1}{A_j} \leq \log A + 1$ by Lemma 7. $\qquad \square$

**Lemma 19.** *For every action elimination index $i \in A_\Delta$, it holds that*

$$\sum_{j=1, j \in S_\tau}^{i} \frac{\tau_j^2}{4 A_j} + \sum_{j=1, j \in G_\tau \setminus S_\tau}^{i} \frac{\tau_j}{A_j} m \leq \frac{544 \iota}{\Delta_i^2}$$

*where $S_\tau := \{j | j \in G_\tau \ \& \ \tau_j/4 < m\}$, and $\{\tau_j | j \in [A]\}$ are the stage lengths.*

*Proof.* From Lemma 17,

$$\sum_{j=1, j \in G_\tau}^{i} \frac{\tau_j}{A_j} |N_{\leq \tau_j/4}| \leq \frac{544 \iota}{\Delta_{a_i}^2}.$$

On the other hand, using Lemma 6

$$\sum_{j=1, j \in G_\tau}^{i} \frac{\tau_j}{A_j} |N_{\leq \tau_j/4}| \geq \sum_{j=1, j \in G_\tau}^{i} \frac{\tau_j}{A_j} \min\{m, \tau_j/4\}$$

$$= \sum_{j=1, j \in S_\tau}^{i} \frac{\tau_j^2}{4 A_j} + \sum_{j=1, j \in G_\tau \setminus S_\tau}^{i} \frac{\tau_j}{A_j} m$$

$\qquad \square$

**Proof of Theorem 8.** Let us write again the Right-Hand-Side of Equation (8)

$$2 \sum_{i \in A_\Delta} \sum_{j=1, j \in G_\tau}^{i} \frac{\tau_j}{A_j} \Delta_i + \sqrt{\frac{TA \iota}{m}} + 44 A \iota.$$

Note that the bound on the regret that is depicted in Equation (8) assumes that the good event holds, and we later will remove this assumption. Let's assume that the good event hold. We'll break the first sum in the Right-Hand-Side of Equation (8) as

$$\sum_{i\in A_\Delta}\sum_{j=1,j\in S_\tau}^{i}\frac{\tau_j}{A_j}\Delta_i + \sum_{i\in A_\Delta}\sum_{j=1,j\in G_\tau\setminus S_\tau}^{i}\frac{\tau_j}{A_j}\Delta_i. \tag{10}$$

and we remind that $S_\tau = \{j : j \in G_\tau \ \& \ \tau_j/4 < m\}$. For the first term above, using Lemma 19, for every $i \in A_\Delta$,

$$\sum_{j=1,j\in G_\tau\setminus S_\tau}^{i}\frac{\tau_j}{A_j}\Delta_i = \frac{\Delta_i}{m}\sum_{j=1,\tau_j\in G_\tau\setminus S_\tau}^{i}\frac{\tau_j}{A_j}m$$
$$\leq \frac{544\iota}{m\Delta_i} \tag{11}$$
$$\leq 544\sqrt{\frac{T\iota}{mA}}.$$

where the second inequality is since $i \in A_\Delta$. Summing over all elimination indices in $A_\Delta$ we get that the first term in Equation (10) is bounded by $544\sqrt{\frac{TA\iota}{m}}$.

For the second term, for every $i$, using Cauchy–Schwarz inequality

$$\sum_{j=1,j\in S_\tau}^{i}\frac{\tau_j}{A_j}\Delta_i \leq \Delta_i\sqrt{\sum_{j=1,j\in S_\tau}^{i}\frac{\tau_j^2}{A_j}}\sqrt{\sum_{j=1,j\in S_\tau}^{i}\frac{1}{A_j}}$$
$$\leq \Delta_i\sqrt{\frac{4\cdot544\iota}{\Delta_i^2}}\sqrt{\sum_{j=1}^{i}\frac{1}{A_j}}$$
$$\leq \sqrt{2176\iota}\sqrt{\sum_{j=1}^{i}\frac{1}{A_j}},$$

where the second inequality is from Lemma 19. Using Lemma 12, summing over all actions we get the second term in Equation (10) is bounded by $47A\sqrt{\iota}$. Combining this with the other terms in Equation (8) yields the part of the bound corresponding to the good event. From Lemma 15, the complementary event of the good events adds no more than 1 to the regret. We get,

$$\mathfrak{R}_T \leq 2\cdot544\sqrt{\frac{TA\iota}{m}} + 2\cdot47A\sqrt{\iota} + \sqrt{\frac{TA\iota}{m}} + 44A\iota + 1$$
$$\leq 1088\sqrt{\frac{TA\iota}{m}} + 94A\iota + \sqrt{\frac{TA\iota}{m}} + 44A\iota + 1$$
$$= 1089\sqrt{\frac{TA\iota}{m}} + 138A\iota + 1$$
$$= 1089\sqrt{\frac{TA\log(3mTA)}{m}} + 138A\log(3mTA) + 1.$$

$\square$

## F.4 Instance Dependent Bound

It is important to note that when the analysis is not split into large and small gaps, a bound specific to the problem instance can also be derived. We can conclude that the individual regret is bounded by,

$$\tilde{O}\Big(\sum_{a:\Delta_a>0}\frac{1}{m\Delta_a}\Big)$$

as depicted in Theorem 9.

Despite being a suitable bound for various scenarios, there are cases where it fails to provide a good approximation. For example, two action and the gap is $\Delta_a = 1/T \cdot m$. We will get regret which is linear in $T$. We have made this distinction between large and short gaps to be problem independent.

Although the changes that yield the instance dependent bound are simple, we provide for clarity the relevant parts where the proof changes.

**Lemma 20.** *Under the good event, the regret of agent $v$ is bounded by*

$$\mathfrak{R}_T \leq 2 \sum_{i \in [A]} \sum_{j=1, j \in G_\tau}^{i} \frac{\tau_j}{A_j} \Delta_i + 44A\iota. \tag{12}$$

*Proof.* The proof follows the same steps as Lemma 18, but without splitting the gaps as in Equation (9). □

**Lemma 21.** *Under the good event, the following holds,*

$$\sum_{i \in [A], \Delta_i > 0} \sum_{j=1, j \in G_\tau}^{i} \frac{\tau_j}{A_j} \Delta_i \leq \sum_{i \in [A], \Delta_i > 0} \frac{544\iota}{m\Delta_i} + 47A\sqrt{\iota}.$$

*Proof.* The proof follows the same steps as the proof of Theorem 8, but treating all non optimal actions the same, and stopping the analysis in Equation (11), i.e., without bounding the expression with $\sqrt{T\iota/mA}$. □

**Proof of Theorem 9.** The proof follows by combining the results of Lemma 20 and Lemma 21, and with the fact from Lemma 15 that the complementary event of the good events adds no more than 1 to the regret. We get,

$$\mathfrak{R}_T \leq 2 \sum_{i \in [A]} \sum_{j=1, j \in G_\tau}^{i} \frac{\tau_j}{A_j} \Delta_i + 44A\iota + 1$$

$$\leq 2 \cdot \sum_{i \in [A], \Delta_i > 0} \frac{544\iota}{m\Delta_i} + 2 \cdot 47A\sqrt{\iota} + 44A\iota + 1$$

$$\leq \sum_{i \in [A], \Delta_i > 0} \frac{1088\iota}{m\Delta_i} + 94A\iota + 44A\iota + 1$$

$$= \sum_{i \in [A], \Delta_i > 0} \frac{1088\iota}{m\Delta_i} + 138A\iota + 1$$

$$= \left( 1088 \sum_{i \in [A], \Delta_i > 0} \frac{\log(3mTA)}{m\Delta_i} \right) + 138A\log(3mTA) + 1.$$

□

## G  Lower Bound

**Theorem 10.** *For any algorithm, there exists an instance of the cooperative MAB over a communication graph problem, for which the individual regret of any agent is bounded from below by*

$$\Omega(\sqrt{A}) \leq \mathfrak{R}_T.$$

*Proof.* Let the graph be a line of length at least $T$. I.e., $m \geq T$. Let $A$ be the number of actions such that $\sqrt{A} > 20$. Let $a^\star$ be the only best action. Let $\Delta_a = 1$ for every $a \neq a^\star$. Namely the reward of $a^*$ is 1 and the rewards of the other actions $a \neq a^*$ is 0.

Let $v$ be an agent in the graph. After $t$ timesteps, the maximum number of samples $v$ sees, for all actions together, is no more than $2 \cdot (2 + t + 1)t/2 = (t + 3)t$ (twice the sum of arithmetic series). At timestep $\lfloor \sqrt{A}/20 \rfloor$ the agent sees at most $\frac{A+30\sqrt{A}}{400}$ samples for all the actions together.

From the assumption on $A$, $\frac{3\sqrt{A}}{40} \leq \frac{A}{200}$. It implies that

$$\frac{A + 30\sqrt{A}}{400} \leq \frac{A}{200} + \frac{3\sqrt{A}}{20} \leq \frac{A}{100}.$$

It means that until this timestep, the agent didn't see at least $0.99A$ of the actions.

Let us randomly choose an instantiation of the best action $a^*$. Define the random variable $X$ that chooses the best action uniformly. I.e., $\mathbb{P}(X = a) = \frac{1}{A}$. Denote the event in which the agent doesn't see the best action until timestep $\lfloor \frac{\sqrt{A}}{20} \rfloor$ with $\mathcal{E}$. From the above, event $\mathcal{E}$ happens with probability at least $\frac{99}{100}$. I.e., $\mathbb{P}(\mathcal{E}) \geq \frac{99}{100}$. Under event $\mathcal{E}$, from the assumption that $\Delta_a = 1$, the regret until this timestep is $\lfloor \frac{\sqrt{A}}{20} \rfloor$, and we get $\frac{\sqrt{A}}{20} - 1 \leq \mathfrak{R}_T$.

For any algorithm the agents play,

$$\mathbb{E}_X(\mathfrak{R}_T) \geq \frac{99}{100} \cdot (\frac{\sqrt{A}}{20} - 1).$$

Therefore, for any algorithm, there exists an instance such that $\mathfrak{R}_T \geq \frac{99}{100} \cdot (\frac{\sqrt{A}}{20} - 1)$. $\qquad\square$

# H   Bounded Communication

This section relies on the definitions and theorems that are depicted in Appendix E.

We introduce a new event type, the aggregated event for many rewards.

**Definition 23.** *A reward-many event is a tuple $(\texttt{rwdMany}, v, a, r, n)$ that represents an aggregation of many rewards, where $v$ is the agent's ID, $a$ is the action, $r$ is the reward, and $n$ is the number of samples of this event.*

**Remark 4.** *The good event occurs with probability higher than or equal to $1 - 1/T^2$, when all agents play Algorithm 9. Although this algorithm uses the $\texttt{rwdMany}$ events, the same proof of Lemma 2 applies also to them, but the graph is the induced tree.*

*Proof of restricted communication.* In Algorithm 9, we do not have duplicated messages. We achieve this by the tree structure, and by not sending to a neighbor $u$ information that $u$ already sent to $v$. The tree structure guarantees that there is only one path from an agent to another. This property ensures that a message originating from one agent will reach all other agents exactly once, as it traverses the tree along the single possible route. Consequently, the combination of the spanning tree structure and the selective forwarding of messages allows for efficient and duplicate-free communication among all agents.

The `Coop-SE-Restricted` algorithm aggregate all events regarding an action $a$ into two events: `rwdMany` for rewards and `elim` for elimination. The message contains information about action $a$, its elimination status, observation count, and sum of observed rewards, requiring $O(A \log(Am))$ bits. This is all the information agents need from multiple messages.

Therefore, the agent has exactly the same information if all agents had played Algorithm 6 on that spanning tree. The individual regret bound that is induced from `Coop-SE` does not depend on the structure of the graph, therefore the same regret bound applied for `Coop-SE-Restricted` as well.

The agent sends to each neighbor $2A$ events. Each event has $O(\log(Am))$ bits. Therefore each message is bounded by $O(A \log(Am))$ bits. This completes the proof. $\qquad\square$

## H.1   CONGEST Model: $O\left(\log(AmT)\right)$ Bits

**Remark 5.** *The problem independent regret bound for `Coop-SE-CONGEST` is*

$$\mathfrak{R}_T^v = O\left(\sqrt{\frac{TA\log(mTA)}{m}} + A^2 + A\log(mTA)\right).$$

**Lemma 22.** *Let $v, u$ be two agents such that either $v$ is a descendant of $u$ or $u$ is a descendant of $v$ (with respect to the root $w$). When all agents play Algorithm 11 every message sent from $v$ to $u$ arrives $d_{\mathcal{T}}(v, u)$ timesteps after it has been sent (and vice versa).*

*Proof.* Let us first assume that $v$ is descendant of $u$. Every message contains information about one action. Denote that action by $a$ for the message that has been sent from $v$. We will prove the lemma by induction on $d_{\mathcal{T}}(v, u)$.

$d_{\mathcal{T}}(v, u) = 1$: Immediately true.

Let's denote the timestep when the message was sent with $t_0$. Let's assume the claim is true for $d$, now assume the distance is $d + 1$.

The message is sent toward the root at $t_0 + d_{\mathcal{T}}(w, v) \equiv a \pmod{A}$. One of $u$'s children, $x$, is on the path between $v$ and $u$ and is with distance $d$ from $v$. Since $v$ is descendant of $u$, it is also a descendant of $x$. Therefore, from the induction hypothesis, at timestep $t_0 + d$, $x$ receives the message. The message is sent from $x$ to $u$ at timestep $t$ such that $t + d_{\mathcal{T}}(x, w) \equiv a \pmod{A}$. $t = t_0 + d$, since $t_0 + d + d_{\mathcal{T}}(x, w) = t_0 + d_{\mathcal{T}}(v, w) \equiv a \pmod{A}$. Then $u$ gets the message after $d_{\mathcal{T}}(v, x) + 1 = d_{\mathcal{T}}(v, u)$ timesteps.

Similarly, assume $u$ is a descendant of $v$. We will prove by induction on $d_{\mathcal{T}}(v, u)$. $d_{\mathcal{T}}(v, u) = 1$: Immediately true. Let's denote the timestep when the message was sent out with $t_0$. Let's assume it is true for $d$, now assume the distance is $d + 1$.

The message is sent from $v$ outward from the root at $t_0 - d_{\mathcal{T}}(w, v) \equiv a \pmod{A}$. Let $x$ be $u$'s parent and note that $x$ is also a descendant of $v$. Therefore, from the induction hypothesis, at timestep $t_0 + d$, $x$ receives the message. The message will be sent from $x$ to $u$ at timestep $t$ such that $t - d_{\mathcal{T}}(x, w) \equiv a \pmod{A}$. $t = t_0 + d$ since $t_0 + d - d_{\mathcal{T}}(x, w) = t_0 - d_{\mathcal{T}}(v, w) \equiv a \pmod{A}$. Then $u$ gets the message after $d_{\mathcal{T}}(v, x) + 1 = d_{\mathcal{T}}(v, u)$ timesteps. $\square$

**Lemma 23.** *When all agents play Algorithm 11 every reward information that arrives to one agent $v$ at timestep $t$, and was not produced by another agent $u$, arrives to agent $u$ at most at $t + d_{\mathcal{T}}(v, u) + 2A$. Where reward information is a reward from some action some agent experienced.*

*Proof.* Let's denote with $x$ the common ancestor of $v$ and $u$, i.e., the closest agent to the root among all the agents on a shortest path from $v$ to $u$. Notice that it is possible that $v = x$, and that $u = x$. We have that both $v$ and $u$ are either $x$ itself or descendants of $x$. The reward information that reaches $v$ at timestep $t$ can wait $A$ timesteps at $v$ before being sent, since $v$ sends the actions in round robin. From Lemma 22, after being sent from $v$ toward the root, the message that contains the reward information arrives to $x$ after $d_{\mathcal{T}}(v, x)$. At $x$, it might wait again for $A$ timesteps, because of the round-robin sending of the actions. After the message is sent from $x$ to $u$, it takes $d_{\mathcal{T}}(x, u)$ timesteps to arrive at $u$, as per Lemma 22. Overall it took the message to pass from $v$ to $u$ no more than $d_{\mathcal{T}}(v, u) + 2A$ timesteps. $\square$

### H.1.1 Good Event

The good event for this section is exactly as in Appendix E.

### H.1.2 Adjusting the Proofs

**Definition 24.** *The "good intervals", $G'_{\tau}$, from now on are $\tau_j > 32A$. $G'_{\tau} = \{j | \tau_j > 32A\}$.*

**Lemma 24.** *Assume all agents play* `Coop-SE-CONGEST`*. Let $j$ be a stage index such that $\tau_j > 32A$. Then every agent $u \in N_{\leq \tau_j/4}$ plays the same policy (i.e., has the same set of active actions) at time interval $[t_j + \lceil 3\tau_j/8 \rceil, t_j + \lfloor \tau_j/2 \rfloor]$.*

*Proof.* Let $t \in [t_j + \lceil 3\tau_j/8 \rceil, t_j + \lfloor \tau_j/2 \rfloor]$ and let $u \in N_{\leq \tau_j/4}$.

Denote the active actions of $v$ in the $j$'th stage as $\mathcal{A}_j$. We will show that an action $a$ is active for $u$ at $t$ iff $a \in \mathcal{A}_j$.

Let $a$ be an active action of $u$ at time $t$. Since $u \in N_{\leq \tau_j/4}$, we have $d_{\mathcal{G}}(u, v) \leq \tau_j/4$.

From Lemma 23 $u$ gets all $v$'s eliminations (the first $j-1$ eliminated actions) until the beginning of round $t_j + \lfloor \tau_j/4 \rfloor + 2A$. Since $\tau_j > 16A$ we get $t_j + \lfloor \tau_j/4 \rfloor + 2A < t_j + \tau_j/4 + \tau_j/8 = t_j + 3\tau_j/8$.

By the stage's definition, the agent $v$ does not send any elimination event about one of her active actions until the end of the stage. Therefore, for any $t' \geq t_j + \lceil 3\tau_j/8 \rceil$, $u$ does not have any active action which is not in $\mathcal{A}_j$. Hence, $a \in \mathcal{A}_j$.

Let $a$ be an action in $\mathcal{A}_j$. We will show that $a$ is an active action of $u$ at time $t$. Assume for contradiction that $u$, at timestep $t_j + \lfloor \tau_j/2 \rfloor$ or before, encounters an elimination of $a$. From Lemma 23, the elimination event should arrive to $v$ in no more than $\lfloor \tau_j/4 + 2A \rfloor < \lfloor 3\tau_j/8 \rfloor$ timesteps, so $v$ should get the elimination event at most at timestep $t_j + \lfloor \tau_j/2 \rfloor + \lfloor 3\tau_j/8 \rfloor \leq t_j + \frac{7\tau_j}{8}$. But $\tau_j > 16A$, then $\tau_j/8 > 2A \geq 2$, so $t_j + \frac{7\tau_j}{8} < t_{j+1} - 2$. Therefore, the elimination event about an action in $\mathcal{A}_j$ should arrive to $v$ at least 2 timesteps before stage $j+1$ begin. Contradiction. Therefore, $a$ is an active action of $u$ at $t$.

We get that for every $t \in [t_j + \lceil 3\tau_j/8 \rceil, t_j + \lfloor \tau_j/2 \rfloor]$ and for every $u \in N_{\leq \tau_j/4}$, the active actions of $u$ at $t$ are exactly $\mathcal{A}_j$. In other words, we get that in time interval $[t_j + \lceil 3\tau_j/8 \rceil, t_j + \lfloor \tau_j/2 \rfloor]$ all agents in $N_{\leq \tau_j/4}$ plays the same policy, i.e., choosing randomly from $\mathcal{A}_j$. $\qquad \square$

**Lemma 25.** *When all agent play* `Coop-SE-CONGEST` *(Algorithm 11), each sends no more than* $O(\log(mA))$ *bits per messages.*

*Proof.* According to `Send-One-Action` procedure, agents sends only one `elim` event or one `rwdMany` event. An `elim` message is of size $1 + \log(m) + \log(A)$. A `rwdMany` message is of size $1 + \log(m) + \log(A) + 2\log(m)$. Together we get $O(\log(mA))$ bits. $\qquad \square$

**Lemma 26.** *For every action $a$ that was not eliminated before the end of stage $i$, we have*

$$n_{t_{i+1}-1}(a) \geq \sum_{j=1, j \in G_\tau}^{i} \frac{\tau_j}{32 A_j} |N_{\leq \tau_j/4}|.$$

*Proof.* By Lemma 24, all agents $u \in N_{\leq \tau_j/4}$ have the same active set on the interval $t \in [t_j + \lceil 3\tau_j/8 \rceil, t_j + \lfloor \tau_j/2 \rfloor]$. By lemma 23, every pull of $u \in N_{\leq \tau_j/4}$ that is sampled before time $t_j + \lfloor \tau_j/2 \rfloor$ is observed by $v$ by time

$$t_j + \lfloor \tau_j/2 \rfloor + \lfloor \tau_j/4 \rfloor + 2A \leq t_j + \lfloor \tau_j/2 \rfloor + \lfloor \tau_j/4 \rfloor + \tau_j/16 \qquad \text{(since } j \in G'_\tau)$$
$$\leq t_j + \tau_j = t_{j+1} - 1 \leq t_{i+1} - 1 \qquad \text{(for } j \leq i)$$

The interval $[t_j + \lceil 3\tau_j/8 \rceil, t_j + \lfloor \tau_j/2 \rfloor]$ is of size at least $\tau_j/16$, since $t_j + \lfloor \tau_j/2 \rfloor - t_j - \lceil 3\tau_j/8 \rceil \geq \frac{\tau_j}{2} - 1 - 3\frac{\tau_j}{8} - 1 = \frac{\tau_j}{8} - 2 \geq \frac{\tau_j}{16}$, when the last inequality follows from the that for every $j \in G'_\tau, \tau_j > 32A > 32$. Thus, the number of samples from each active action at stage $j$ that each agent in $N_{\leq \tau_j/4}$ gathers is at least $\lfloor \frac{\tau_j}{16A_j} \rfloor \geq \frac{\tau_j}{16A_j} - 1 \geq \frac{\tau_j}{32A_j}$ for $j \in G'_\tau$. Moreover, these samples are observed by $v$ by time $t_{i+1} - 1$. In total,

$$n_{t_{i+1}-1}(a) \geq \sum_{j=1, j \in G_\tau}^{i} \frac{\tau_j}{32 A_j} |N_{\leq \tau_j/4}|.$$

$\qquad \square$

**Lemma 27.** *When all agent play Algorithm 11, for every action $a$ that was not eliminated before the end of stage $i$,*

$$\sum_{j=1, j \in G_\tau}^{i} \frac{\tau_j}{A_j} |N_{\leq \tau_j/4}| \leq \frac{1088\iota}{\Delta_a^2}.$$

*Proof.* The proof follows the same steps as the proof of Lemma 5 but employs Lemma 26 instead of Lemma 4. The claim involves a slightly different constants due to the factor of $1/32$ in Lemma 26 as opposed to $1/16$ in Lemma 4. $\qquad \square$

**Lemma 28.** *When all agent play Algorithm 11, the regret of agent $v$ (under the good event) is bounded by*

$$\mathfrak{R}_T \leq 2 \sum_{i \in A_\Delta} \sum_{j=1, j \in G_\tau}^{i} \frac{\tau_j}{A_j} \Delta_i + \sqrt{\frac{TA\iota}{m}} + 16A^2. \tag{13}$$

*Proof.* Similar to the proof of Lemma 8 and Lemma 18,

$$\mathfrak{R}_T \leq 2 \sum_{i \in A_\Delta} \sum_{j=1, j \in G_\tau}^{i} \frac{\tau_j}{A_j} \Delta_i + \sum_{i \in A_\Delta} \sum_{j=1, j \notin G_\tau}^{i} \frac{\tau_j}{A_j} \Delta_i + \sqrt{\frac{TA\iota}{m}}$$

$$\leq 2 \sum_{i \in A_\Delta} \sum_{j=1, j \in G_\tau}^{i} \frac{\tau_j}{A_j} \Delta_i + \sqrt{\frac{TA\iota}{m}} + 16A^2.$$

where the last is since,

$$\sum_{i \in A_\Delta} \sum_{j=1, j \notin G_\tau}^{i} \frac{\tau_j}{A_j} \Delta_i \leq \sum_{i \in A_\Delta} \sum_{j=1}^{i} \frac{32A_j}{A_j} \leq 32 \sum_{i \in A_\Delta} \sum_{j=1}^{i} 1 \leq 32 \frac{A(A-1)}{2} \leq 16A^2.$$

$\square$

**Proof of Theorem 4.** Let us write again the Right-Hand-Side of Equation (13)

$$2 \sum_{i \in A_\Delta} \sum_{j=1, j \in G_\tau}^{i} \frac{\tau_j}{A_j} \Delta_i + \sqrt{\frac{TA\iota}{m}}.$$

Similarly to the proof of Theorem 2, we'll break the first sum in the Right-Hand-Side of Equation (13) as

$$\sum_{i \in A_\Delta} \sum_{j=1, j \in S_\tau}^{i} \frac{\tau_j}{A_j} \Delta_i + \sum_{i \in A_\Delta} \sum_{j=1, j \in G_\tau \setminus S_\tau}^{i} \frac{\tau_j}{A_j} \Delta_i. \tag{14}$$

We can adjust Lemma 10, the only change is the constant of 1088 instead of 256. Using this adjusted lemma, we get that for every $i \in A_\Delta$,

$$\sum_{j=1, j \in G_\tau \setminus S_\tau}^{i} \frac{\tau_j}{A_j} \Delta_i = \frac{\Delta_i}{m} \sum_{j=1, \tau_j \in G_\tau \setminus S_\tau}^{i} \frac{\tau_j}{A_j} m$$

$$\leq \frac{1088\iota}{m\Delta_i} \tag{15}$$

$$\leq 1088 \sqrt{\frac{T\iota}{mA}}.$$

Similarly to the proof of Theorem 2, for the second term, for every $i$, using Cauchy–Schwarz inequality

$$\sum_{j=1, j \in S_\tau}^{i} \frac{\tau_j}{A_j} \Delta_i \leq \Delta_i \sqrt{\sum_{j=1, j \in S_\tau}^{i} \frac{\tau_j^2}{A_j}} \sqrt{\sum_{j=1, j \in S_\tau}^{i} \frac{1}{A_j}}$$

$$\leq \Delta_i \sqrt{\frac{4 \cdot 1088\iota}{\Delta_i^2}} \sqrt{\sum_{j=1}^{i} \frac{1}{A_j}}$$

$$\leq \sqrt{4352\iota} \sqrt{\sum_{j=1}^{i} \frac{1}{A_j}},$$

Using Lemma 12, summing over all actions we get the second term in Equation (14) is bounded by $66A\sqrt{\iota}$. Combining this with the other terms in Equation (13) yields the part of the bound corresponding to the good event. From Lemma 3, the complementary event of the good events adds no more than 1 to the regret. We get,

$$\mathfrak{R}_T \le 2 \cdot 1088\sqrt{\frac{TA\iota}{m}} + 2 \cdot 66A\sqrt{\iota} + \sqrt{\frac{TA\iota}{m}} + 16A^2 + 1$$

$$= 2177\sqrt{\frac{TA\iota}{m}} + 132A\sqrt{\iota} + \sqrt{\frac{TA\iota}{m}} + 16A^2 + 1$$

$\square$

A problem-specific flavor of an individual regret bound can also be found: The proof is similar to the proof of Theorem 6.

**Lemma 29.** *Under the good event, the regret of agent $v$ is bounded by*

$$\mathfrak{R}_T \le 2 \sum_{i \in [A]} \sum_{j=1, j \in G_\tau}^{i} \frac{\tau_j}{A_j}\Delta_i + 16A^2. \tag{16}$$

*Proof.* The proof follows the same steps as Lemma 28, but without splitting the analysis for small gaps and large gaps.

$\square$

**Lemma 30.** *Under the good event, the following holds,*

$$\sum_{i \in [A], \Delta_i > 0} \sum_{j=1, j \in G_\tau}^{i} \frac{\tau_j}{A_j}\Delta_i \le \sum_{i \in [A], \Delta_i > 0} \frac{1088\iota}{m\Delta_i} + 66A\sqrt{\iota}.$$

*Proof.* The proof follows the same steps as the proof of Theorem 4, but treating all non-optimal actions the same, and stopping the analysis in Equation (15), i.e., without bounding the expression with $\sqrt{T\iota/mA}$.
$\square$

**Proof of Theorem 4 (instance-dependent bound).** The proof follows by combining the results of Lemma 29 and Lemma 30, and with the fact from Lemma 3 that the complementary event of the good events adds no more than 1 to the regret. We get,

$$\mathfrak{R}_T \le 2 \sum_{i \in [A]} \sum_{j=1, j \in G_\tau}^{i} \frac{\tau_j}{A_j}\Delta_i + 16A^2 + 12A\iota + 1$$

$$\le 2 \cdot \sum_{i \in [A], \Delta_i > 0} \frac{1088\iota}{m\Delta_i} + 2 \cdot 66A\sqrt{\iota} + 16A^2 + 1$$

$$\le \sum_{i \in [A], \Delta_i > 0} \frac{2176\iota}{m\Delta_i} + 132A\sqrt{\iota} + 16A^2 + 1.$$

$\square$

## H.2 Communication Cost - Low Number of Messages

---

**Algorithm 4** `Create-Clusters`

---

 1: **Input:** Communication tree $\mathcal{T} = (V, E)$ with root $w \in V$, stage $i$
 2: **Define:** $V_u := \{v \in V \mid d_{\mathcal{T}}(u, v) \leq 2^i - 1\}$
 3: Initialize the set of clusters: $\mathcal{C} \leftarrow \emptyset$
 4: **for** each $u$ such that $d_{\mathcal{T}}(w, u) = 2^i$ **do**
 5:     Let $\mathcal{T}_u = (V_u, E_u)$ be the sub-tree of $\mathcal{T}$ rooted at $u$
 6:     **if** $|V_u| < 2^i$ **then**
 7:        $V_w \leftarrow V_w \cup V_u$
 8:     **else**
 9:        // then $|V_u| = 2^i$
10:        run `Create-Clusters` on $\mathcal{T}_u$ with root $u$ and stage $i$, and get output $\mathcal{C}_u$
11:        Add $\mathcal{C}_u$, the set of clusters when $u$ is the root, to the set of all clusters: $\mathcal{C} \leftarrow \mathcal{C} \cup \mathcal{C}_u$
12:     **end if**
13: **end for**
14: $\mathcal{C} \leftarrow \mathcal{C} \cup \{V_w\}$
15: **return** $\mathcal{C}$

---

**Definition 25.** *Let $\mathcal{T}$ be a tree, $\mathcal{T} = (V, E)$. We say that $\mathcal{C}$ is a clustering of $\mathcal{T}$ if $\mathcal{C}$ is a partition of $V$ and each set of nodes $W \in \mathcal{C}$ is connected.*

**Definition 26.** *Let $\mathcal{T}$ be a tree, $\mathcal{T} = (V, E)$ and $\mathcal{C}$ a clustering of $\mathcal{T}$. For each cluster $W \in \mathcal{C}$ we define the cluster's root of $W$ to be the root node of the sub-tree that is defined by $W$. The cluster's root of the cluster in which agent $v$ resides in the phase $i$ is denoted with* `cluster-root`$_i(v)$*, or simply* `cluster-root` *when the context is clear. Every connected sub-graph of a tree is a tree, so the existence and uniqueness immediate follows.*

**Definition 27.** *Let a tree $\mathcal{T} = (V, E)$ and $\mathcal{C}$ a clustering of $\mathcal{T}$. For each cluster $W \in \mathcal{C}$, we say that $u \in W$ is a cluster boundary if $u$ has no children in $W$. When $u$ is a cluster boundary node for a cluster in which agent $v$ resides in the phase $i$, we denote it with* `cluster-boundary`$_i(v)$*, or simply* `cluster-boundary` *if the context is clear.*

**Lemma 31.** *Let $\mathcal{C}$ be the clustering that* `Create-Clusters` *outputs with stage $i$. The following properties hold:*

1. *Each cluster $W \in \mathcal{C}$ is at a size of at least $\min\{2^i, m\}$.*

2. *For each cluster $W \in \mathcal{C}$ and its associated cluster root $w$, for any agent $u \in W$, $d_{\mathcal{T}}(w, u) \leq 2^{i+1}$.*

*Proof.* We will prove by induction on the number of times `Create-Clusters` is called recursively. For the base case, assume that `Create-Clusters` is called only once. For the first property, note that since `Create-Clusters` is not called again, the only cluster added is $V_w$, which at this point is $V$ itself, and the size is $m$.

For the second property, note that the only cluster can either contain nodes that are at most $2^i - 1$ distance from the root $w$ (added in line 2), or nodes that are added if there are not enough nodes to create a full cluster in the tail (when $|V_u| < 2^i$). For the latter, since $|V_u| < 2^i$ for any $v \in V_u$, it follows that $d_{\mathcal{T}}(v, u) \leq 2^i$. Also $d_{\mathcal{T}}(w, u) = 2^i$ by definition, and thus $d_{\mathcal{T}}(w, v) \leq 2^{i+1}$.

Moving to the induction step, assume that the properties hold whenever `Create-Clusters` is called at most $n$ times. Now, consider a run where `Create-Clusters` is called $n + 1$ times. It means that every call for `Create-Clusters` in this run which is not the outer call, holds the two properties. Now we will show that the outer call (the first call) to the algorithm holds these two conditions as well. For the first property, if a cluster is added in line 14, then since $V_w$ is at least of size $\min\{m, 2^i\}$ (line 2) and can only increase in line 7, it satisfies the property. If it is added in line 11, then since we call `Create-Clusters` on a sub-tree with at least $2^i$ nodes in line 10, by the induction assumption any cluster that it outputs is of size at least $2^i$.

For the second property, similar to the base case, for the cluster added in line 14, any node is added either in line 2 (in which case it satisfies the condition) or in line 7, in which case since $|V_u| < 2^i$, for

any $v \in V_u$, $d_\mathcal{T}(v, u) \leq 2^i$. And for the latter $d_\mathcal{T}(w, v) \leq d_\mathcal{T}(w, u) + d_\mathcal{T}(u, v) \leq 2^{i+1}$. Since the induction assumption hold for `Create-Clusters` in line 10, the all clusters that are added in line 11 satisfy the second property.

$\square$

**Lemma 32.** *Let $\mathcal{C}$ be the clustering that* `Create-Clusters` *outputs at phase $i$ when run on the tree $\mathcal{T}$ rooted at $w$. $u \neq w$ is a cluster root **if and only if** $d_\mathcal{T}(u, w) = k \cdot 2^i$ for some $k \in \mathbb{N}$ and it has at least $2^i - 1$ descendants.*

*Proof.* Cluster roots are at distance $d_\mathcal{T}(u, w) = k \cdot 2^i$ for some $k \in \mathbb{N} \cup \{0\}$ by construction: The initial cluster root is $w$ itself. At the first recursive level, cluster roots are $2^i$ away from $w$. At the second level, they are $2^i$ from a cluster root at the first level, that is, at a distance of $2 \cdot 2^i$ from $w$. And so on for subsequent levels. Hence, if $u \neq w$ is a cluster root, then $d_\mathcal{T}(u, w) = k \cdot 2^i$ for some $k \in \mathbb{N}$. Furthermore, $u$ must have served as a root at line 10 of the algorithm. Specifically, under the "else" condition, $u$ is guaranteed to have at least $2^i - 1$ descendants.

Conversely, if $d_\mathcal{T}(u, w) = k \cdot 2^i$ for some $k \in \mathbb{N}$, it will be iterated over at some recursion level in line 4. If it also has $2^i - 1$ descendants, then it will reach line 10 as the tree root and thus will be a cluster root.

$\square$

**Lemma 33.** *Let $\mathcal{C}_i$ and $\mathcal{C}_{i+1}$ be the clustering that* `Create-Clusters` *outputs at phases $i$ and $i + 1$ respectively. If $U \in \mathcal{C}_i$ then $U$ is contained in some cluster in $\mathcal{C}_{i+1}$.*

*Proof.* Let $U \in \mathcal{C}_i$, let $u$ its cluster root and let some $v \in U$.

- If $u$ is also a cluster root at $\mathcal{C}_{i+1}$. We will show that $v$ is in $u$'s cluster also at phase $i + 1$.

    - If $d(u, v) \leq 2^i - 1$, then since $u$'s cluster at phase $i$ contains $\{v' \in V \mid d_\mathcal{T}(u, v') \leq 2^{i+1} - 1\}$, it also contains $v$.
    - Otherwise, $v \in U$ is of distance at least $2^i$ from $u$. Let $v' \in U$ be the node between $v$ and $u$ that is of distance exactly $2^i$ from $u$. $v'$ is not a cluster root (since there is a unique cluster root in $U$). Thus, by Lemma 32, it has no more than $2^i - 1$ decedents. Therefore, $v'$ is not a cluster root also in phase $i + 1$. We conclude that there are no cluster roots between $v$ and $u$ also at phase $i + 1$, meaning that $v$ must be in $u$'s cluster also in phase $i + 1$.

- If $u$ is not a cluster root at $\mathcal{C}_{i+1}$

    - If $u$ is at distance $k' \cdot 2^{i+1}$ ($k' \in \mathbb{N}$) from $w$ then since $u$ is not a cluster root at phase $i + 1$ we know that it has less than $2^{i+1} - 1$ descendants (otherwise, Lemma 32, it was still be a cluster root). In this case, none of the descendants of $u$ are cluster roots. In particular, they are all descendants of the same cluster root at phase $i + 1$ and thus contained in the same cluster as $u$ itself.
    - Otherwise, we want to claim that each $v \in U$ is not a cluster root in phase $i$. Assume by contradiction that $v \in U$ is a cluster root at phase $i + 1$, then by Lemma 32, $d(v, w) = k' \cdot 2^{i+1}$ ($k' \in \mathbb{N}$) and has at least $2^{i+1} - 1$ descendants. In this case, again by Lemma 32, $v$ was also a cluster root at phase $i$. In particular $v \notin U$ since $u$ is the unique cluster root in $U$. A contradiction. Since $U$ does not contain cluster roots at phase $i + 1$ they all must be descendants of the same cluster root at phase $i + 1$ and thus contained in the same cluster as $u$ itself.

$\square$

**Remark 6.** *Throughout this section, we assume that, with a lack of other context, we always refer to a specific agent $v$.*

**Remark 7.** *The good event in this section is the same as $G^1$ in Definition 14, and from now on we assume it holds.*

**Definition 28.** *The length of the cluster is the distance between the* `cluster-root` *and the farthest* `cluster-boundary`.

**Definition 29.** *A phase in the context of* `Coop-SE-Comm-Cost` *is all the timesteps between the changes of the counter $i$. Specifically, a phase $i$ is all timesteps $[3 \cdot (2^i - 1), 3 \cdot (2^{i+1} - 1))$, and for the last phase, it is $[3 \cdot (2^i - 1), T]$ (for $i = \lceil \log(T/6) \rceil - 1$).*

**Definition 30.** *The active set of actions of $v$ at phase $i$ is the set of actions that* `cluster-root` *sent in that phase, and it is denoted by $\mathcal{A}_i$. We denote its size by $|\mathcal{A}_i| = A_i$. An action $a$ is denoted as "active" in phase $i$ if it belongs to $\mathcal{A}_i$. The last phase of $a$ is sometimes denoted with $i_a$. We denote $A_0 := A$ and $A_{i'+1} := 0$ where $i'$ is the global last phase.*

**Remark 8.** *Notice that while some actions that are not active for $v$ in phase $i$ may be played in phase $i$ by $v$. This is since the active set of actions didn't propagate yet to $v$. Eventually, the set of active actions will arrive $v$, and in the last third of the phase $v$ will play only active actions.*

**Lemma 34.** *The set of active actions (for vertex $v$) is non-increasing. I.e., for any phase $i$, $\mathcal{A}_i \supseteq \mathcal{A}_{i+1}$.*

*Proof.* Let $u_i$ and $u_{i+1}$ be $v$'s cluster root at phases $i$ and $i + 1$, respectively.

If $u_i = u_{i+1}$ then by definition $\mathcal{A}_{i+1} = \mathcal{A}_i \backslash B$ where $B$ is the set of actions eliminated in that phase (either due to the elimination step of due to elimination messages). In particular $\mathcal{A}_{i+1} \subseteq A_i$.

If $u_{i+1} \neq u_i$, then by Lemma 33, since $v$ and $u_i$ are on the same cluster at phase $i$, they are also on the same cluster in phase $i + 1$. That is $u_{i+1}$ is also the cluster root of $u_i$ in phase $i + 1$. In particular, at the beginning of the phase $u_i$ pass a message to $u_{i+1}$ with the aggregated eliminations which contain at least all of the inactive actions at phase $i$. In particular, if an action was inactive at phase $i$ it will be also inactive at phase $i + 1$. That is, $\mathcal{A}_{i+1} \subseteq A_i$. $\square$

**Lemma 35.** *When all agents play* `Coop-SE-Comm-Cost` *(Algorithm 13) then for every phase $i \geq \log_2(A)$ except the last phase, and for every active action in that phase $a$, each agent in the cluster of $v$ plays action $a$ at least $2^i / A_i$ times in phase $i$.*

*Proof.* In the middle of each phase, there are $2^{i+1}$ rounds in which the agent plays the active actions $\mathcal{A}_i$. The number of times the agent plays active action is at least $\lfloor 2^{i+1}/A_i \rfloor$. We get

$$\lfloor \frac{2^{i+1}}{A_i} \rfloor \geq \frac{2^{i+1}}{A_i} - 1 \geq \frac{2^i}{A_i},$$

where the last inequality is from $i \geq \log_2(A_i)$. $\square$

**Lemma 36.** *When all agents play* `Coop-SE-Comm-Cost` *(Algorithm 13) then after the gathering part of phase $i$ s.t. $i \geq \log_2(A) + 1$, the root of the cluster has at least $2^{i-1} \cdot \min\{2^{i-1}, m\}/A_{i-1}$ samples for every active action in that phase.*

*Proof.* Let $a$ be an active action in phase $i$, i.e., $a \in \mathcal{A}_i$. From lemma 34, $a$ was active at $\mathcal{A}_{i-1}$. From Lemma 35, at phase $i - 1$ each agent in the same cluster of $v$ in the previous stage $i - 1$ plays at least $2^{i-1}/A_{i-1}$ times action $a$. The size of the cluster at phase $i - 1$ is at least $\min\{2^{i-1}, m\}$, from Lemma 31. By 2. in Lemma 31 the distance between each agent in the cluster and the cluster root is at most $2^{i+1}$, hence all agents contribute all their samples. Together it completes the proof. $\square$

**Lemma 37.** *Assume that $i$ is a phase in which action $a$ is active, and $i \geq \log_2(A) + 1$. Then*

$$\Delta_a \leq 4\sqrt{2 \log(3mTA) \cdot \frac{A_{i-1}}{2^i \cdot \min\{2^i, m\}}}.$$

*Proof.* Since action $a$ wasn't eliminated at phase $i$ for the agent $v$, it means the `cluster-root` of $v$ at this phase didn't eliminate it in the beginning of phase $i$.

From Lemma 36, we know that the counters of `cluster-root`, when the `cluster-root` is doing the eliminations at phase $i$, are at least $n(a) \geq 2^{i-1} \cdot \min\{2^{i-1}, m\}/A_{i-1}$, $n(a^\star) \geq 2^{i-1} \cdot \min\{2^{i-1}, m\}/A_{i-1}$. Notice that $2^{i-1} \cdot \min\{2^{i-1}, m\}/A_{i-1} \geq \frac{1}{4} \cdot 2^i \cdot \min\{2^i, m\}/A_{i-1}$.

The root didn't eliminate the action $a$, hence

$$\mu_a + \sqrt{\frac{2\iota}{n(a)}} \geq \mu_{a^\star} - \sqrt{\frac{2\iota}{n(a)}}.$$

We get

$$\Delta_a \leq 4\sqrt{2\iota \cdot \frac{A_{i-1}}{2^i \cdot \min\{2^i, m\}}}.$$

$\square$

**Lemma 38.** *Assume that $i$ is the last phase in which action $a$ is active, and $i \geq \log_2(A)$. Then the number of times $v$ plays action $a$ in phases $j \geq \log_2(A)$ is at most*

$$48\frac{2^i}{A_i}.$$

*Proof.* Let us analyze each sub-phase part of a phase $j$ in which action $a$ is active. In the first third of the phase, where agents gather the information in the cluster and send it to the root, the agent plays action $a$ at most $\lceil 2^{j+1}/A_{j-1}\rceil$. From lemma 34 we have $A_j \leq A_{j-1}$, than in this sub-phase the agent plays action $a$ at most $\lceil 2^{j+1}/A_j\rceil$ times. In the second third of the phase the agent get the new active set of actions, $\mathcal{A}_j$. So part of this third is with $\mathcal{A}_j$ and part with $\mathcal{A}_{j-1}$. Then we can bound the number of plays in this part with $\lceil 2^{j+1}/A_j\rceil$. In the last third the agent plays actions from $\mathcal{A}_j$, then this third contributes no more than $\lceil 2^{j+1}/A_j\rceil$ samples. We get that in each phase $j$ where action $a$ is active, agent $v$ plays action $a$ at most $3\lceil 2^{j+1}/A_j\rceil$.

If $i$ is not the global last phase, it is possible that $v$ plays action $a$ at phase $i+1$. In this case, $v$ plays only this action no more than $2\lceil 2^{i+2}/A_i\rceil \leq 3\lceil 2^{i+2}/A_i\rceil$. Notice that the denominator has $A_i$ and not $A_{i+1}$, since $v$ didn't eliminate any action in this round yet.

Since $j \geq \log_2(A)$ we get $2^{j+1}/A_j \geq A/A_j \geq 1$. Then we get $\lceil 2^{j+1}/A_j\rceil \leq 2 \cdot 2^{j+1}/A_j \leq 2 \cdot 2^{j+1}/A_i$. Hence, the agents play this action in phases $j \geq \log_2(A)$ no more than

$$3(\sum_{j=1}^{i} 2\frac{2^{j+1}}{A_i} + 2\frac{2^{i+2}}{A_i}) \leq 3(\frac{4}{A_i}2^{i+1} + 4\frac{2^{i+1}}{A_i}) \leq 48\frac{2^i}{A_i}.$$

$\square$

**Lemma 39.** *Let $i$ be the last phase in which action $a$ is active, and $i \geq \log_2(A) + 1$. The regret that action $a$ contributes for phases $j \geq \log_2(A)$ is bounded by*

$$64\sqrt{2\log(3mTA)}\frac{1}{A_i}\sqrt{\frac{2^i \cdot A_{i-1}}{\min\{2^i, m\}}}. \tag{17}$$

*Proof.* Let $i$ be the last phase in which action $a$ is active.

From Lemma 38, the regret is bounded by

$$\Delta_a \cdot 48\frac{2^i}{A_i}.$$

From Lemma 37,

$$\Delta_a \leq 4\sqrt{2\iota \cdot \frac{A_{i-1}}{2^i \cdot \min\{2^i, m\}}}.$$

we get

$$\Delta_a \cdot 16\frac{2^i}{A_i} \cdot 4\sqrt{2\iota \cdot \frac{A_{i-1}}{2^i \cdot \min\{2^i, m\}}} = \Delta_a \cdot 64\sqrt{2\iota}\frac{1}{A_i}\sqrt{\frac{2^i \cdot A_{i-1}}{\min\{2^i, m\}}}.$$

Since $\Delta_a \leq 1$, we get the full result. $\square$

**Lemma 40.** *Let $i \geq \log_2(A) + 1$ be a phase. The regret from phases $j \geq \log_2(A)$ of all actions that $i$ was their last phase is bounded by*

$$64\sqrt{2\log(3mTA)}\frac{A_i - A_{i+1}}{A_i}\sqrt{\frac{2^i \cdot A_{i-1}}{\min\{2^i, m\}}}. \tag{18}$$

*Proof.* From Lemma 39, the regret from phases $j \geq \log_2(A)$ of an action that $i$ was its last active phase is bounded by

$$64\sqrt{2\iota}\frac{1}{A_i}\sqrt{\frac{2^i \cdot A_{i-1}}{\min\{2^i, m\}}}.$$

There are $A_i - A_{i+1}$ such actions. Hence, we get the results. $\qquad\square$

**Definition 31.** *Let us denote the regret bound of an action from phases $j \geq \log_2(A)$, Equation (17), with $b_j$. I.e.,*

$$b_j := 64\sqrt{2\log(3mTA)}\frac{1}{A_j}\sqrt{\frac{2^j \cdot A_{j-1}}{\min\{2^j, m\}}}.$$

**Definition 32.** *Denote the expression in Equation (18) (the regret bound of all actions that their last active phase is $j$) with $B_j$. I.e.,*

$$B_j := (A_j - A_{j-1})b_j = 64\sqrt{2\log(3mTA)}\frac{A_j - A_{j+1}}{A_j}\sqrt{\frac{2^j \cdot A_{j-1}}{\min\{2^j, m\}}}. \tag{19}$$

The following lemma captures the core insight of our amortized analysis. It bounds the regret in high-ratio phase with low-ratio phase. Formally,

**Lemma 41.** *Let $j \geq \log_2(A) + 1$ be a phase such that $A_{j-1} > 2A_j$. Let $i$ be the first phase such that*

(i) $i \geq \log_2(A)$.

(ii) *The phase $i$ is the first such that there exists a sequence $i$ to $j$ such that $A_i > 2A_{i+1} > \cdots > 2^{j-i-1}A_{j-1} > 2^{j-i}A_j$.*

*Then,*

$$B_j \leq 4B_i.$$

*Proof.* First we prove that $i \neq j$. The existence of $i$ is immediate, from its definition. It is either $i$ is the closest phase to $j$ that has $A_{i-1}/A_i \leq 2$ (breaking the sequence, while $i \leq j$) or that $i = \lceil\log_2(A)\rceil$. Assume by contradiction that $i = j$. We will see that $j - 1$ holds these two conditions. First, $j \geq 1 + \log_2(A)$, then $j - 1 \geq \log_2(A)$. Second, $A_{j-1} > 2A_j$ by the definition of $j$, a contradiction to the condition that says that $i$ is the first phase to start this sequence. Therefore, $i = j$.

Since $i < i + 1 \leq j$, we get that $A_i/A_{i+1} > 2$. Then $A_{i+1}/A_i \leq 1/2$, and $1 - A_{i+1}/A_i \geq 1/2$. We will use this inequality later in the proof.

The ratio $B_j/B_i$ is

$$\frac{\frac{A_j - A_{j+1}}{A_j} \cdot \sqrt{\frac{2^j A_{j-1}}{\min\{2^j, m\}}}}{\frac{A_i - A_{i+1}}{A_i} \cdot \sqrt{\frac{2^i A_{i-1}}{\min\{2^i, m\}}}} = \frac{1 - \frac{A_{j+1}}{A_j}}{1 - \frac{A_{i+1}}{A_i}}\sqrt{2^{j-i}\frac{A_{j-1}\min\{2^i, m\}}{A_{i-1}\min\{2^j, m\}}}$$

$$\leq \frac{1 - \frac{A_{j+1}}{A_j}}{1 - \frac{A_{i+1}}{A_i}}\sqrt{2^{j-i}\frac{A_{j-1}}{A_{i-1}}} \tag{a}$$

$$\leq 2\sqrt{2^{j-i}\frac{A_{j-1}}{A_{i-1}}} \tag{b}$$

$$\leq 2\sqrt{2\frac{A_i}{A_{i-1}}} \leq 4 \tag{c}$$

Where the first inequality (a) is since $\frac{\min\{2^i, m\}}{\min\{2^j, m\}} \leq 1$ as $i < j$ and $\min\{2^x, m\}$ is increasing with $x$.

The next inequality (b) used the facts that $1 - \frac{A_{i+1}}{A_i} \geq \frac{1}{2}$, $1 - \frac{A_{j+1}}{A_j} \leq 1$, and $2^{j-i-1}A_{j-1} \leq A_i$. We showed earlier that $1 - \frac{A_{i+1}}{A_i} \geq \frac{1}{2}$. The inequality $1 - \frac{A_{j+1}}{A_j} \leq 1$ holds since $\frac{A_{j+1}}{A_j} \geq 0$ (even where $j$ is the global last phase, there we defined $A_{j+1} := 0$), and $2^{j-i-1}A_{j-1} \leq A_i$ holds because of the definition of $i$ and from the fact that $i \neq j$. The last inequality (c) uses the fact that $A_i \leq A_{i-1}$ by lemma 34. $\qquad\square$

The following lemma bounds the regret of all high ratio phases ($A_{i-1}/A_i > 2$), with other low ratio phases. Each phase that has low ratio between the previous and the current number of actions ($A_{i-1}/A_i \leq 2$) is paying on a phase with high ratio. But there might be a sequence of phases that has high ratio, so one phase with low ratio can't pay for the rest by multiplying just with constant. Since there are at most $\log_2(A)$ high ratio phases, we can bound the regret of all high ratio phases with low ratio phases. Formally we get,

**Lemma 42.** *Let us denote the set of phases with low ratio between the number of actions. Specifically, denote $\mathcal{I} := \{i \in \mathbb{N}^+ | i \geq \log_2(A), A_{i-1}/A_i \leq 2\} \cup \{\lceil \log_2(A) \rceil\}$. Then,*

$$\sum_{j \geq \log_2(A)} B_j \leq \sum_{i \in \mathcal{I}} (5\log_2(A) \cdot B_i),$$

*where*

$$B_j := (A_j - A_{j-1})b_j.$$

*Proof.* From Lemma 41, for each $j \geq \log_2(A) + 1$ such that $A_{j-1}/A_j > 2$, there exists $i \in \mathcal{I}$ such that $B_j \leq 4B_i$. The number of remaining actions decreases by more than a half each round. We have $A \geq A_i > 2^{j-i}A_j$, then $j - i < \log_2(A)$. Let $i, i+1, \ldots, i+j$ be a sequence as defined in Lemma 41. I.e., $i \geq \log_2(A)$. And the phase $i$ is the first such that there exists a sequence $i$ to $j$ that holds $A_i > 2A_{i+1} > \cdots > 2^{j-i-1}A_{j-1} > 2^{j-i}A_j$. Then

$$\sum_{k=i}^{j} B_k \leq B_i + \sum_{k=i+1}^{j} 4B_i \leq B_i + \log_2(A)4B_i \leq 5\log_2(A)B_i.$$

Hence, each such sum of $\sum_{k=i}^{j} B_k$ a sequence $A_i > 2A_{i+1} > \cdots > 2A_j$ is bounded by the $5\log_2(A)B_i$, where $i$ is the beginning of the sequence. We can break the sum $\sum_{j \geq \log_2(A)} B_j$ into sums of such sequences, and the results follows. $\qquad\square$

**Lemma 43.** *All the actions for which their last active phase $i$ is smaller or equal to $\log_2(A)$ contribute to the regret no more than $24 \cdot A$.*

*Proof.* In the phase $j$ there are $3 \cdot 2^{j+1}$ timesteps. An action that its last active phase is $i$ can be played until the stage $i + 1$, included. So until the phase $i + 1$ the number of timesteps is at most

$$3\sum_{j=1}^{i+1} 2^{j+1} = 6\sum_{j=1}^{i+1} 2^j = 6 \cdot (2^{i+2} - 1).$$

Therefore, for actions that were eliminated at phases smaller than $\log_2(A)$ we get that the regret that is contributed from all these actions is bounded by

$$6 \cdot (2^{\log_2(A)+2} - 1) \leq 6 \cdot 2^{\log_2(A)+2} = 24 \cdot A.$$

$\qquad\square$

**Lemma 44.** *Let us denote with $\mathcal{I}^+$ the set of low-ratio phases, without the phase $\lceil \log_2(A) \rceil$. Specifically, denote $\mathcal{I}^+ := \{i \in \mathbb{N}^+ | i \geq \log_2(A) + 1, A_i/A_{i-1} \leq 2\}$. Let $\mathcal{A}_{\leq 2}$ be the set of action such their last active phase is in $\mathcal{I}^+$. Then we get*

$$\sum_{a \in \mathcal{A}_{\leq 2}} b_{i_a} \leq \sum_{a \in \mathcal{A}} \left(\frac{1024\iota}{\Delta_a \cdot m}\right) + 157A\log(3mTA).$$

*Proof.* We first focus on actions $a$ for which $2^{i_a} < m$. We get that

$$64\sqrt{2\iota}\frac{1}{A_{i_a}}\sqrt{\frac{2^{i_a} \cdot A_{i_a-1}}{\min\{2^{i_a}, m\}}} = 64\sqrt{2\iota}\frac{\sqrt{A_{i_a-1}}}{A_{i_a}}$$

Let us order all the actions in a weak linear order of which they were eliminated. For the simplicity of the notation, assume it is their order in $A$ (and $\mathcal{A} = [A]$). Let us define two vectors of length $A$ each.

$$u = \left(\sqrt{\frac{A_{i_a-1}}{A_{i_a}}}\right)_{a\in[A]},$$

and

$$v = \left(\sqrt{\frac{1}{A_{i_a}}}\right)_{a\in[A]}.$$

With these vector notations we get

$$\sum_{a\in\mathcal{A}_{\leq 2}, 2^{i_a}<m} 64\sqrt{2\iota}\frac{\sqrt{A_{i_a-1}}}{A_{i_a}} \leq \sum_{a\in[A], 2^{i_a}<m} 64\sqrt{2\iota}\frac{\sqrt{A_{i_a-1}}}{A_{i_a}}$$
$$\leq 64\sqrt{2\iota}|\langle u, v\rangle|.$$

For $u$ we get

$$\|u\|_2^2 = \sum_a \frac{A_{i_a-1}}{A_{i_a}} \leq \sum_a A_{i_a-1} \leq A^2.$$

For $v$ we get

$$\|v\|_2^2 = \sum_j \frac{1}{A_{i_a}} \leq 1 + \log A \leq 3\log A.$$

where the first inequality is since

$$\sum_{a\in\mathcal{A}} \frac{1}{A_{i_a}} \leq \sum_{i=1}^A \frac{1}{i} = 1 + \sum_{i=2}^A \frac{1}{i} \leq 1 + \int_1^A \frac{1}{x}dx = 1 + \log A.$$

The first inequality holds since if phase $i$ was the last active phase for $x$ actions, then $x/A_i \leq 1/(A_i) + 1/(A_i - 1) + \cdots + 1/(A_i - (x-1))$. The last inequality is since we can assume $A \geq 2$.

From Cauchy-Schwartz we get

$$|\langle u, v\rangle| \leq \|u\|_2 \cdot \|v\|_2 \leq \sqrt{A^2 \cdot 3\log A} = A\sqrt{3\log A}.$$

For all actions which their last active phase is smaller than $m$ we have that their contribution to the regret is no more than

$$64\sqrt{2\iota} \cdot A\sqrt{3\log A} \leq 157A\log(3mTA).$$

We now focus on actions $a$ for which $2^{i_a} \geq m$.

$$b_{i_a} := 64\sqrt{2\log(3mTA)}\frac{1}{A_{i_a}}\sqrt{\frac{2^{i_a} \cdot A_{i_a-1}}{\min\{2^{i_a}, m\}}} = 64\sqrt{2\iota}\frac{1}{A_{i_a}}\sqrt{\frac{2^{i_a} \cdot A_{i_a-1}}{m}}$$

Assume $2^i \geq m$. Then, from Lemma 37

$$\Delta_a \leq 4\sqrt{2\iota \cdot \frac{A_{i-1}}{2^i \cdot m}}.$$

Square everything and we will get

$$\Delta_a^2 \leq 32\iota \cdot \frac{A_{i-1}}{2^i \cdot m},$$

$$2^i \le 32\iota \cdot \frac{A_{i-1}}{\Delta_a^2 \cdot m}.$$

Substituting $2^{i_a}$ with the previous term and we get,

$$64\sqrt{2\iota}\frac{1}{A_{i_a}}\sqrt{\frac{2^{i_a} \cdot A_{i_a-1}}{m}} \le 64\sqrt{2\iota}\frac{1}{A_{i_a}}\sqrt{\frac{32\iota \cdot A_{i_a-1}}{\Delta_a^2 \cdot m}\frac{A_{i_a-1}}{m}}$$

$$= 64\iota\sqrt{64}\frac{A_{i_a-1}}{A_{i_a}}\frac{1}{\Delta_a \cdot m}$$

$$\le 64\iota\sqrt{64} \cdot 2\frac{1}{\Delta_a \cdot m}$$

$$= \frac{1024\log(3mTA)}{\Delta_a \cdot m}.$$

where the last inequality is since $a \in \mathcal{A}_{\le 2}$. Overall we get

$$\sum_{a \in \mathcal{A}_{\le 2}} 64\sqrt{2\iota}\frac{1}{A_{i_a}}\sqrt{\frac{2^{i_a} \cdot A_{i_a-1}}{\min\{2^{i_a}, m\}}} \le \sum_{a \in \mathcal{A}_{\le 2}} (\frac{1024\iota}{\Delta_a \cdot m}) + 157A\iota$$

$$\le \sum_{a \in \mathcal{A}} (\frac{1024\log(3mTA)}{\Delta_a \cdot m}) + 157A\log(3mTA).$$

$\square$

**Proof of Theorem 5.** We can bound the overall regret in the following way. First, the regret of the first $\lfloor \log_2(A) \rfloor$ phases is bounded by $24A$, from Lemma 43. For actions that were eliminated at phases $i \ge \log_2(A)$, from Lemma 40 the regret from phases $j \ge \log_2(A)$ of all these actions is bounded by

$$\sum_{i \ge \log_2(A)} \left( 64\sqrt{2\log(3mTA)}\frac{A_i - A_{i+1}}{A_i}\sqrt{\frac{2^i \cdot A_{i-1}}{\min\{2^i, m\}}} \right).$$

Notice that this is is exactly $B_i$ from Equation (19). From Lemma 42, we get a bound for this part of the sum,

$$\sum_{i \ge \log_2(A)} 64\sqrt{2\log(3mTA)}\frac{A_i - A_{i+1}}{A_i}\sqrt{\frac{2^i \cdot A_{i-1}}{\min\{2^i, m\}}} \le \sum_{i \in \mathcal{I}} 5\log_2(A) \cdot B_i.$$

Converting the analysis into actions-based analysis and we get

$$\sum_{i \in \mathcal{I}} 5\log_2(A) \cdot B_i = 5\log_2(A) \left( \sum_{a \in \mathcal{A}_{\le 2}} 64\sqrt{2\iota}\frac{1}{A_i}\sqrt{\frac{2^i \cdot A_{i-1}}{\min\{2^i, m\}}} + 64\sqrt{2\iota}\frac{1}{A_{\lceil\log_2(A)\rceil}}\sqrt{\frac{2^{\lceil\log_2(A)\rceil} \cdot A_{\lceil\log_2(A)\rceil-1}}{\min\{2^{\lceil\log_2(A)\rceil}, m\}}} \right),$$

Where $\mathcal{A}_{\le 2}$ is the set of action in that their last active phase is in $\{i \in \mathbb{N}^+ | i \ge \log_2(A) + 1, A_i/A_{i-1} \le 2\}$.

Focusing on the part that belongs the phase $i = \lceil\log_2(A)\rceil$ we get

$$64\sqrt{2\iota}\frac{1}{A_i}\sqrt{\frac{2^{\lceil\log_2(A)\rceil} \cdot A_{\lceil\log_2(A)\rceil-1}}{\min\{2^{\lceil\log_2(A)\rceil}, m\}}} \le 64\sqrt{2\iota}\frac{1}{1}\sqrt{\frac{2A \cdot A}{\min\{A, m\}}}$$

$$\le 64\sqrt{2\iota} \cdot \sqrt{2}A$$

$$= 256A\sqrt{\iota}$$

$$\le 256A\log(3mTA).$$

For the rest of the actions, from Lemma 44 we know that

$$\sum_{a \in \mathcal{A}_{\leq 2}} 64 \sqrt{2 \log(3mTA)} \frac{1}{A_{i_a}} \sqrt{\frac{2^{i_a} \cdot A_{i_a-1}}{\min\{2^{i_a}, m\}}} \leq \sum_{a \in \mathcal{A}} \left(\frac{1024 \log(3mTA)}{\Delta_a \cdot m}\right) + 157A \log(3mTA).$$

Multiplying everything with $5 \log_2(A)$ we get

$$5 \log_2(A) \left(\sum_{a \in \mathcal{A}} \left(\frac{1024 \log(3mTA)}{\Delta_a \cdot m}\right) + (157 + 256)A \log(3mTA)\right)$$
$$= \sum_{a \in \mathcal{A}} \left(\frac{5120 \log_2(A) \log(3mTA)}{\Delta_a \cdot m}\right) + 2065A \log_2(A) \log(3mTA).$$

The complementary event to the good event adds no more than $1$ to the regret. The overall regret is bounded by

$$\mathfrak{R}_T \leq \sum_{a \in \mathcal{A}} \left(\frac{5120 \log_2(A) \log(3mTA)}{\Delta_a \cdot m}\right) + 2065A \log_2(A) \log(3mTA) + 24A + 1.$$

$\square$

**Lemma 45.** *When all agents play* `Coop-SE-Comm-Cost` *(Algorithm 13) with spanning tree $\mathcal{T}$ the number of messages each agent $v$ sends is no more than*

$$\lceil \log_2(T/6) \rceil \cdot \deg_{\mathcal{T}}(v),$$

*where $\deg_{\mathcal{T}}(v)$ is the degree of $v$ in the spanning tree graph (the number of neighbors).*

*Proof.* In each phase each agent sends messages in no more than 2 timesteps: One message of the collected information from the boundary vertices (`cluster-boundary`) of the cluster. This message is sent to the parent agent. Note that the agent waits until all messages from her descendants arrive before she sends the message to the parent agent. This adds 1 message for each phase. The other timesteps the agent sends messages is when she tells her descendants the set of active actions. This adds $\deg_{\mathcal{T}}(v) - 1$ messages for each phase. There are $\lceil \log_2(T/6) \rceil$ phases, and the result follows. $\square$

# I Auxiliary Lemmas

**Lemma 46** (Lemma F.4 in [10]). *Let $\{X_t\}_{t=1}^T$ be a sequence of Bernoulli random variables and a filtration $\mathcal{F}_1 \subseteq \mathcal{F}_2 \subseteq ...\mathcal{F}_T$ with $\mathbb{P}(X_t = 1 \mid \mathcal{F}_t) = P_t$, $P_t$ is $\mathcal{F}_t$-measurable and $X_t$ is $\mathcal{F}_{t+1}$-measurable. Then, for all $t \in [T]$ simultaneously, with probability $1 - \delta$,*

$$\sum_{k=1}^t X_k \geq \frac{1}{2} \sum_{k=1}^t P_k - \log \frac{1}{\delta}.$$

**Lemma 47** (Consequence of Freedman's Inequality, e.g., Lemma E.2 in [9]). *Let $\{X_t\}_{t \geq 1}$ be a sequence of random variables, supported in $[0, R]$, and adapted to a filtration $\mathcal{F}_1 \subseteq \mathcal{F}_2 \subseteq ...\mathcal{F}_T$. For any $T$, with probability $1 - \delta$,*

$$\sum_{t=1}^T X_t \leq 2\mathbb{E}[X_t \mid \mathcal{F}_t] + 4R \log \frac{1}{\delta}.$$

## J Detailed Algorithms

---

**Algorithm 5** Elimination Step (`Elim-Step`)

---

1: **Input:** active actions $\mathcal{A}$, number of samples $n(a)$ for each active action $a$, empirical mean for every active action $\hat{\mu}(a)$.
2: $E = \emptyset$
3: **for** $a \in \mathcal{A}$ **do**
4:

$$\lambda(a) = \sqrt{\frac{2\iota}{n(a) \vee 1}}, \quad UCB(a) = \hat{\mu}(a) + \lambda(a), \quad LCB(a) = \hat{\mu}(a) - \lambda(a) \qquad (20)$$

where $\iota := \log(3mTA)$.
5: **end for**
6: **for** $a \in \mathcal{A}$ **do**
7:    **if** exists $a'$ with $UCB(a) < LCB(a')$ **then**
8:       $E = E \cup \{a\}$
9:    **end if**
10: **end for**
11: Return $E$

---

**Algorithm 6** Cooperative Successive Elimination (`Coop-SE`) - detailed

---

1: **Input:** number of rounds $T$, neighbor agents $N$, number of actions $A$, ID of current agent $v$.
2: **Initialization:** $t \leftarrow 1$; Set of *active* actions $\mathcal{A} = \mathbb{A}$; $R_t(a) = 0, n_t(a) = 0$ for every action $a$; $M_{\texttt{in}} = \emptyset$; $M_{\texttt{updates}} = \emptyset$; $M_{\texttt{sent}} = \emptyset$; $M_{\texttt{seen}} = \emptyset$;
3: **for** $t = 1, ..., T$ **do**
4:   **for** $event \in M_{\texttt{updates}}$ **do**
5:     **if** $event \notin M_{\texttt{seen}}$ **then**
6:       $M_{\texttt{seen}} = M_{\texttt{seen}} \cup event$
7:       **if** $event$ is `elim`-event **then**
8:         $\mathcal{A} = \mathcal{A} \setminus event_a$
9:       **else if** $event_a \in \mathcal{A}$ **then**
10:         $n_t(a) = n_t(a) + 1, R_t(a) = R_t(a) + event_r$
11:       **end if**
12:     **end if**
13:   **end for**
14:   $E = \texttt{Elim-Step}(\mathcal{A}, n_t, \hat{\mu}_t = R_t/n_t), \mathcal{A} = \mathcal{A} \setminus E$
15:   Choose action $a_t$ in round robin from $\mathcal{A}$, and get reward $r_t(a_t)$
16:   // Send and receive messages
17:   $M_{\texttt{me}} = \{(\texttt{rwd}, t, v, a_t, r_t(a_t)\} \cup \{(\texttt{elim}, v, a) | \exists a \in E\}, M_t^v = (M_{\texttt{me}} \cup M_{\texttt{in}}) \setminus M_{\texttt{sent}}$
18:   Send message $M_t^v$ to all neighbors, receive messages $M_t^{v'}$ from each neighbor $v' \in N$
19:   $M_{\texttt{sent}} = M_{\texttt{sent}} \cup M_t^v, M_{\texttt{updates}} = M_{\texttt{me}} \cup_{v' \in N} M_t^{v'}, M_{\texttt{in}} = M_{\texttt{in}} \cup_{v' \in N} M_t^{v'}$
20: **end for**

---

---
**Algorithm 7** Successive Elimination with Suspended Act for agent $v$ (Sus-Act)
---

1: **Input:** number of rounds $T$, number of actions $A$, diameter of the graph $D$, number of agents $m$, neighbor agents $N$, factor for the confidence bound $L$.

2: **Initialization:** $t \leftarrow 1$; set of *active* actions $\mathcal{A} \leftarrow \mathbb{A}$; $M_{\text{sent}} = \emptyset$; Set incoming messages $M_{\text{in}} = \emptyset$; Set of seen messages $M_{\text{seen}} = \emptyset$.

3: **while** $t < T$ **do**

4:     Calculate suspended counts and empirical means for each active action from the $M_{\text{seen}}$ messages

$$n_t(a) = \sum_{\tau=1}^{t-D} \sum_v \mathbb{I}\{a_\tau^v = a\}; \quad \hat{\mu}_t(a) = \frac{1}{n_t(a) \vee 1} \sum_{\tau=1}^{t-D} \sum_v r_\tau^v(a)\mathbb{I}\{a_\tau^v = a\}$$

5:     $\mathcal{A} = \mathcal{A} \setminus \text{Elim-Step}(\mathcal{A}, \{n_t(a)|a \in \mathcal{A}\}, \{\hat{\mu}_t(a)|a \in \mathcal{A}\})$

6:     Choose one action $a_t \in \mathcal{A}$ in round robin and receive $r_{a_t}$

7:     Let $M_{\text{me}} = \{(RWD, t, v, a_t, r_{a_t})\}$

8:     Let $M_t^v = (M_{\text{me}} \cup M_{\text{in}}) \setminus M_{\text{sent}}$

9:     Send message $M_t^v$ to all neighbors

10:     $M_{\text{sent}} = M_{\text{sent}} \cup M_t^v$

11:     Receive messages $M_t^{v'}$ from each neighbor $v'$

12:     Set incoming messages $M_{\text{in}} = M_{\text{in}} \cup \{M_t^{v'} \mid v' \text{ is a neighbor of } v\}$

13:     Set seen messages $M_{\text{seen}} = M_{\text{seen}} \cup M_{\text{in}} \cup M_{\text{me}}$

14:     $t = t + 1$

15: **end while**

---

---
**Algorithm 8** Update Step Tree - Update the counters with the received information and prepare them for sending in a tree graph (Update-Tree-Step)
---

1: **Input:** Neighbor agents $N$; Set of active actions $\mathcal{A}$; $M_{\text{updates}}$.

2: **for** $a \in \mathcal{A}$ **do**

3:     $n(a) = 0, R(a) = 0$

4:     **for** $u \in N$ **do**

5:         $\mathbf{N}_a^u = 0, \mathbf{R}_a^u = 0$

6:     **end for**

7: **end for**

8: **for** $event \in M_{\text{updates}}$ **do**

9:     **if** $event_a \in \mathcal{A}$ & $event$ is rwdMany **then**

10:         // $event = (\text{rwdMany}, id, a, r, n)$.

11:         $n(a) = n(a) + event_n, R(a) = R(a) + event_r$

12:         **for** $u \in N$ **do**

13:            **if** $event_{id} \neq u$ **then**

14:              $\mathbf{N}_a^u = \mathbf{N}_a^u + event_n; \mathbf{R}_a^u = \mathbf{R}_a^u + event_r$

15:            **end if**

16:         **end for**

17:     **end if**

18: **end for**

19: // Return the self counters and the values to send.

20: Return $n(a), R(a)$ for each $a \in \mathcal{A}$; Return $\mathbf{N}_a^u, \mathbf{R}_a^u$ for each $a \in \mathcal{A}$ and for each $u \in N$.

---

**Algorithm 9** Cooperative Successive Elimination with Restricted Communication (`Coop-SE-Restricted`)

---

1: **Input:** number of rounds $T$, neighbor agents $N$, number of actions $A$, id of current agent $v$, a spanning tree, $\mathcal{T}$, of the communication tree $\mathcal{G}$ (identical to all agents).

2: **Initialization:** $t \leftarrow 1$; Set of *active* actions $\mathcal{A} = \mathbb{A}$; $R_t(a) = 0, n_t(a) = 0$ for every action $a$; $M_{\texttt{in}} = \emptyset$; $M_{\texttt{updates}} = \emptyset$; $M_{\texttt{sent}} = \emptyset$;

3: Set $N$ to be the agent's neighbors in $\mathcal{T}$.

4: **for** $t = 1, ..., T$ **do**

5:     $E_{\texttt{received}} = \{event_a | \exists event \in M_{\texttt{updates}}, event$ is `elim-event`$\}$

6:     $\mathcal{A} = \mathcal{A} \setminus E_{\texttt{received}}$

7:     $n_t, R_t, \mathbf{N}, \mathbf{R} = \texttt{Update-Tree-Step}(N, \mathcal{A}, M_{\texttt{updates}})$

8:     $M_{\texttt{updates}} = \emptyset$

9:     $E = \texttt{Elim-Step}(\mathcal{A}, n_t, R_t/n_t)$

10:     $\mathcal{A} = \mathcal{A} \setminus E$

11:     Choose action $a_t$ uniformly from $\mathcal{A}$, and get reward $r_t(a_t)$

12:     $n_t(a_t) = n_t(a_t) + 1$, $R_t(a_t) = R_t(a_t) + r_t(a_t)$

13:     **for** $u \in N'$ **do**

14:         $\mathbf{N}_{a_t}^u = \mathbf{N}_{a_t}^u + 1$, $\mathbf{R}_{a_t}^u = \mathbf{R}_{a_t}^u + \mathbf{R}_t(a_t)$

15:     **end for**

16:     **for** $u \in N'$ **do**

17:         $M_{\texttt{elim}}(u) = \{(\texttt{elim}, v, a) | \exists a \in E\} \cup \{(\texttt{elim}, v, event_a) | \exists event \in E_{\texttt{received}}, event_{id} \neq u\}$

18:         $M_{\texttt{rwd}}(u) = \{(\texttt{rwdMany}, v, a, \mathbf{R}_a^u, \mathbf{N}_a^u) | a \in \mathcal{A}\}$

19:         $M_t^v(u) = M_{\texttt{elim}}(u) \cup M_{\texttt{rwd}}(u)$

20:         Send $M_t^v(u)$ and receive $M_t^u(v)$

21:         $M_{\texttt{updates}} = M_{\texttt{updates}} \cup M_t^u(v)$

22:     **end for**

23: **end for**

---

**Algorithm 10** Send one action - CONGEST (`Send-One-Action`)

---

1: **Input:** Neighbor agent $u$; Received eliminations-events $E_{\texttt{received}}$; New eliminations $E$; Action $a'$; Counters to send $\mathbf{N}_{a'}^u$; Rewards to send $\mathbf{R}_{a'}^u$.

2: **if** $\exists a' \in E$ **then**

3:     send $(\texttt{elim}, v, a')$ to $u$; return.

4: **else if** $\exists event \in E_{\texttt{received}}, event_{id} \neq u, event_a = a'$ **then**

5:     send $(\texttt{elim}, v, a')$ to $u$; return.

6: **else**

7:     send $(\texttt{rwdMany}, v, a', \mathbf{R}_{a'}^u, \mathbf{N}_{a'}^u)$ to $u$; return.

8: **end if**

---

**Algorithm 11** Cooperative Successive Elimination CONGEST (`Coop-SE-CONGEST`) - detailed
___

1: **Input:** number of rounds $T$, neighbor agents $N$, number of actions $A$, id of current agent $v$, a spanning tree, $\mathcal{T}$, of the connected communication graph $\mathcal{G}$, with a root agent $w$ (same node for all agents).
2: **Initialization:** $t \leftarrow 1$; Set of *active* actions $\mathcal{A} = \mathbb{A}$; $R_0(a) = 0, n_0(a) = 0$ for every action $a$; $M_{\texttt{updates}} = \emptyset$;
3: Calculate the distance between the root $w$ and the current agent $v$, $d := d_{\mathcal{T}}(v, w)$, where $d_{\mathcal{T}}$ is the distance in the tree.
4: Set $N$ to be the agent's neighbors in $\mathcal{T}$.
5: Set $N' \subseteq N$ to be the set of $v$'s children on the tree $\mathcal{T}$ rooted at $w$.
6: Set $\tilde{u} \in N$ to be $v$'s parant in the tree $\mathcal{T}$ rooted at $w$. I.e., $\{\tilde{u}\} = N \setminus N'$. Notice that $\tilde{u}$ exists only if $v$ is not the root.
7: **for** $t = 1, ..., T$ **do**
8:     $E_{\texttt{received}} = \{event_a | \exists event \in M_{\texttt{updates}}, event \text{ is } \texttt{elim-event}\}$
9:     $n, R, \mathbf{N}, \mathbf{R} = \texttt{Update-Tree-Step}(N, \mathcal{A}, M_{\texttt{updates}})$
10:     $n_t = n_{t-1} + n, R_t = R_{t-1} + R$
11:     $E = \texttt{Elim-Step}(\mathcal{A}, n_t, \hat{\mu}_t = R_t/n_t)$
12:     $\mathcal{A} = \mathcal{A} \setminus E$
13:     Choose action $a_t$ in round-robin from $\mathcal{A}$, and get reward $r_t(a_t)$
14:     $n_t(a_t) = n_t(a_t) + 1, R_t(a_t) = R_t(a_t) + r_t(a_t)$
15:     **for** $u \in N$ **do**
16:       $\mathbf{N}_t^u(a_t) = \mathbf{N}_t^u(a_t) + 1, \mathbf{R}_t^u(a_t) = \mathbf{R}_t^u(a_t) + r_t(a_t)$
17:     **end for**
18:     Choose the action $a'$, the action to send to $v$'s children: $a' \equiv t - d \pmod{A}$
19:     Choose the action $\tilde{a}$, the action to send to $v$'s parent: $\tilde{a} \equiv t + d \pmod{A}$
20:     **for** $u \in N'$ **do**
21:       // Send the messages outward from the root.
22:       $\texttt{Send-One-Action}(u, E_{\texttt{received}}, E, a', \mathbf{N}_{a'}^u, \mathbf{R}_{a'}^u)$
23:     **end for**
24:     $\texttt{Send-One-Action}(\tilde{u}, E_{\texttt{received}}, E, \tilde{a}, \mathbf{N}_{\tilde{a}}^u, \mathbf{R}_{\tilde{a}}^u)$ // Send the messages toward the root.
25:     $M_{\texttt{updates}} = \emptyset$
26:     **for** $u \in N$ **do**
27:       Receive $M_t^u(v)$
28:       $M_{\texttt{updates}} = M_{\texttt{updates}} \cup M_t^u(v)$
29:     **end for**
30: **end for**
___

**Algorithm 12** Play round robin and increment the timestep counter (`Play-Action-Round-Robin`)
___

1: **if** $t = T + 1$ **then**
2:     Terminate the program
3: **end if**
4: Play action $a_t$ in round robin from the set of active actions, get reward $r_t(a_t)$
5: $t = t + 1$
6: **return** $a_t, r_t(a_t)$
___

**Algorithm 13** Cooperative Successive Elimination with Communication Cost
(`Coop-SE-Comm-Cost`)

---

1: Input: A spanning tree $\mathcal{T}$ (same tree for all vertices); Children $C$, and parent $p$ in the tree; Actions $A$.

2: Initialize: $t = 1$; $R^v(a) = 0, n^v(a) = 0$ for each $a \in A$; Set of active actions $\mathcal{A} = A$; `forward=false`.

3: **for** phase $i = 0, \ldots, \lceil \log_2(T/6) \rceil - 1$ **do**

4:     Based on $\mathcal{T}$, compute for this phase $i$ if this agent $v$ is `cluster-root`, `cluster-boundary` or none of these

5:

6:     **Sub-Phase 1: Gather and Aggregate Information**

7:     **for** $k = 1, \ldots, 2^{i+1}$ **do**

8:       $a_t, r_t = $ `Play-Action-Round-Robin`     *// Play one action from the current active actions*

9:       Update rewards and counters $R^v(a_t) \leftarrow R^v(a_t) + r_t, n^v(a_t) \leftarrow n^v(a_t) + 1$

10:       **if** $k = 1$ **and** $v$ is `cluster-boundary` for $i$ **then**

11:         Send to parent agent:

12:           1. Set of active actions: $\{\text{is-active}(a) = (a \in \mathcal{A}) \mid a \in A\}$

13:           2. Rewards and counts for each action: $\{R^v(a), n^v(a) \mid a \in A\}$

14:       **else if** received messages from all children **then**

15:         Aggregate children's rewards with local rewards: $R^v(a) \leftarrow R^v(a) + \sum_{u \in C} R^u(a)$

16:         Aggregate eliminations, set `is-active(a)=false` if at least one message contains an elimination about this action

17:         `forward=true`

18:       **else if** `forward=true` **then**

19:         `forward=false`

20:         Forward aggregated data, $\{(\text{is-active}(a), R^v(a), n^v(a)) \mid a \in A\}$, to the parent agent

21:         Set $R^v(a) = 0, n^v(a) = 0$ for each $a \in A$

22:       **end if**

23:     **end for**

24:

25:     **Sub-Phase 2: Synchronize Active Actions**

26:     **for** $k = 1, \ldots, 2^{i+1}$ **do**

27:       $a_t, r_t = $ `Play-Action-Round-Robin`

28:       Update rewards and counters $R^v(a_t) \leftarrow R^v(a_t) + r_t, n^v(a_t) \leftarrow n^v(a_t) + 1$

29:       **if** $k = 1$ **and** $v$ is `cluster-root` for $i$ **then**

30:         $\mathcal{A} \leftarrow \mathcal{A} \setminus \{a \in A \mid \text{is-active}(a)\text{=false}\}$
          Eliminate actions based on aggregated eliminations

31:         $\mathcal{A} \leftarrow \mathcal{A} \setminus \text{Elim-Step}(\mathcal{A}, n^v, \hat{\mu}_t = R^v/n^v)$
          Eliminate actions based on aggregated rewards

32:         Send $\mathcal{A}$, the new active action set, to descendants

33:       **else if** received active-actions message **then**

34:         Update local active actions set

35:         **if** $v$ is not `cluster-boundary` **then**

36:           Forward message to descendants in the next timestep

37:         **end if**

38:       **end if**

39:     **end for**

40:

41:     **Sub-Phase 3: Execute with Updated Actions**

42:     **for** $k = 1, \ldots, 2^{i+1}$ **do**

43:       $a_t, r_t = $ `Play-Action-Round-Robin`

44:       Update rewards and counters $R^v(a_t) \leftarrow R^v(a_t) + r_t, n^v(a_t) \leftarrow n^v(a_t) + 1$

45:     **end for**

46: **end for**

---

