# OpenReview forum: "Individual Regret in Cooperative Stochastic Multi-Armed Bandits"
_NeurIPS.cc/2025/Conference — NeurIPS 2025 poster_

### Official Review · Reviewer_LDks · 2025-06-17

**Clarity:** 3
**Significance:** 2
**Originality:** 3
**Rating:** 4
**Confidence:** 3

**Summary:**

The paper considers the problem in which multiple agents (vertices of a graph) face the same stochastic bandit problem and can communicate over the edges of the graph. The authors give algorithms for this problem with strong individual regret guarantees. Importantly, these bounds are independent of the diameter of the graph. The first algorithm requires much information to be passed across each edge on each trial. However, the later algorithms require much less, and one algorithm only requires a logarithmic number of trials in which to communicate. The authors also give an almost matching lower bound on the individual regret.

**Questions:**

I have no questions for the authors as I feel I understand the results. Since there is no section for comments I will point out some typos here:

In Line 149 you say that the agent observes a reward but you don’t say that the reward is drawn from the SMAB distribution.

You have some small brackets in Theorem 2 which should be big brackets. Same for Line 194.

Typo in definition 10 - begging instead of beginning

**Ethical Concerns:**

["NO or VERY MINOR ethics concerns only"]

**Final Justification:**

Surprising result - I am on the fence between borderline and full accept although tend towards borderline.

**Limitations:**

No limitations noticed.

**Paper Formatting Concerns:**

None noticed.

**Quality:**

3

**Strengths And Weaknesses:**

I think that the results of this paper are quite remarkable. At first I didn’t believe that the results could be possible (especially the O(log(m)) message size results) but after thinking about it I now believe that they could be true. Unfortunately, I haven’t had the time to read through the proof and verify the results.

The lower bound is also a nice addition to the paper.

The results of the paper are clearly communicated.

Although this is not a weakness of the result itself, I feel that, in order to make the paper self contained, there should be some explanation of how SE works before we jump into the description of Coop-SE as the explanation of the latter assumes familiarity with the former.

---

> ### Author Rebuttal · Authors · 2025-07-30
>
> We thank the reviewer for their highly positive review.
> We're happy to address any questions the reviewer might have during the author-reviewer discussion. If we have resolved the reviewer's concerns, we would appreciate it if they could consider raising their scores.
>
>
> >"Although this is not a weakness of the result itself, I feel that, in order to make the paper self contained, there should be some explanation of how SE works before we jump into the description of Coop-SE as the explanation of the latter assumes familiarity with the former."
>
> We will certainly include this in the final version.
>
>
> # Questions
>
> >”In Line 149 you say that the agent observes a reward but you don’t say that the reward is drawn from the SMAB distribution.”
>
> >”You have some small brackets in Theorem 2 which should be big brackets. Same for Line 194.”
>
> >”Typo in definition 10 - begging instead of beginning”
>
> Thank you for the suggestions. We will address all of them in the final version.

---

> > ### Comment · Reviewer_LDks · 2025-08-04
> >
> > Thank you for the response. My score is very much on the fence between borderline and full accept - but since the other reviewers have issues with the $\sqrt{ln(T)}$ factor I will keep it as it is.

---

### Official Review · Reviewer_oAWf · 2025-06-24

**Clarity:** 3
**Significance:** 3
**Originality:** 3
**Rating:** 4
**Confidence:** 4

**Summary:**

This paper studies the cooperative stochastic multi-agent multi-armed bandits problem, where multiple agents are connected by a undirected connected graph. In this model, multiple agents aggregate information via communication to speedup the learning process. Importantly, this paper proposes a cooperative algorithm with a regret bound, independent of any graph parameter. Authors also provide a lower bound and discusses the regret achievability under two communication constraints, including communication-round constraint and communication-bit constraint.

**Questions:**

- Can authors discuss more why the current method does not work for UCB?

- I am a bit worried about the significance of individual regret in the stochastic multi-agent MAB setting, though it is indeed interesting in non-stationary setting, Many existing papers in stochastic setting use group regret (or equivalently average regret) as the metric, but those algorithms also have a low individual regret, and they just not give a formal proof on that. This is because in stochastic setting, the high-level behind speedup is to aggregate samples across agents, in light of the Hoeffding bound. Can author mention a paper in the stochastic setting that have low group regret, but individual regret is large?

**Ethical Concerns:**

["NO or VERY MINOR ethics concerns only"]

**Final Justification:**

While there is room for further improvement, I find the overall contributions good and lean toward a weak acceptance.

**Limitations:**

yes

**Quality:**

3

**Strengths And Weaknesses:**

Strength:

- This paper proposes a simple cooperative algorithm, ensuring a low individual regret for each agent. Importantly, the regret bound is independent of any graph parameter. This result implies that if the number of agents goes to infinity, the regret bound is entirely independent of arm gap.

- This paper presents a lower bound for the individual regret nearly matching the upper bound in the minimax form.

- Further, authors consider the constrained communication setting. For both logarithmic communication bit and logarithmic communication  rounds, two algorithms are proposed to achieve regret bounds, independent of graph parameter.


Weakness: I don't see any major weakness of this paper, and authors have acknowledged that there exists a gap between the upper and lower bounds. Though the design of algorithm is simple, ie., extending elimination based algorithms to multi-agent setup, it is indeed effective, and authors do a careful analysis to get a good regret bound.

The reason why I don't give a higher score is that this paper does not give any algorithmic contribution and the analysis, though careful, seem not to be applicable to broader cases (i.e., applying to UCB or TS algorithm, but the current analysis only works for elimination algorithms).


Some minor suggestions on presentation:

- For Table 1, I'd suggest adding the definitions of $f(G)$ and $h(G)$ in either the table caption or the footnote to remind readers.
- In warmup section, authors do not provide any pseudocode, even no informal version and informative description, which makes the section less informative. After quick passing section 3 (warmup section), I don't feel any connection between section 3 and section 4. This means I can directly read section 4 without reading section 3, which makes section 3 redundant. I would suggest removing section 3 if it does not give any warmup functionality.
- In line 12, algorithm 6, it should take union over all neighbors $v'$.

---

> ### Author Rebuttal · Authors · 2025-07-30
>
> Thank you for your comments. We hope to have addressed all your comments in the rebuttal below, and would be happy to continue discussing this with you during the author-reviewer discussion period.
> If you find our answers appropriate and satisfactory, we would be grateful if you would consider raising your score.
>
>
> # Weaknesses
>
> >“The reason why I don't give a higher score is that this paper does not give any algorithmic contribution and the analysis, though careful, seem not to be applicable to broader cases (i.e., applying to UCB or TS algorithm, but the current analysis only works for elimination algorithms)”
>
> > Q1: “Can authors discuss more why the current method does not work for UCB?”
>
> We will answer the weakness and the question together.
>
> The algorithm we introduced, Coop-SE, is new since it is not just SE with message passing, but in our implementation each agent eliminates what **other** agents have eliminated according to **their** observations (up to the delay) as well as what the agent observed and calculated. This is a critical modification, which makes it possible to align the policies.
>
> Without this modification, an agent might not eliminate a sub-optimal action, even if other agents eliminated it.
> In such a case, that agent will continue sampling this sub-optimal action alone and might incur high individual regret in this scenario.
>
> In Coop-SE, we eliminate this action due to the elimination messages from the other agents.
> This algorithmic change is crucial in our analysis of individual regret.
>
> The following is a discussion of the challenges of achieving individual regret using UCB.
>
> In UCB there is no guarantee regarding the order in which the agent explores.
> The UCB algorithm analysis bounds the number of rounds in which the sub-optimal actions are chosen.
>
> In the multi-player setting, if we only bound the number of times the sub-optimal actions are played, this is not sufficient to guarantee that the following scenario will not happen.
> It could be that many agents played the best action in many rounds. They enjoy extremely low individual regret, but these agents didn’t contribute to the exploration of sub-optimal actions.
> On the other hand, a few agents that played the sub optimal actions can play them many times. Hence, in such a scenario we have high individual regret, yet small group regret.
>
> If one wants to use UCB and achieve individual regret, one must show that the individual exploration of different agents is similar. Alternatively, one can use a reduction to a complete graph. In a reduction to a complete graph, one introduces a delay of $D$, which is added to the regret.
>
>
> >“Can author mention a paper in the stochastic setting that have low group regret, but individual regret is large?”
>
> To see the main difference between group regret and individual regret, note that when we have $m=T$ agents, we can take any agent and let it play randomly, and this increases the group regret by only $1$. This simple example shows the challenges in establishing a general transformation from group regret to individual regret.
>
> From a theoretical perspective, it is highly challenging to show individual regret and somewhat easier to prove group regret. As a concrete example, UCB-like algorithms are highly challenging for providing individual regret guarantees, although they have been shown to achieve good group regret.

---

> > ### Comment · Reviewer_oAWf · 2025-08-04
> >
> > I thank the authors for addressing my questions and have no further concerns. I agree with the other reviewers that the paper’s main contributions lie in extending existing algorithms and providing careful analysis. Also, for Coop-SE, the additive term may dominate in certain cases. While there is room for further improvement, I find the overall contributions good and lean toward a weak acceptance.

---

### Official Review · Reviewer_r6fA · 2025-06-27

**Clarity:** 2
**Significance:** 2
**Originality:** 3
**Rating:** 3
**Confidence:** 4

**Summary:**

This paper studies individual regret in cooperative stochastic multi-armed bandit (MAB) problems, where multiple agents interact over a communication graph. The authors propose Coop-SE, a fully decentralized extension of the Successive Elimination algorithm, and prove that it achieves individual regret bounds of $O(\mathcal{R}/m+A^2+A\sqrt{\log T})$, which is importantly independent of the graph's diameter. They also introduce two variants that handle communication constraints: Coop-SE-CONGEST (for small message sizes) and Coop-SE-Comm-Cost (for few communication rounds), both maintaining diameter independent regret guarantees. A lower bound is also presented, supporting the near-optimality of their approach.

**Questions:**

1. As noted earlier, the primary reason for my current score is the $T$-dependent gap between the regret upper bounds and the lower bound. That said, I do recognize the motivation outlined in Section 1.1 for pursuing diameter-independent bounds, particularly through the use of $T$-dependent terms to illustrate worst-case scenarios. I also acknowledge that the bounds presented in this work are strictly stronger than those in prior studies such as [1]. Given this, could the authors clarify how their algorithms are expected to perform on a fixed problem instance as a function of $T$? Specifically, do the proposed methods offer practical advantages over existing algorithms in such settings?
2. Could the authors provide an intuitive explanation for the emergence of the $O(\sqrt{\log T})$ and $O(\log T)$ terms in the regret upper bounds? What underlying factors contribute to this dependence? Additionally, have any attempts been made to reduce or eliminate this $T$-dependence, and if so, what challenges made it difficult to achieve?
3. In Sections 7.1 and 7.2, the authors describe preprocessing phases required to construct a spanning tree or clustering for the limited communication algorithms. How much global knowledge about the communication graph is assumed or required from the agents during these phases? Additionally, could the authors clarify how regret is accounted for during these preprocessing steps? Since each round contributes to the overall regret, it would be helpful to understand how these initial phases impact the total regret accumulation.

[1] Wang, Xuchuang, and Lin Yang. "Achieving near-optimal individual regret low communications in multi-agent bandits." The Eleventh International Conference on Learning Representations (ICLR). 2023.

**Ethical Concerns:**

["NO or VERY MINOR ethics concerns only"]

**Final Justification:**

As I detailed in my response to the discussion initiated by the AC, I will not be changing my score at this time.

**Limitations:**

Yes

**Paper Formatting Concerns:**

I did not notice any major formatting issues.

**Quality:**

2

**Strengths And Weaknesses:**

**Strengths:**
- **Quality:** The paper’s main claims are backed by thorough and well-structured proofs, demonstrating a strong theoretical foundation.
- **Clarity:** The authors effectively explain the motivation behind removing diameter dependence, using intuitive examples. The algorithms and proof techniques are clearly described and mostly self-contained within the main paper.
- **Significance:** Although cooperative multi-agent MAB has been studied extensively, this is the first work to achieve individual regret bounds that are independent of both the graph diameter and the number of agents.
- **Originality:** While the Successive Elimination and message passing ideas are not new in this setting, the paper introduces original contributions—such as implicit synchronization across agents and cluster-based strategies—to obtain diameter-independent bounds.

Overall, I am convinced that achieving diameter-independent bounds is important, and appreciate the authors’ effort to also discuss limited communication settings.

**Weaknesses:**
Rather than separating into broad quality, clarity, significance and originality categories, I will outline my main concerns in a more detailed manner below.
- While the regret bound in Theorem 2 (As well as Theorems 4 and 5) is claimed to be significant due to its independence from the graph diameter, I am not fully convinced of its practical value because of the presence of a $O(\sqrt{\log T})$-dependent term. As $T$ grows, this term can dominate, potentially nullifying the advantage of removing diameter dependence. In fact, as noted in line 194, the *actual* bound is $O(A\min (A+\sqrt{\log T},D))$, meaning that for sufficiently large horizons, the regret essentially matches the bound of [1]. This raises concerns about the tightness and novelty of the result. The absence of experimental validation to support the claimed improvements further amplifies this concern, as it leaves open the question of whether the proposed algorithm offers tangible advantages in realistic settings.
- This concern is further reinforced by the significant gap between the $O(A\sqrt{\log T})$ additive term in the upper bound and the $\Omega(\sqrt{A})$ term in the lower bound. It remains unclear whether the $T$-dependence in the additive term is truly necessary, or whether a tighter, $T$-independent bound might be achievable.
- While the treatment of limited communication settings in Sections 7 and 8 is valuable and introduces novel ideas, I believe addressing the substantial gap in all of the regret bounds is a more pressing issue. This gap makes the overall theoretical contribution feel somewhat incomplete. Furthermore, the regret bound for Coop-SE-Comm-Cost includes a larger additive term of $O(A\log T)$, which—aside from being independent of the gaps—effectively offsets the cooperative benefit of the $O(1/m)$ term by dominating it for large $T$.
- As previously noted, this work does not include any experimental evaluation. While the primary contributions are theoretical, it would still be valuable to empirically demonstrate that the proposed algorithms outperform baseline methods in practice—at least in representative scenarios. This is particularly important given the noticeable gap between the upper and lower bounds on regret, which raises questions about the practical tightness of the theoretical results.

[1] Wang, Xuchuang, and Lin Yang. "Achieving near-optimal individual regret low communications in multi-agent bandits." The Eleventh International Conference on Learning Representations (ICLR). 2023.

Some other concerns –
- The overall structure of the paper could benefit from refinement. The Introduction and Related Work sections extend to page 4, which feels overly long. Consider moving parts of the Related Work—particularly the final paragraph, which discusses broad extensions of cooperative bandits not directly relevant to the paper’s main contributions—to the appendix. This would free up space to include a more detailed explanation of the Coop-SE algorithm in Section 4, and to add a proper Conclusion and Future Work section to the main paper.
- Additional proofreading would improve the paper. For instance, there appears to be a typo in line 73 regarding the regret bound for Coop-SE-Comm-Cost, which is inconsistent with the other results presented. The algorithms proposed in the paper should also be mentioned explicitly in the abstract. In line 184, the word “is” should be removed. Moreover, Algorithm 2—a simplified version of Coop-SE—is included in the main text but is never referenced or discussed, which may confuse readers. There is also a mismatch in the presentation of regret bounds: the lower bound is given in minimax form, while the upper bounds are instance-dependent. This transition should be explicitly addressed in the main paper. Lastly, the spanning tree notation $\mathcal{T}$, introduced in line 335, is used without prior definition in the main text.

---

> ### Author Rebuttal · Authors · 2025-07-30
>
> Thank you for your comments. We hope to have addressed all your comments in the rebuttal below, and would be happy to continue discussing this with you during the author-reviewer discussion period.
> If you find our answers appropriate and satisfactory, we would be grateful if you would consider raising your score.
>
> # Weaknesses
>
> >”...I am not fully convinced of its practical value because of the presence of a $O(\sqrt{\log(T)})$-dependent term. As $T$ grows, this term can dominate, potentially nullifying the advantage of removing diameter dependence...”
>
> We emphasize that the main focus of our paper is theoretical. We would like to understand the inherent regret bound and its dependence on the various parameters of the problem.
> Previous works had an additive term of $D$, the diameter of the graph.
> We highlight that one needs to think of the diameter $D$ as a function of $m$, the number of nodes in the graph, which is also the number of agents.
> An interesting communication network is a grid graph where $D=\sqrt{m}$. Since the ideal regret would be at least $\mathcal{R} / m$, if we aim for low regret, the number of agents $m$ should be a function of $T$.
> This implies that $D$ is also a function of $T$ (indirectly, through $m$).
> Hence, the dependency on $D$ hides a dependency on $T$, which is potentially polynomial for small gaps.
>
> About the actual parameters, even in the extreme case that $T=10^{50}$, the number of atoms on Earth, our bound is very reasonable, since $\sqrt{\log(T)} < 11$. Having a diameter more than 11 is definitely reasonable in many cases. We stress again that our focus is theoretical, and this numerical example is only to highlight why in many cases the main term would be $D$.
>
>
> As discussed in Section 1.1, one can see that small sub-optimality gaps ($\Delta \approx 1/\sqrt{T}$) may contribute a lot to the regret. If the regret bound depends on the diameter, one remains with a polynomial dependency on $T$ (for example, $T^{1/6}$ for a grid graph), and increasing the number of agents will not help. But with the bounds from Coop-SE, for a sufficiently large number of agents, one remains with only $A^2+A\sqrt{\log(T)}$ regret.
>
> >”...It remains unclear whether the $T$-dependence in the additive term is truly necessary, or whether a tighter, $T$-independent bound might be achievable.”
>
>
> Let us give an intuition about the $\log(T)$ and $\sqrt{\log(T)}$ in the additive terms. The $\log(T)$ terms (or $\sqrt{\log(T)}$) are there to ensure high-probability concentration bounds.
> Since we need the concentration bound to hold simultaneously for any agent, action and time step, we have an actual $\sqrt{log(mAT)}$ term.
>
> The additive $\sqrt{\log(mAT)}$ term arises in cases where the number of agents is large compared to the stage length. In sparse graphs like the line graph, even with many agents, only $O(t^2)$ samples can be collected by time $t$. This leads to a $\sqrt{\log(mAT)}$ term from concentration bounds. See line 977 in the appendix.
>
> To summarize, these logarithmic terms are due to the concentration bounds, which we require for our analysis. The tightness of the lower bound remains an open question, as we mention in lines 201-202.
>
>
> >"While the treatment of limited communication settings in Sections 7 and 8 is valuable and introduces novel ideas, I believe addressing the substantial gap in all of the regret bounds is a more pressing issue.”
>
> We agree that achieving an algorithm that gives the best possible regret bound would be a substantial contribution, however, we feel that we made significant progress in that direction.
> It is still an open question whether the lower bound of $\sqrt{A}$ is indeed the tightest, and we mention this open problem in the paper (lines 201-202 and in Appendix A lines 818-819).
>
> >”This gap makes the overall theoretical contribution feel somewhat incomplete.”
>
> The main contribution of the paper is a simple algorithm, and an analysis that shows that this algorithm achieves $\mathcal{R}/m + A^2 + A\sqrt{\log(T)}$ individual regret for any graph.
> Other methods depend on the diameter $D$.
> As we mentioned before, the diameter naturally depends on the size of the graph, $m$, the number of agents. The number of agents is a function of $T$, if we aim to achieve low regret. Please see Section 1.1 for more explanation of why in some natural cases, those methods yield $T^c$ regret, for $c>0$.
>
> >”Furthermore, the regret bound for Coop-SE-Comm-Cost includes a larger additive term of $O(A\log T)$, which—aside from being independent of the gaps—effectively offsets the cooperative benefit of the $O(1/m)$ term by dominating it for large $T$.”
>
> We discussed before the relationship between $A\sqrt{\log(T)}$ and the diameter $D$.
> An additional issue that one needs to consider is the sub-optimality gaps.
> The gaps are parameters that may contribute substantially to the regret. The benefit of $O(1/m)$ still has a high impact, since the gaps might contribute $\sqrt{T}$ regret, as explained in Section 1.1. The $(1/m)$ term overcomes this potentially large $\sqrt{T}$ regret.
>
> >”As previously noted, this work does not include any experimental evaluation...”
>
> This work's main focus is theoretical and therefore does not include experiments.
>
> # Other concerns
>
> We thank the reviewer for their suggestions.
> We will correct the typos and inconsistencies.
> For completeness, Theorem 4 should be stated as: $O(\mathcal{R}/m + A\sqrt{\log(mAT)} + A^2)$, and Theorem 5 as: $O(\mathcal{R}\log(A)/m + A\log(A)\log(mAT)) $.
>
>
> # Questions
>
> >Q1 “...could the authors clarify how their algorithms are expected to perform on a fixed problem instance as a function of $T$? Specifically, do the proposed methods offer practical advantages over existing algorithms in such settings?”
>
> Other methods suffer from a regret of at least $D$, where $D$ is the diameter of the graph. This holds independently from the number of agents.
> Our algorithm offers a much better regret.
>
> Here is how we expect our algorithm to work on a cycle graph with sufficiently many agents.
> An agent at time $t$ will receive about $t^2$ samples. This implies that it will be able to discard a sub-optimal action $i$ after roughly $\log (T) / \Delta_i$ rounds, in contrast to the single-agent setting where the elimination occurs only after $\log (T) / \Delta_i^2$ . Note that this time can be much smaller than the diameter $D$, and this is the main advantage of our algorithms and analysis.
> An important aspect of the algorithm is that agents send the actions they discarded, and therefore guarantee that the agents have similar policies.
>
>
> **A more detailed comparison with other methods:**
> In UCB there is no guarantee regarding the order in which the agent explores.
> The UCB algorithm analysis bounds the number of rounds the sub-optimal actions are chosen.
>
> In the multi-player setting, if we only bound the number of times the sub-optimal actions are played, this is not sufficient to guarantee that the following scenario will not happen.
> It could be that many agents played the best action in many rounds. They enjoy extremely low individual regret, but these agents didn’t contribute to the exploration of sub optimal actions.
> On the other hand, a few agents that played the sub optimal actions can play them many times. Hence, in such a scenario we have high individual regret, yet small group regret.
>
> If one wants to use UCB and achieve individual regret, one must show that the individual exploration of different agents is similar. Alternatively, one can use a reduction to a complete graph, as mentioned in Wang et al. [1]. In a reduction to a complete graph, one introduces a delay of $D$, which is added to the regret.
>
> Wang et al. [1] use a version of UCB and give a precise analysis for the complete graph with extremely low communication. To address a general graph, they give a general reduction.
> Their reduction implicitly assumes that all agents’ policies are the same (as far as we understand). The reviewer may look at Wang et al. [1], section G.2: “the algorithm solves a problem equivalent to a CMA2B with communication delay D on a complete graph”. (CMA2B is an abbreviation of “Cooperative multi-agent multi-armed bandits”, and $D$ is the diameter). The reduction to a complete graph requires delay of $D$ which is added to the regret.
>
> Coop-SE does not depend on the diameter $D$, and enjoys better regret.
>
>
>
> >Q2 ”Could the authors provide an intuitive explanation for the emergence of the  $O(\sqrt{\log(T)})$  and $O(\log(T))$ terms in the regret upper bounds? ... Additionally, have any attempts been made to reduce or eliminate this $T$-dependence…?”
>
> As we mentioned before, the $\sqrt{\log(T)}$ and the $\log(T)$ terms in the additive term are due to the use of concentration bounds, which are required for our analysis.
> The reviewer may refer to our answer to a previous comment they made.
>
>
> > Q3 “How much global knowledge about the communication graph is assumed or required from the agents during these phases? Additionally, could the authors clarify how regret is accounted for during these preprocessing steps?”
>
> Since we assume the agents have common knowledge of the communication graph, the spanning tree is also part of it, and all the agents know it in advance.
>
> Constructing a spanning tree in a distributed way has been studied in the communication networks community; see Peleg [2].
>
> Constructing the spanning tree in an online fashion while maintaining low regret under communication constraints remains an open question that is beyond the scope of this paper.
>
> --------------
>
> [1] Wang, Xuchuang, and Lin Yang. "Achieving near-optimal individual regret low communications in multi-agent bandits." The Eleventh International Conference on Learning Representations (ICLR). 2023.
>
> [2] Peleg, David. “Distributed computing: a locality-sensitive approach”. SIAM, 2000.

---

> > ### Comment · Reviewer_r6fA · 2025-08-04
> >
> > My first concern relates to the presence of $O(\sqrt{\log T})$-dependent terms in the additive component of the regret upper bounds. I appreciate the authors’ efforts to clarify this point, both in the rebuttal and in the well-written Section 1.1. When $m$ and the gap parameters depend on $T$, I acknowledge that the bounds can indeed offer meaningful improvements over prior results. However, I do not fully agree with the claim that $m$ must necessarily be a function of $T$, as the number of agents is not always under the designer’s control. This is why I initially raised concerns about settings with fixed $T$. The authors' argument that the $O(\sqrt{\log T})$ term is practically negligible, even for large horizons, partially addresses this issue.
> >
> > That said, my concerns about the gap between the lower and upper bounds remain. While the authors explain that this term is required to establish concentration bounds within their proof framework and note that closing this gap is an open question, it nonetheless remains a limitation of the current analysis. This stems from the authors stressing the main focus of their work is theoretical, and they aim to identify the inherent dependence of the regret bound on the various parameters of the problem. Furthermore, as mentioned in my initial review, the lack of any experimental validation of the proposed algorithm also limits the practical impact of the work.
> >
> > In light of these concerns, and considering the points raised by other reviewers, I will maintain my current score for now. I will continue to consider all information during the reviewer-AC discussion period.

---

> > > ### Author Response · Authors · 2025-08-05
> > >
> > > Thank you again for your comments. We agree that in some practical settings the number of agents $m$ may be small and not under the designer’s control. In such cases, our bound still recovers the optimal improvement from single-agent regret $\mathcal{R}$ to $\mathcal{R} / m$, as in prior works. Both in our case and in prior works, the improvement over single-agent regret is not very significant for small constant $m$, and only becomes significant when $m$ gets larger — this is also where our improvement over prior work becomes meaningful.

---

### Official Review · Reviewer_Kf67 · 2025-06-28

**Clarity:** 3
**Significance:** 2
**Originality:** 3
**Rating:** 4
**Confidence:** 4

**Summary:**

This paper examines individual regret in multi-agent stochastic MABs residing on a communication graph. This paper explicitly investigates how to remove the dependency on the graph diameter from the regret upper bound, which is achieved through a more sophisticated analysis of the Coop-SE algorithm. Then, the paper extends the Coop-SE algorithm to two types of low-communication variants, which have reasonably worse regret upper bounds.

**Questions:**

- Line 37: why omit the $\log T$ regret term here? It dominates the $\sqrt{\log T}$ term.
- Line 65: the $\mathcal R$ still dependent on the gaps.
- Line 67: It would be better to present the upper bound in minimax form here as well.
- Line 74—79: the open problem can be addressed by other prior works mentioned in the related work section, why emphasizes it here? Is there a special insight of the current work on the problem?
- The communication round of TCOM should be additive $D$ instead of multiplicative $D$.

**Ethical Concerns:**

["NO or VERY MINOR ethics concerns only"]

**Final Justification:**

While the reviewer decides to maintain the borderline accept evaluation of this paper, the reviewer feels that the contribution of this paper does not reach the bar of NeurIPS, and is ok with a rejection decision.

This paper studies a known multi-agent bandits problem, analyses a known algorithm Coop-SE, and only provides a new regret upper bound, which is tighter than the known bounds only in some special cases, i.e., $D>\sqrt{\log T}$.

**Limitations:**

yes

**Quality:**

3

**Strengths And Weaknesses:**

- S1: The analysis for Coop-SE (especially Lemma 1) is more sophisticated than prior ones, enabling the removal of dependence on $D$.  The reviewer thinks that it is an interesting new approach.
- S2: The writing of this paper is clear and motivating. S1.1 did a good job of motivating the study, and S5 provides an easy-to-follow proof sketch.
---
- W1: Although concretely new progress on the direction of multi-agent MAB, the reviewer thinks that the result in this paper is not very significant (even the authors did a good job in the discussion in S1.1). The main improvement of this paper is from $D$ to $\min\{A+\sqrt{\log T}, D\}$, where the latter can remove the dependence of $D$ when the graph has a large diameter. However,
    - (1) the $O(\sqrt{\log T})$ term is still relatively large since the time horizon $T$ in the bandit setting is typically the largest constant.
    - (2) The improvement only happens when the graph has a large diameter (larger than $\sqrt{\log T}$!).
- W2: The current results are not tight. The upper bound $O(A^2)$ has a large gap in comparison with the $\Omega(\sqrt A)$ lower bound.
- W3: Although the analysis is new, the Coop-SE algorithm has been proposed multiple times in various prior works (as the authors are also aware). It would be better to posit the contribution of this work on the analysis side and attribute the merit of the Coop-SE algorithm design to these prior works (e.g., remove “ours” after Coop-SE in Table 1).

---

> ### Author Rebuttal · Authors · 2025-07-30
>
> Thank you for your comments. We hope to have addressed all your comments in the rebuttal below, and would be happy to continue discussing this with you during the author-reviewer discussion period.
> If you find our answers appropriate and satisfactory, we would be grateful if you would consider raising your score.
>
>
> >W1: “Although concretely new progress on the direction of multi-agent MAB, the reviewer thinks that the result in this paper is not very significant (even the authors did a good job in the discussion in S1.1). The main improvement of this paper is from $D$  to $\min \{A+\sqrt{\log(T)},D\}$, where the latter can remove the dependence of $D$ when the graph has a large diameter. However, (1) the $O(\sqrt{log(T)})$ term is still relatively large since the time horizon $T$ in the bandit setting is typically the largest constant.
> (2) The improvement only happens when the graph has a large diameter (larger than $\sqrt{\log(T)}$!).”
>
>
> We emphasize that the main focus of our paper is theoretical. We would like to understand the inherent regret bound and its dependency on the various parameters of the problem.
>
> Previous works had an additive term of $D$, the diameter of the graph.
> We highlight that one needs to think of the diameter $D$ as a function of $m$, the number of nodes in the graph, which is also the number of agents.
> An interesting communication network is a grid graph where $D=\sqrt{m}$. Since the ideal regret would be at least $\mathcal{R} / m$, if we aim for low regret, the number of agents $m$ should be a function of $T$.
> This implies that $D$ is also a function of $T$ (indirectly, through $m$).
> Hence, the dependency on $D$ hides a dependency on $T$, which is potentially polynomial for small gaps.
>
> Regarding the actual parameters, even in the extreme case that $T=10^{50}$, the number of atoms on Earth, our bound is very reasonable, since $\sqrt{\log(T)} < 11$. Having a diameter greater than $11$ is definitely reasonable in many cases. We stress again that our focus is theoretical, and this numerical example is only to highlight why in many cases the main term would be $D$.
>
> As discussed in Section 1.1, one can see that small sub-optimality gaps ($\Delta \approx 1/\sqrt{T}$) may contribute a lot to the regret. If the regret bound depends on the diameter, one remains with a polynomial dependency on $T$ (for example, $T^{1/6}$ for a grid graph), and increasing the number of agents will not help. But with the bounds from Coop-SE, for a sufficiently large number of agents, one remains with only $A^2+A\sqrt{\log(T)}$ individual regret.
>
>
> > W2: “The current results are not tight. The upper bound $O(A^2)$ has a large gap in comparison with the $\Omega(\sqrt A)$ lower bound.”
>
>
> Regarding the gap in the additive term between the lower bound and the upper bound, the reviewer may find it interesting that when the played action is selected randomly from the set of active actions, we get $\mathcal{R}/m + A \log(mAT)$ (instead of the $\mathcal{R}/m + A^2 + A\sqrt{\log(mAT)}$  with round-robin).
>
> It is still an open question whether the lower bound of $\sqrt{A}$ is indeed the tightest, and we mention it in the future work section (lines 201-202 and in Appendix A, lines 818-819).
>
>
> > W3: “Although the analysis is new, the Coop-SE algorithm has been proposed multiple times in various prior works (as the authors are also aware). It would be better to posit the contribution of this work on the analysis side and attribute the merit of the Coop-SE algorithm design to these prior works (e.g., remove “ours” after Coop-SE in Table 1).”
>
>
> We will emphasize in the final version that SE with message passing in the cooperative setting is not a new algorithm.
>
> The algorithm we introduced, Coop-SE, is new since it is not just SE with message passing, but in our implementation each agent eliminates what **other** agents eliminated according to **their** observations (with elimination messages), as well as what the agent observed and calculated. This is a critical modification, which makes it possible to align the policies of the different agents.
>
> Without this modification, an agent might not eliminate a sub-optimal action, even if other agents eliminated it.
> In such a case, such an agent will continue sampling this sub-optimal action alone and might incur high individual regret in this scenario.
>
> In Coop-SE, we eliminate this action due to the elimination messages from other agents.
> This algorithmic change is crucial in our analysis for the individual regret.
>
>
> # Questions
>
> >”Line 37: why omit the $\log(T)$ regret term here? It dominates the $\sqrt{\log(T)}$ term.”
>
> We omitted the $\log(T)$ term since for a large number of agents this term vanishes.
>
> >”Line 65: the $\mathcal{R}$ still dependent on the gaps.”
>
> Thank you, we will change this.
>
> >”Line 67: It would be better to present the upper bound in minimax form here as well.”
>
> Yes, we agree, and we will present both bounds in the minimax form.
>
> >”Line 74—79: the open problem can be addressed by other prior works mentioned in the related work section, why emphasizes it here? Is there a special insight of the current work on the problem?”
>
> We will definitely add a discussion regarding this point.
> We acknowledge that this question has been addressed by prior works. While our contribution lies in analyzing individual regret independent of the graph’s diameter, we will explicitly state in the final version that prior works have also contributed to this specific question.
>
> >”The communication round of TCOM should be additive $D$ instead of multiplicative $D$.”
>
> Thanks, we will change this.

---

> ### Comment · Reviewer_Kf67 · 2025-08-02
>
> The reviewer thanks the authors for their responses.
>
> Coop-SE with both message passing and eliminated arm signal is already a common technique in the literature.
>
> See Algorithm 2 of "Yang, Lin, et al. "Distributed bandits with heterogeneous agents." IEEE INFOCOM 2022-IEEE Conference on Computer Communications. IEEE, 2022." and papers cited it.

---

> > ### Author Response · Authors · 2025-08-03
> >
> > We thank the reviewer for pointing out the connection to Yang et al. (2022), and we agree that the use of elimination signals was proposed by them. However, the way this signal is utilized differs substantially. In the algorithm of Yang et al., elimination signals are used solely to track the set of active arms of other agents and to reduce communication, but agents may still sample arms eliminated by others. In contrast, in our algorithm, agents eliminate arms based on received elimination signals, which is crucial for achieving synchronization across the network - a key property required for our individual regret analysis. We will give the appropriate credit to Yang et al. (2022) and will clarify this distinction in the final version.

---

> > > ### Comment · Reviewer_Kf67 · 2025-08-03
> > >
> > > See:
> > > - Algorithm 1 of Yang, Lin, et al. "Cooperative Multi-agent Bandits: Distributed Algorithms with Optimal Individual Regret and Communication Costs." Coordination and Cooperation for Multi-Agent Reinforcement Learning Methods Workshop. 2023.
> > > - Algorithm 1 of Zhang, Haoran, et al. "Near-Optimal Regret Bounds for Federated Multi-armed Bandits with Fully Distributed Communication." The 41st Conference on Uncertainty in Artificial Intelligence.

---

> > > > ### Author Response · Authors · 2025-08-03
> > > >
> > > > We thank the reviewer for the helpful references and will ensure that the key algorithmic ideas are properly attributed to prior work.
> > > > We will add a discussion on this prior work and their algorithmic contributions.

---

> > > > > ### Comment · Reviewer_Kf67 · 2025-08-04
> > > > >
> > > > > In light of the additional reference, the authors should attribute the Coop‑SE algorithm to prior work and limit their claimed contribution to the analytical aspects only. Since this change likely goes beyond a simple edit, the reviewer recommends the authors provide a detailed revision plan specifying exactly how and where the Discussion (and any related sections) will be revised. Without such a plan, reviewers will have insufficient guidance to assess whether the current manuscript is suitable for publication.

---

> > > > > > ### Author Response · Authors · 2025-08-04
> > > > > >
> > > > > > While Algorithm 2 in Yang et al. does not involve message passing (as they consider a complete graph), it is nonetheless closely related to Coop-SE, as it also employs successive elimination and makes use of elimination signals from other agents. However, to our knowledge, no prior work combines both the communication mechanism and structural aspects used in Coop-SE. That said, we agree that the algorithmic ideas underlying Coop-SE have appeared in previous works, although not in combination. Accordingly, we will remove the algorithmic contribution of Coop-SE from our stated contributions—especially from the key contributions—and emphasize that our main contribution lies in the analysis. We will also remove the “ours” label from Table 1 to make this clear to the reader.
> > > > > >
> > > > > > To address this more explicitly, we will add a short discussion as a subsection following the key contributions and preceding the related work, clarifying the connections between Coop-SE and prior algorithms. In addition, we will revise Section 4 to state that the algorithm builds on prior work, and that the focus of our contribution is the analysis showing individual regret that does not scale with the graph diameter.

---

> > > > > > > ### Comment · Reviewer_Kf67 · 2025-08-04
> > > > > > >
> > > > > > > Both of the following references used the idea of eliminating arms that other agents eliminated and many other papers as well, which weakens the contributions of this paper. See:
> > > > > > >
> > > > > > > Algorithm 2 of Yang, Lin, et al. "Cooperative Multi-agent Bandits: Distributed Algorithms with Optimal Individual Regret and Communication Costs." Coordination and Cooperation for Multi-Agent Reinforcement Learning Methods Workshop. 2023.
> > > > > > > Algorithm 1 of Zhang, Haoran, et al. "Near-Optimal Regret Bounds for Federated Multi-armed Bandits with Fully Distributed Communication." The 41st Conference on Uncertainty in Artificial Intelligence.
> > > > > > >
> > > > > > > Given that, the reviewer feels that the authors do not fully address their concern on the novelty of this paper, and will bring that up in the AC-reviewer discussion.

---

> > > > > > > > ### Author Response · Authors · 2025-08-05
> > > > > > > >
> > > > > > > > As we stated, we will revise the paper to include all relevant prior works. This includes the two papers by Yang et al., the paper by Zhang, Haoran, et al., and of course any other paper that we find relevant.
> > > > > > > >
> > > > > > > > To clarify a possible confusion, we referred to two different algorithms from two different papers by Yang et al.
> > > > > > > >
> > > > > > > > In the reviewer’s first comment, they pointed us to Algorithm 2 in Yang et al., “Distributed Bandits with Heterogeneous Agents”. We addressed this algorithm in our initial response and acknowledged its relevance.
> > > > > > > >
> > > > > > > > In the follow-up comment, the reviewer referred to Algorithm 1 in Yang et al., “Cooperative Multi-agent Bandits: Distributed Algorithms with Optimal Individual Regret and Communication Costs”. However, we believe this was a typo. In our previous response, we were referring to Algorithm 2 of that same paper, which is the relevant bandit algorithm. We encourage the reviewer to refer back to our comment to verify this.
> > > > > > > >
> > > > > > > > We will include a discussion of both Yang et al. papers, as well as the related work by Zhang et al., which uses successive elimination and elimination signals too, while using consensus estimation.
> > > > > > > >
> > > > > > > > Our algorithm, Coop-SE, is not a novel algorithm, as its main components were introduced in prior work. We will explain this clearly in the revised version, as outlined in our response plan above.
> > > > > > > >
> > > > > > > > Accordingly, we will focus our stated contributions solely on the analysis side—on providing individual regret guarantees that do not scale with the diameter of the graph.

---

### Official Review · Reviewer_xuc7 · 2025-07-02

**Clarity:** 3
**Significance:** 3
**Originality:** 3
**Rating:** 5
**Confidence:** 3

**Summary:**

This paper addresses the cooperative stochastic multi-armed bandit problem, where a number of agents ($m$), connected via a fixed graph, face the same bandit instance and can exchange information with their neighbours. The goal here is to guarantee small regret for every individual agent in the network. The authors noted that previous works achieve regret bounds of the form $\mathcal{R}/m + D$, where $\mathcal{R}$ is the optimal single-player regret and $D$ denotes the diameter (or some other function) of the graph that can be as large as $m$ in some cases. This additive dependence on $m$ means that the contribution of $\mathcal{R}$ to the regret bound cannot be made arbitrarily small in general as $m$ increases. The authors address this issue by proposing an algorithm that enjoys a similar regret bound but replaces $D$ with $A \cdot \min(D, A + \sqrt{\log T})$, where $A$ is the number of actions and $T$ the time horizon. A worst-case lower bound of $\sqrt{AT / m} + \sqrt{A}$ is also provided, matching the first term of the upper bound in a worst-case sense and showing that some additive dependence on $A$ can be unavoidable for some graphs. The adopted algorithm, Coop-SE, is an adaptation of the successive elimination algorithm to the cooperative setting that is completely decentralized and uses only local graph information. Similar regret guarantees are also provided for two extensions of the algorithm that respect constraints on the message size or the number of communication rounds.

**Questions:**

Concerning the additive term, do you believe that the shortcoming is in the upper bound? If so, could this be a limitation of the successive elimination algorithm? Could it be a limitation of only using local graph information?

**Ethical Concerns:**

["NO or VERY MINOR ethics concerns only"]

**Final Justification:**

My evaluation remains on the positive side. I think the regret bound presented in the paper brings an interesting improvement over the cited prior results, particularly in the worst-case regime where the sub-optimality gaps are small. The presentation of the algorithm and the results is fairly nice and clear. However, I think the authors should do a more thorough job in highlighting the limitations of their results; e.g., regimes where their bound fails to bring an improvement, the gap between the upper and lower bounds, and the fact that the lower bound has a worst-case flavour in terms of the graph structure. In addition, of course, to highlighting more clearly the similarities with algorithms from prior works as mentioned by other reviewers.

**Limitations:**

Yes, a brief discussion on the limitations of this work are discussed in the future work section.

**Paper Formatting Concerns:**

No formatting concerns.

**Quality:**

3

**Strengths And Weaknesses:**

The paper seems to provide a solid contribution to the literature on this problem. Unlike the cited regret bounds previous works, the dependence of the new regret bound on the reciprocals of the sub-optimality gaps can get arbitrarily small as the number of agents increase, regardless of the structure of the graph, so long as it is connected. The adopted algorithm is simple and intuitive, and the explanations are fairly clear.

On the other hand, the presented results still do not provide a clear idea as to what extent the structure of the graph can be exploited, particularly concerning the second (additive term) in the regret. This term seems to be at least $A$ in the upper bound for any graph (including, e.g., the complete graph), while the opposing $\sqrt{A}$ term in the lower bound only concerns a particular structure (the line graph) and does not seem to apply in a graph-specific sense. Perhaps this should be emphasized more. Aside from this, I see no major weaknesses in the paper. Some minor points concerning the introduction; it does not seem that the additive terms in Theorems 4 and 5 are exactly as reported in Section 1.2 and Table 1; moreover, in lines 91-93, it seems that the argument should be that $h(\mathcal{G})$ can increase (potentially up to $m$) if $\lambda_2$ is close to $1$, not $0$.

---

> ### Author Rebuttal · Authors · 2025-07-30
>
> Thank you for your comments. We hope to have addressed all your comments in the rebuttal below, and will be happy to continue discussing this with you during the author-reviewer discussion period.
>
>
> > “the presented results still do not provide a clear idea as to what extent the structure of the graph can be exploited”
>
> We will emphasize in the final version that leveraging the graph structure can potentially reduce the additive term in the regret. For instance, in the case of a complete graph, the regret can be as low as $\mathcal{R}/m$, and if $m > \mathcal{R}$ the regret is constant.
> An interesting open problem is to characterize the dependencies of the regret on the graph’s structure. We will add it to the Future Work section.
>
> Our goal is to derive graph-independent regret bounds and to give regret bounds that hold for any graph. As you remark, the only potential gains are in the dependency of the additive term, which is already rather insignificant compared to the regular single-agent regret.
>
>
> >”Some minor points concerning the introduction; it does not seem that the additive terms in Theorems 4 and 5 are exactly as reported in Section 1.2 and Table 1”
>
> Thank you for noticing the mismatch between Theorems 4 and 5 and to the statements elsewhere. We will fix this. Theorem 4 should be: $O(\mathcal{R}/m + A\sqrt{\log(mAT)} + A^2)$, and Theorem 5 should be: $O(\mathcal{R}\log(A)/m + A\log(A)\log(mAT)) $.
>
> >”moreover, in lines 91-93, it seems that the argument should be that $h(\mathcal{G})$ can increase (potentially up to $m$) if $\lambda_2$ is close to $1$, not $0$.
>
> It indeed should be “The eigenvalue … can be very close to \textbf{one} for some graphs” (not to zero). Thanks again.
>
> # Questions
>
> >”Concerning the additive term, do you believe that the shortcoming is in the upper bound? If so, could this be a limitation of the successive elimination algorithm? Could it be a limitation of only using local graph information?”
>
> We believe that this is an artifact of the analysis rather than of the algorithm. The additive $A$ term arises in part because we treat each stage as having a length of at least $1$, which leads to a regret of at least $A$. However, potentially, multiple stages may have length $0$. Unfortunately, we were unable to refine our analysis to account for this.
> When the agents are completely connected (there is no issue of local graph information), i.e., the graph is a complete graph, the additive term is eliminated with our algorithm.

---

> > ### Comment · Reviewer_xuc7 · 2025-08-03
> >
> > Thank you for your response.
> >
> > As also hinted at by Reviewer r6fA, I am not sure I completely agree that the additive term is insignificant compared to the first term. Sure, in a worst case scenario where the gaps are very small, the dependence of the additive term on $T$ is relatively insignificant; hence, in a graph with large $m$ and large $D$, your bound seems indeed advantageous compared to prior results. However, in a favourable instance with constant gaps, if both $m$ and $D$ are large, the additive term would come to dominate while being in some sense comparable to $\mathcal{R}$, seemingly failing in this case to offer a significant improvement compared to prior results with the additive $D$.
> >
> > Do the authors agree with this reasoning?
> >
> > This does not nullify your contributions; but it would be better to highlight such possible limitations and whether they are shortcomings of the adopted algorithm/analysis.

---

> > > ### Author Response · Authors · 2025-08-03
> > >
> > > We thank the reviewer for their comment.
> > >
> > > Our additive term is $A\min\\{A + \sqrt{\log T}, D\\}$.
> > >
> > > When both $m$ and $D$ are large, the $\sqrt{\log T}$ term remains small—especially compared to the $AD$ additive term used in other works.
> > >
> > > When $m$ is small, cooperation becomes less relevant. Recall that the Coop-SE algorithm essentially reduces to successive elimination for a single agent, with some enhancements for cooperation.

---

> > > > ### Comment · Reviewer_xuc7 · 2025-08-04
> > > >
> > > > Thank you for your response.
> > > >
> > > > I think my remark was not very clear, so let me expand on it a bit. Consider a favourable instance where the gaps are constant, so that $\mathcal{R}$ is essentially of order $A \log(T)$. Now assume that the graph is a grid graph with $m = \log(T)$ nodes, hence $D$ is of order $\sqrt{m}$ as mentioned in the paper. In this case, the first term in the regret bound is essentially constant (does not depend on $T$), while the additive term would be $A \sqrt{\log T}$ (same order as $AD$), which is comparable to $\sqrt{A \mathcal{R}}$. It seems to me that in cases like this, the dependence of the additive term on the structure of the graph (here, through $D$) is significant, and it might be worth studying to what extent the graph could be exploited.

---

> > > > > ### Author Response · Authors · 2025-08-04
> > > > >
> > > > > We agree that in this specific example, our bound only recovers previous regret bounds. However, in many other cases (particularly when D is large) we may achieve a significant improvement. We also agree that leveraging the graph’s structure offers an additional avenue for improvement. In your example of a grid graph with a relatively short diameter of $\log T$, a more refined analysis might reveal that Coop-SE achieves a smaller additive term.
> > > > > We will include a discussion on exploiting the graph’s structure in the future work section

---

> > > > > > ### Comment · Reviewer_xuc7 · 2025-08-07
> > > > > >
> > > > > > Thank you for your responses. My overall view of the paper remains on the positive side.

---

### Note · Authors · 2025-08-15

Our contributions, as acknowledged by all reviewers, include **state-of-the-art individual regret bounds** and a **novel analysis** for the cooperative stochastic MAB problem over a graph. Our individual regret bound is independent of the graph’s diameter, and the analysis is not a reduction to a fully-connected graph, but instead relies on new ideas based on elimination stages.
In addition, our paper addresses network constraints, and our analysis in this context is also novel.

>”The paper seems to provide a solid contribution to the literature on this problem.” xuc7

>”S1: The analysis for Coop-SE (especially Lemma 1) is more sophisticated than prior ones, enabling the removal of dependence on $D$ The reviewer thinks that it is an interesting new approach.” Kf67

>”While the Successive Elimination and message passing ideas are not new in this setting, the paper introduces original contributions—such as implicit synchronization across agents and cluster-based strategies—to obtain diameter-independent bounds.” r6fA

>”I don't see any major weakness of this paper, and authors have acknowledged that there exists a gap between the upper and lower bounds. Though the design of algorithm is simple, ie., extending elimination based algorithms to multi-agent setup, it is indeed effective, and authors do a careful analysis to get a good regret bound.” oAWf

>”I think that the results of this paper are quite remarkable.” LDks

Related works: The reviewers have pointed out additional prior works that use action-elimination in the cooperative setting.
We would definitely give proper and fair reference to all of those works in the final version, and we will explain that they also imply (various versions) of action elimination, and clearly pre-dated our work.
We would like to stress that all the reviewers agree that our analysis and the derived regret bound are indeed new and novel.

Additive $\sqrt{\log T}$ term: Some reviewers have expressed concern regarding the additive $\sqrt{\log T}$ term. While in practice this term is very small even for large $T$, it is noteworthy that it does not depend on the sub-optimality gaps.
As we explained in Section 1.1, when a regret bound includes an additive term proportional to $D$, even if the number of agents can be chosen, the regret may still be $T^{c}$ for some $c>0$. In our analysis, the additive term is only $\sqrt{\log T}$.

---

### Decision · Program_Chairs · 2025-09-17

**Decision:**

Accept (poster)

**Comment:**

**Summay**

This paper provides individual regret bounds for cooperative multi-agent multi-armed bandits over an underlying topology, independent of graph diameter, and also analyzes trade-offs with message size and communication rounds.

** Strengths**
* The paper has a solid and clear contribution in the relatively crowded space of multi-agent stochastic bandits.
* While fundamental ideas have been proposed in the literature, this paper elegantly leverages these ideas to achieve the improved individual regret in this setting.

** Weaknessess**
* The paper presentation in its current form could be improved in terms of distinguishing the claimed contributions from what is already known in the literature.

** Final justifications**
* There were extensive and engaging discussions between the authors and reviewers regarding the novelty of the ideas and the theoretical analysis. While the authors acknowledged that the paper could better clarify how the proposed algorithms and analysis differ from prior work, the AC and most reviewers remain positive about its merits. Therefore, I recommend acceptance. That said, the paper should go through substantial revision in the camera-ready version to address the reviewers’ comments.